# Trip-to-Gaussian: A Versatile Framework for Unconditional 3D Generation

**Youngwoo Jeon**                                                        *youngwoo.jeon@unist.ac.kr*
*Graduate School of Artificial Intelligence, UNIST*

**Inhyeok Choi**                                                          *inhyeok.choi@unist.ac.kr*
*Graduate School of Artificial Intelligence, UNIST*

**Jaehyeok Shim**                                                          *jh.shim@unist.ac.kr*
*Graduate School of Artificial Intelligence, UNIST*

**Sangjune Park**                                                          *psj9116@unist.ac.kr*
*Graduate School of Artificial Intelligence, UNIST*

**Kyungdon Joo**                                                          *kyungdon@unist.ac.kr*
*Graduate School of Artificial Intelligence, UNIST*

**Reviewed on OpenReview:** *https://openreview.net/forum?id=9uL23Jcjvj*

Figure 1: **Gallery of unconditional 3D generation results on ShapeNet (Chang et al., 2015), G-Objaverse (Deitke et al., 2022), and 3D-FRONT (Fu et al., 2021a) datasets.** Trip2GS enables versatile 3D generation from objects to scenes, producing geometrically coherent high-quality 3D Gaussians.

## Abstract

Unconditional 3D generation is a classical task which focuses on learning the underlying distribution of 3D assets by exploring 3D representations, model architectures, and pipeline design. However, most existing methods limits in versatility, struggling to scale from object-to scene-level generation. Achieving such versatility critically depends on how 3D representations are designed in the latent and output spaces, and how these spaces are connected. In this work, we focus on leveraging the expressiveness of triplane representation together with the fast and high-fidelity 3D Gaussian Splatting (3DGS). Yet, integrating these two representations remains a challenge due to their fundamentally different natures – the structured triplane and unstructured 3DGS. Our core idea is a coarse-to-fine generation scheme

that first extracts reliable geometric priors from triplane and subsequently refines them to capture detailed geometry and textures through 3D Gaussians. To this end, we introduce `Trip-to-Gaussian`, a versatile 3D generation framework that seamlessly integrates two distinct representations. We propose a Gaussian indicator module (GIM) along with surface occupancy fields (SOF), which generates coarse anchor points that serve as reliable geometric prior for 3D Gaussians. Building upon this, we present a point upsampling module (PUM) that maps discontinuous and coarse anchor points into a continuous space, densifying them to ensure fine-grained representation. Extensive experiments demonstrate that our approach outperforms recent methods in both unconditional object and scene generation, establishing a versatile paradigm for 3D generation. Project page: https://vision3d-lab.github.io/trip2gs

## 1 Introduction

Unconditional 3D generation aims to generate realistic samples which follow the learned 3D data distribution, revealing the fundamental generative capacity of a model and laying the groundwork for subsequent advances in conditional generation.

While the rapidly evolving field of conditional generation (Xiang et al., 2025b; Zhao et al., 2025; Li et al., 2025; Chen et al., 2025b; Xiang et al., 2025a; Ye et al., 2025) has recently benefited from increasingly large datasets and computational resources, unconditional generation remains an important and complementary research direction. In particular, this work focuses on improving generation quality under limited data and computational budgets, which is an important issue specifically for user-level programs and content creation, and simulation.

Recent advances in unconditional 3D generation have shown remarkable progress in producing high-quality textured objects (Gao et al., 2022; Müller et al., 2023; Cao et al., 2024; Lan et al., 2024a; Zhou et al., 2025; Zhang et al., 2024; Yang et al., 2025; Rai et al., 2025), yet most previous studies have been limited in *versatility*: scaling from object- to scene-level generation. Scene-level generation is crucial for real-world applications; however, most existing works struggle with scene-level tasks due to their limited capacity and design of the framework.

To design a versatile and high-quality unconditional generative model, there are a few key challenges. 1) *Output representation:* The joint distribution of 3D geometry and texture is complex. This motivates our use of 3D Gaussian Splatting (3DGS) (Kerbl et al., 2023), which naturally encodes geometry and texture jointly. 2) *Latent representation:* Learning a compact, expressive latent for 3D data is crucial. We address this with the efficient and versatile triplane representation (Chan et al., 2022; Peng et al., 2020; Shue et al., 2023). 3) *Bridging representations:* Mapping from a triplane latent to 3DGS remains a challenging, underexplored problem because triplane has a regular grid structure, whereas 3DGS is an irregular and unstructured representation.

In this study, we propose `Trip-to-Gaussian` (`Trip2GS` in short), a novel 3D generation framework that achieves unconditional generation scalable from objects to scenes, without requiring additional data or computation (see Fig. 1). Our framework is motivated by the observation that existing unconditional 3D Gaussian generation methods either model the complex 3DGS distribution (Zhang et al., 2024; Zhou et al., 2025; Rai et al., 2025) or directly generate 3DGS representations from latent features (Zhou et al., 2025; Yang et al., 2025; Ju & Li, 2025). These approaches often struggle to preserve thin structures and fine geometric details, especially when extended to 3D scenes with complicated geometry. To overcome these limitations, we introduce a coarse-to-fine generation scheme as a key to achieving versatility. We first construct a coarse but reliable geometric prior from triplane, and then progressively refine it to capture fine-grained geometry and texture details. This design enables effective modeling of complex 3D structures and seamlessly bridges the structural gap between triplane and 3DGS representations, yielding a versatile solution for unconditional 3D generation.

In particular, `Trip2GS` comprises two main stages: a Trip2GS VAE and a triplane diffusion model. The Trip2GS VAE encodes complex 3D geometry and appearance into a compact triplane latent and decodes it into 3D Gaussians through a coarse-to-fine process. At its core lies the Trip2GS decoder, which comprises

Table 1: **Comparison of recent unconditional 3D Gaussian generation methods.** Prior works require additional 3DGS fitting stage and lack coarse-to-fine design, limiting scene-level scalability. In contrast, `Trip2GS` employs a coarse-to-fine generation scheme to generate 3DGS from triplane features, achieving versatile 3D generation.

| Method | Conference | 3DGS Fitting Stage | Latent Representation | Coarse-to-Fine | Scene-level |
|---|---|---|---|---|---|
| DiffGS (Zhou et al., 2025) | NeurIPS' 24 | Required | Vector | × | × |
| GaussianCube (Zhang et al., 2024) | NeurIPS' 24 | Required | – | × | × |
| L3DG (Roessle et al., 2024) | SIGGRAPH Asia' 24 | Required | Voxel grid | × | △ |
| AtlasGaussians (Yang et al., 2025) | ICLR' 25 | Not required | Point cloud | × | × |
| DirectTriGS (Ju & Li, 2025) | CVPR' 25 | Required | Triplane | × | × |
| UVGS (Rai et al., 2025) | CVPR' 25 | Required | Image | × | × |
| **Trip2GS (Ours)** | – | Not required | Triplane | ✓ | ✓ |

three main components: Gaussian indicator module (GIM), which extracts coarse anchor points from the triplane; point upsampling module (PUM), which generates refined anchor points by densifying and smoothing the coarse anchor points; and the 3DGS decoder, which predicts the final Gaussian parameters. The triplane diffusion then learns the latent distribution, enabling diverse and high-quality generation. Notably, our framework directly generates 3D Gaussians without any external fitting stage or optimization process. Quantitative and qualitative results across multiple datasets demonstrate that `Trip2GS` outperforms existing unconditional generation methods in generation quality and geometric fidelity.

Our key contributions are as follows:

- We propose `Trip2GS`, a versatile 3D generation framework that extends the capability of unconditional generation by seamlessly bridging structured triplane and unstructured 3DGS representations.

- We introduce a coarse-to-fine generation strategy, where GIM extracts coarse yet reliable geometric priors and PUM progressively refines them into high-quality geometry and textures.

- We present SOF, a representation that enables effective 3D Gaussian generation from triplane features, surpassing conventional SDF-based approaches.

- Extensive experiments demonstrate that `Trip2GS` consistently outperforms prior works in both unconditional object and scene generation.

## 2 Related Work

**Latent Representations for 3D Generation**. The design of 3D generative models is closely tied to the choice of intermediate representation. Vector-based methods (Chou et al., 2023; Yariv et al., 2024; Zhou et al., 2025) lack geometric priors and often produce low-quality structures. Voxel-based methods (Shim et al., 2023; Müller et al., 2023; Liu et al., 2023; Zhang et al., 2024; Roessle et al., 2024) intuitively encode 3D information but are limited by prohibitive memory costs at high resolution. Point cloud-based approaches (Zhou et al., 2021; Vahdat et al., 2022; Yang et al., 2025; Lan et al., 2024b) capture geometry well but introduce complexity due to their irregular structure.

In contrast, triplane-based methods (Gao et al., 2022; Gupta et al., 2023; Shue et al., 2023; Lan et al., 2024a; Zhou et al., 2025; Ju & Li, 2025; Shim & Joo, 2024; Cao et al., 2025) can leverage CNN geometric priors through a regular grid while maintaining efficient memory usage. Recent works (Wu et al., 2024c; Lee et al., 2024; Yan et al., 2024; Meng et al., 2025) demonstrate the expressiveness of triplane for generating detailed scene geometry. Building on these strengths, we adopt triplane as our latent representation, enabling `Trip2GS` to efficiently generate high-quality textured 3D objects and scenes.

**3DGS Generation with Grid Representations**. Recently, 3DGS has emerged as a powerful 3D primitive representation, but generating unstructured 3DGS from a structured grid latent is challenging.

Image-based approaches such as Splatter Image (Szymanowicz et al., 2024) and subsequent works (Shen et al., 2024; Tang et al., 2024; Lin et al., 2025; Rai et al., 2025; Henderson et al., 2024) treat each image pixel as a Gaussian element, allowing fast and lightweight 3DGS generation. While efficient, these methods lack a 3D structural prior, leading to inferior geometric consistency. Another related line of work studies 3D generative modeling from 2D supervision only (Szymanowicz et al., 2023; Anciukevičius et al., 2024; Tewari et al., 2023), where the model is trained from posed multi-view images without direct 3D supervision such as meshes, point clouds, or Gaussian annotations. These methods address a more weakly supervised and real-world setting, whereas our work focuses on improving unconditional 3D generation quality when explicit 3D training data are available. Voxel-based approaches provide stronger 3D priors. GaussianCube (Zhang et al., 2024) parameterizes each voxel grid cell as a Gaussian, but its dense voxelization constrains the number of Gaussians and wastes model capacity on learning unnecessary offsets. Another work, L3DG (Roessle et al., 2024), leverages a sparse voxel grid to encode 3D Gaussians in a structured manner and shows 3D scene generation results. While its results highlight the potential of structured Gaussian representations, our work provides a more systematic study on versatility with quantitative evaluations across diverse generation settings. Triplane-based approaches exploit the 2D-structured triplane latent for efficient and robust 3D generation. Unlike conditional generation methods (Zou et al., 2024; Jiang et al., 2024; Barthel et al., 2024), which obtain geometric information from image or text priors, unconditional generation requires extracting geometry directly from the triplane without any external guidance, making the generation of 3D Gaussians substantially more challenging. Several works (Zhou et al., 2025; Ju & Li, 2025) have attempted to generate 3DGS from triplane for unconditional generation, but still suffer from limited geometric fidelity. In contrast to the stochastic sampling of DiffGS (Zhou et al., 2025) and the deformable SDF-based extraction of DirectTriGS (Ju & Li, 2025), we propose to generate 3D Gaussians in a coarse-to-fine manner, which preserves geometry fidelity and enables extension to scene-level generation.

The most related work is TRELLIS (Xiang et al., 2025b), which introduces a structured latent (SLat) by constructing a sparse voxel grid and mapping DINOv2 (Oquab et al., 2024) image features into the voxel space. While TRELLIS has shown both object- and scene-level generation, it trains four independent generative models on separately prepared datasets, resulting in a complex data construction and multi-stage training. In contrast, `Trip2GS` parameterizes triplanes into SOF within a single decoding process, removing additional modeling stages. Furthermore, our 2D triplane representation maintains a lightweight architecture with fewer parameters and more efficient 2D operations than the 3D voxel-based SLat.

Tab. 1 compares recent unconditional 3D Gaussian generation methods. `Trip2GS` is the first framework to introduce a coarse-to-fine scheme, surpassing prior approaches.

## 3 Method

### 3.1 Overview

The proposed `Trip2GS` mainly consists of three parts: Trip2GS encoder/decoder and triplane diffusion (see Fig. 2). Here, we outline each part of our framework and then discuss the details in the following.

**Trip-to-Gaussian Encoder.** The Trip2GS encoder encodes input to a low-dimensional latent triplane. Given a colored point cloud $\mathbf{X} \in \mathbb{R}^{N \times (3+3)}$, where $N$ is the number of points, we first construct a high-resolution triplane feature by projecting the point features onto three orthogonal planes using PointNet (Qi et al., 2017)-based encoder, similar to ConvONet (Peng et al., 2020). This produces a high-resolution triplane feature $\mathbf{Z} \in \mathbb{R}^{3 \times R^2 \times C}$, where $R$ is the resolution of each plane, and $C$ is the channel size (we set $R = 256$, $C = 32$). Subsequently, $\mathbf{Z}$ is compressed into a perceptually rich low-dimensional latent triplane $\hat{\mathbf{Z}} \in \mathbb{R}^{3 \times \hat{R}^2 \times \hat{C}}$ using our triplane encoder, equipped with pooling-based 3D-aware convolutions (Wang et al., 2023; Wu et al., 2024a), where $\hat{R}$ and $\hat{C}$ denote the resolution and channel size of the latent triplane, respectively (we set $\hat{R} = 32$, $\hat{C} = 4$). It should be noted that to generate 3D Gaussians, we directly encode colored point cloud instead of 3DGS, avoiding the time/memory-consuming 3DGS fitting stage.

**Trip-to-Gaussian Decoder.** The Trip2GS decoder generates 3DGS from a triplane feature $\tilde{\mathbf{Z}}$, where we exploit a coarse-to-fine strategy to bridge the gap between triplane and 3DGS representations. Concretely,

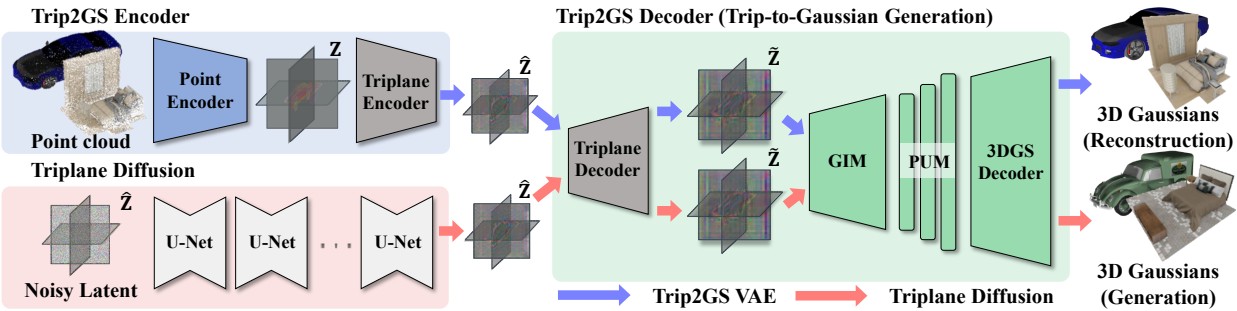

Figure 2: **The overall architecture of the proposed `Trip2GS` framework.** Trip2GS VAE comprises the Trip2GS encoder (blue box) and Trip2GS decoder (green box), which embed an input colored point cloud into a triplane latent representation and then generate 3D Gaussians. Triplane diffusion (red box), trained on triplane latents, generates new triplane latents that are subsequently decoded into 3D Gaussians using the trained Trip2GS decoder.

we upsample $\hat{\mathbf{Z}}$ into a high-resolution triplane feature $\tilde{\mathbf{Z}} \in \mathbb{R}^{3 \times R^2 \times C}$ using a triplane decoder to restore detailed 3D geometry and texture. We project $\tilde{\mathbf{Z}}$ into surface occupancy fields (SOF), our newly introduced concept designed for seamless extraction of 3D Gaussians from the triplane, which implicitly describes near-surface regions. We then regularly sample SOF in 3D space to extract coarse anchor points. Next, we densify and smooth the sparse and discrete coarse anchor points into upsampled ones. Finally, we predict Gaussian parameters by extracting triplane features at each upsampled anchor point to generate final 3D Gaussians.

**Triplane Diffusion.** Triplane diffusion is a variant of the latent diffusion model (Rombach et al., 2022) that generates a new triplane latent $\hat{\mathbf{Z}}$. The network architecture of triplane diffusion is based on a U-Net (Ronneberger et al., 2015) with 3D-aware convolutions (Wang et al., 2023; Wu et al., 2024a) and self-attention blocks (Vaswani et al., 2017), designed to effectively learn the distribution of triplane latents.

## 3.2 Trip-to-Gaussian Generation

To generate 3DGS, we need to extract 3D Gaussians from the triplane feature $\tilde{\mathbf{Z}}$ decoded by the triplane decoder. A key challenge here is that triplane and 3DGS differ in nature. Prior approaches such as optimization- or SDF-guided sampling (Zhou et al., 2025; Ju & Li, 2025) only loosely couple the two representations, resulting in degraded geometry fidelity and limited scalability to complex scene-level generation. To address this limitation, we explicitly decompose this problem into a coarse-to-fine generation process (see Fig. 3). We first extract a coarse yet reliable geometry prior from the triplane, which is then progressively refined to recover fine-grained geometry and high-quality textures. This decomposition effectively bridges the structural gap between the triplane and 3DGS, while enabling versatile 3D generation from objects to scenes.

To achieve this, we propose two complementary modules. The Gaussian indicator module (GIM) extracts coarse anchor points for 3D Gaussians by parameterizing triplane features into surface occupancy fields (SOF). The point upsampling module (PUM) then densifies and smooths these sparse and discrete anchor points to upsampled anchor points. Finally, the 3DGS decoder estimates the full Gaussian parameters from these refined anchor points and triplane features.

**Gaussian Indicator Module.** GIM parameterizes the triplane feature $\tilde{\mathbf{Z}}$ into SOF, a variant of occupancy fields (Mescheder et al., 2019; Peng et al., 2020), to acquire unstructured 3D Gaussians from structured triplane. SOF estimates the probability of 3D Gaussian existence near the surface by assigning 1 for near-surface regions, unlike traditional occupancy fields that serve inside-outside indicators. Subsequently, we sample surface occupancy at regular grid points from the SOF to construct a Gaussian probability volume $\mathbf{V} \in \mathbb{R}^{R_{\text{voxel}}^3}$, a voxel-shaped probability volume with resolution $R_{\text{voxel}}$. Then, we construct coarse anchor points $\mathbf{P}_{\text{coarse}} \in \mathbb{R}^{M_{\text{coarse}} \times 3}$ by filtering voxel cells that exceed threshold probability, where $M_{\text{coarse}}$ is the number of coarse anchor points. For clarity, we present a 2D illustration of the GIM process in Fig. 4.

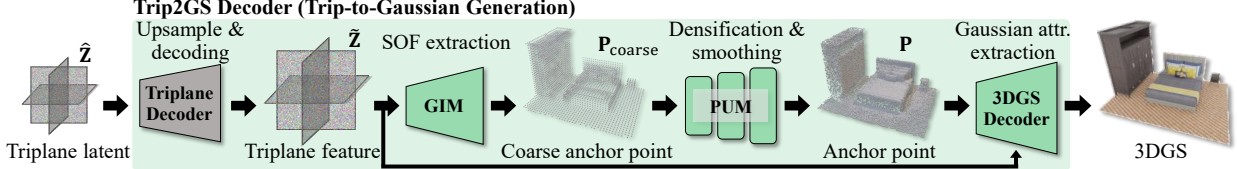

Figure 3: **Illustration of the Trip2GS decoder.** We decode the triplane latent $\hat{\mathbf{Z}}$ into a high-resolution triplane feature $\tilde{\mathbf{Z}}$ using our triplane decoder. Then, GIM generates coarse anchor points $\mathbf{P}_{\text{coarse}}$ from SOF. Next, PUM refines $\mathbf{P}_{\text{coarse}}$ into anchor points $\mathbf{P}$ by increasing their density and smoothness. Finally, the 3DGS decoder extracts Gaussian attributes by sampling $\tilde{\mathbf{Z}}$ at $\mathbf{P}$.

The proposed SOF representation reformulates Gaussian position prediction as a binary classification, improving 3DGS decoding quality by reducing the learning burden on the network. By using SOF, we also eliminate computationally expensive and non-differentiable steps like marching cubes (Lorensen & Cline, 1987). Further discussion on SOF in comparison to other representations is provided in the ablation study.

**Point Upsampling Module.** 3DGS requires hundreds of thousands of Gaussians to represent an object or a scene with detailed geometry and texture. However, the extracted anchor points

Figure 4: **Illustration of the GIM and PUM processes in 2D.** GIM builds Gaussian probability volume $\mathbf{V}$ by regularly sampling SOF in 3D space. Then, it extracts high-probability regions to generate coarse anchor points $\mathbf{P}_{\text{coarse}}$. Finally, PUM densifies and smooths $\mathbf{P}_{\text{coarse}}$ to produce final anchor points $\mathbf{P}$.

from GIM are sparse, limiting the ability to capture fine details. We also observed that directly querying triplane features at regular anchor points from GIM causes unexpected artifacts, such as grid pattern and texture abruption in rendering results, which degrade the visual quality (refer to appendix).

To address these issues, we present an MLP-based PUM that maps discrete points to a continuous space and densifies the anchor points, both crucial for high-quality 3D Gaussian generation. Specifically, PUM refines coarse anchor points $\mathbf{P}_{\text{coarse}}$ into anchor points $\mathbf{P} \in \mathbb{R}^{M \times 3}$ by performing upsampling and nonlinear mapping, where $M$ is the number of anchor points after upsampling, satisfying $M \geq M_{\text{coarse}}$. As a result, we can generate densified and continuous anchor points for 3D Gaussians, enhancing the geometric accuracy and texture quality of 3DGS (we visualize a 2D illustration of the PUM process in Fig. 4).

**3DGS Decoder.** To generate realistic textures, we present a 3DGS decoder to estimate the Gaussian parameters for each upsampled anchor point from PUM. Concretely, the 3DGS decoder first aggregates the triplane feature $\tilde{\mathbf{Z}}$ of the corresponding anchor point by extracting features from each of the three planes. Next, it predicts Gaussian parameters by leveraging attribute-specific fully connected layers, inspired by (Barthel et al., 2024; Lu et al., 2024; Zou et al., 2024). Each 3D Gaussian is parameterized by a 14-dimensional attribute vector composed of spatial offset, color, opacity, and a covariance matrix that defines the shape and orientation of the Gaussian. This structured approach enables a more effective disentanglement of spatial and appearance features, resulting in more precise and visually coherent 3D Gaussian representations. Please refer to the appendix for further details on network architecture.

### 3.3 Training Objective

We train our VAE with three primary loss terms: rendering loss $\mathcal{L}_{\text{render}}$, geometry loss $\mathcal{L}_{\text{geo}}$, and regularization $\mathcal{L}_{\text{reg}}$:

$$\mathcal{L}_{\text{total}} = \mathcal{L}_{\text{render}} + \mathcal{L}_{\text{geo}} + \mathcal{L}_{\text{reg}}. \tag{1}$$

By optimizing $\mathcal{L}_{\text{total}}$, our Trip2GS VAE learns to generate high-quality 3DGS while preserving geometric structure and ensuring a smooth and stable latent space.

**Rendering Loss.** We follow the original 3DGS (Kerbl et al., 2023) to ensure high-fidelity 3DGS reconstruction from colored point cloud input, applying pixel-wise L2 loss $\mathcal{L}_{\text{L2}}$ and SSIM loss $\mathcal{L}_{\text{SSIM}}$. In addition, we adopt a mask loss $\mathcal{L}_{\text{mask}}$, which is an L2 loss between the predicted and GT binary mask, and LPIPS (Zhang et al., 2018) loss $\mathcal{L}_{\text{LPIPS}}$ for perceptual quality. The rendering loss is:

$$\mathcal{L}_{\text{render}} = \mathcal{L}_{\text{L2}} + \mathcal{L}_{\text{SSIM}} + \mathcal{L}_{\text{mask}} + \mathcal{L}_{\text{LPIPS}}. \tag{2}$$

**Geometry Loss.** We employ two additional loss terms for geometry loss. First, we supervise GIM using binary cross-entropy loss $\mathcal{L}_{\text{BCE}}$ to ensure that GIM predicts correct surface occupancy fields:

$$\mathcal{L}_{\text{GIM}} = \mathcal{L}_{\text{BCE}}(\hat{v}_i, v_i), \tag{3}$$

where $\hat{v}_i \in \hat{\mathbf{V}}$ and $v_i \in \mathbf{V}$ are predicted and the ground truth Gaussian existence probability, respectively. Second, we apply Chamfer distance $\mathcal{L}_{\text{CD}}$ to compare ground truth surface geometry (*i.e.*, input point cloud) with both the anchor points from PUM and Gaussian positions from the 3DGS decoder. The geometry loss is formulated as:

$$\mathcal{L}_{\text{geo}} = \lambda_{\text{GIM}}\mathcal{L}_{\text{GIM}} + \lambda_{\text{CD}}\mathcal{L}_{\text{CD}}, \tag{4}$$

where $\lambda_{\text{GIM}}$ and $\lambda_{\text{CD}}$ are set to 1 and 100, respectively.

**Regularization.** To stabilize the diffusion training and encourage visually smooth rendering, we adopt total variation (TV) regularization (Rudin & Osher, 1994) on both latent space and image space, reducing undesired high-frequency artifacts. Additionally, KL divergence regularization $\mathcal{L}_{\text{KL}}$ is applied to constrain the latent space to follow a Gaussian distribution, which also accelerates the diffusion training process. The regularization term is formulated as:

$$\mathcal{L}_{\text{reg}} = \lambda_{\text{TV}}\mathcal{L}_{\text{TV}} + \lambda_{\text{KL}}\mathcal{L}_{\text{KL}}, \tag{5}$$

where $\lambda_{\text{TV}}$ and $\lambda_{\text{KL}}$ are set to $1e-4$ and $1e-6$, respectively.

### 3.4 Training Strategy

For effective training of the Trip2GS VAE, we introduce two learning strategies. At the same time, we emphasize that Trip2GS VAE is still trained within a *unified training framework*: all modules are optimized as part of a single point-to-3DGS pipeline, rather than requiring additional preprocessing or being trained disjointly in separate pipelines. Our training strategy therefore aims not to separate the framework itself, but to stabilize optimization within this unified pipeline.

**Gradient Control Strategy.** First, to ensure GIM provides a robust geometry prior, we employ a gradient control strategy. Specifically, we detach the coarse anchor points $\mathbf{P}_{\text{coarse}}$ obtained from GIM from gradient computation from other losses except for $\mathcal{L}_{\text{GIM}}$. Without this detachment, gradients from the other losses propagate back to GIM, where we empirically found it interferes with the learning of geometric priors. Since GIM is intended to provide robust geometric priors for object representation, we enforce that it is trained solely with the occupancy loss, without interference from other gradients.

**Warm-Up Training Phase.** Second, to stabilize early-stage training, we train the 3DGS decoder and GIM separately in the initial iterations. Early in training, GIM does not yet provide accurate geometric information, leading to unstable supervision signals for 3DGS generation. These incorrect surface priors fail to provide meaningful guidance to the 3DGS decoder, which negatively impacts the learning process. To mitigate this, in the early training phase, we sample query points for the 3DGS decoder from the ground-truth point cloud rather than using anchor points extracted from GIM.

These two strategies ensure the stable training process of Trip2GS VAE, allowing each module to focus on its designated task, geometry learning for GIM and texture modeling for the 3DGS decoder, which ultimately leads to higher-quality outputs.

Table 2: **Quantitative comparison with other methods on ShapeNet and 3D-FRONT datasets.**
We depict cases with excessively large values ($> 200$) using a triangle ($\triangle$).

| Method | Chair | | | Airplane | | | Car | | | Bedroom | | |
|---|---|---|---|---|---|---|---|---|---|---|---|---|
| | FID | KID | C-FID | FID | KID | C-FID | FID | KID | C-FID | FID | KID | C-FID |
| GET3D | 75.89 | 48.71 | 15.23 | 72.61 | 57.66 | 7.18 | 109.35 | 83.83 | 14.83 | $\triangle$ | $\triangle$ | 60.54 |
| DiffGS | 75.21 | 55.45 | 17.42 | 65.18 | 52.76 | 10.57 | 128.01 | 114.64 | 14.94 | $\triangle$ | $\triangle$ | 58.87 |
| GaussianCube | 23.45 | 14.40 | 6.24 | 17.84 | 11.92 | 3.28 | 20.92 | 13.26 | 4.43 | 96.78 | 97.33 | 22.19 |
| Atlas Gaussians | **13.46** | 4.61 | 2.72 | 12.87 | 6.34 | 2.76 | 19.79 | 12.23 | 3.36 | 74.86 | 62.69 | 15.18 |
| Trip2GS (ours) | 14.13 | **3.97** | **1.96** | **12.24** | **5.76** | **1.28** | **18.63** | **9.56** | **2.54** | **16.30** | **9.10** | **2.87** |

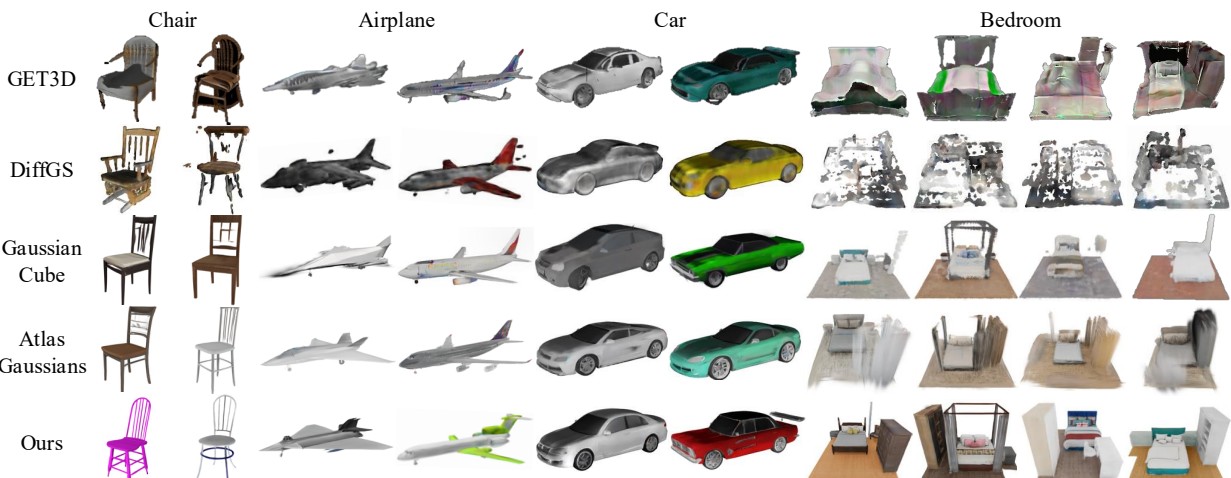

Figure 5: **Qualitative comparisons of unconditional generation on ShapeNet and 3D-FRONT.**

### 3.5 Triplane Diffusion

Unlike conventional diffusion models (Ho et al., 2020), our triplane diffusion is supervised to estimate denoised latent triplane $\hat{\mathbf{Z}}_0$ instead of the applied noise, following (Zhang et al., 2024):

$$\mathcal{L}_{\text{diff}} = \left\| \mu_\theta(\hat{\mathbf{Z}}_t, t) - \hat{\mathbf{Z}}_0 \right\|_2, \tag{6}$$

where $\mu_\theta$ is our triplane diffusion network, $t$ is diffusion timestep. We use a linear noise schedule combined with the Heun sampler from (Karras et al., 2022), which allows fast convergence and high-quality sampling.

For the denoising network, we employ a U-Net (Ronneberger et al., 2015) based architecture following the design of ADM (Nichol & Dhariwal, 2021; Dhariwal & Nichol, 2021). We adopt ResNet (He et al., 2016) blocks consisted of 3D-aware operations (Wang et al., 2023; Wu et al., 2024a) same as triplane encoder of Trip2GS VAE, followed by self-attention block applied to channel-wise concatenated triplanes. By training triplane diffusion, we generate high-fidelity triplane latents, which are subsequently decoded by the Trip2GS decoder to produce diverse and high-quality 3D objects and scenes. Further implementation details can be found in appendix.

## 4 Experiments

We conduct various experiments to demonstrate the effectiveness and versatility of Trip2GS. Sec. 4.1 describes the datasets, evaluation metrics, and baseline models used for comparison. In Sec. 4.2, we evaluate our method for unconditional 3D generation across multiple datasets and demonstrate that it outperforms existing baselines both qualitatively and quantitatively. Sec. 4.3 provides an ablation study on the key components of Trip2GS VAE, analyzing their contributions to high-quality 3DGS generation.

Table 3: **Quantitative comparison with other methods on G-Objaverse dataset.**

| Method | Plants | | | Food | | | Animals | | |
|---|---|---|---|---|---|---|---|---|---|
| | **FID** | **KID** | **C-FID** | **FID** | **KID** | **C-FID** | **FID** | **KID** | **C-FID** |
| GET3D | 122.69 | 110.03 | 15.97 | 108.82 | 79.54 | 18.79 | 104.69 | 83.22 | 14.59 |
| DiffGS | 160.78 | 111.00 | 22.18 | 171.39 | 73.48 | 32.70 | 199.53 | 100.35 | 33.98 |
| GaussianCube | 127.39 | 98.15 | 17.07 | 96.03 | 67.98 | 15.46 | 131.05 | 99.38 | 16.32 |
| Atlas Gaussians | 89.70 | 56.80 | 13.20 | 73.85 | 43.41 | 13.96 | 87.98 | 65.22 | 12.83 |
| `Trip2GS` (ours) | **67.09** | **46.56** | **7.75** | **55.21** | **31.91** | **10.45** | **67.25** | **49.74** | **8.31** |

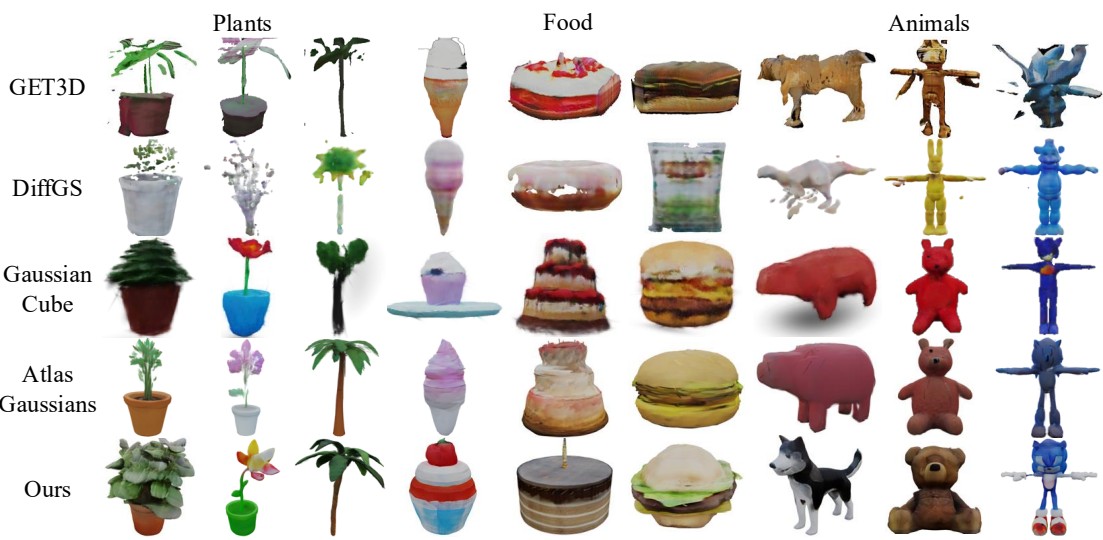

Figure 6: **Qualitative comparisons of unconditional generation on G-Objaverse.**

*It is strongly recommended to refer to the appendix for additional results and evaluations, including conditional generation, a deeper analysis on versatility, and further comparisons with other baselines such as TRELLIS (Xiang et al., 2025b).* In addition, we provide a supplementary video which visualizes and compares our 3D generation results with other baselines.

## 4.1 Dataset, Metrics, and Baselines

**Dataset.** We adopt two datasets for object-level generation. ShapeNetv1 (Chang et al., 2015) serves as a widely used benchmark dataset in unconditional 3D generation, providing clean and category-consistent 3D objects. G-Objaverse (Zuo et al., 2024; Qiu et al., 2024; Deitke et al., 2022; 2023) is a large-scale collection of diverse and high-quality 3D assets, allowing us to evaluate the versatility on detailed geometries and textures. For scene-level generation, we employ 3D-FRONT (Fu et al., 2021a;b), consisting of synthetic indoor scenes with various room layouts, furniture, and styles, to assess the generation capability on complex scenes.

**Metrics.** We use two widely adopted metrics in unconditional 3D generation: FID@50K (Heusel et al., 2017) and KID@50K (Bińkowski et al., 2018). These metrics measure the distributional similarity between 50K rendered images from generated samples and ground-truth images. We also use CLIP-FID@50K (Parmar et al., 2022) (abbreviated as C-FID) as a more semantically aware measurement.

**Baselines.** As baselines, we select state-of-the-art methods that are capable of generating both geometry and texture while supporting training on high-resolution images. GET3D (Gao et al., 2022) is a GAN-based approach that uses DMTet (Shen et al., 2021) for high-resolution 3D generation, often outperforming recent methods in various benchmarks. DiffGS (Zhou et al., 2025), GaussianCube (Zhang et al., 2024),

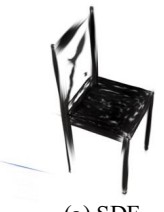 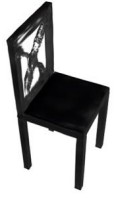 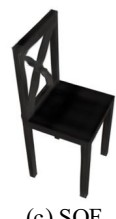

(a) SDF      (b) DMTet      (c) SOF

Figure 7: **Comparison with other geometry representations.** Our SOF demonstrates robustness in the reconstruction of sharp and thin shapes.

Table 4: **Comparison of geometry representations for chair category on ShapeNet test set.** Trip2GS VAE is trained for 300k iterations using each geometry representation.

| Representation | PSNR | SSIM | LPIPS |
|:---:|:---:|:---:|:---:|
| SDF | 20.26 | 0.8411 | 0.2111 |
| DMTet | 23.49 | 0.8960 | 0.1188 |
| SOF (ours) | **24.52** | **0.9125** | **0.1005** |

and AtlasGaussians (Yang et al., 2025) utilize 3DGS as a 3D primitive. DiffGS encodes precomputed 3D Gaussians into triplane and subsequently compresses the triplane to a latent vector, while GaussianCube directly allocates 3D Gaussians to a fixed-size voxel grid (*e.g.*, 32). AtlasGaussians adopts a point cloud-based latent representation to capture the object geometry better.

### 4.2 Generation Results

**3D Object Generation.** Tab. 2 and Fig. 5 present results on three ShapeNet categories: chair, airplane, and car. Across all metrics, `Trip2GS` outperforms existing baselines, producing complete geometry and sharper textures. Notably, our model captures thin structures such as chair backs and legs, demonstrating that the coarse-to-fine design effectively preserves high-frequency geometric details.

We further evaluate `Trip2GS` on G-Objaverse, a large-scale dataset with diverse and high-quality 3D assets across plants, food, and animals categories (see Tab. 3 and Fig. 6). Our method consistently achieves superior visual quality even on complex 3D assets, accurately modeling intricate structures such as leaves and petals.

Overall results confirm that `Trip2GS` generalizes beyond simple shapes, capturing the distribution of diverse, high-quality 3D assets.

**3D Scene Generation.** To assess versatility for more complex geometries, we evaluate our method on the bedroom category of 3D-FRONT, which offers diverse layouts and styles with sufficient samples (3,886 after preprocessing following (Paschalidou et al., 2021)) for training an unconditional model (see Tab. 2 and Fig. 5). `Trip2GS` surpasses all SOTA baselines by a clear margin, both quantitatively and qualitatively. Methods such as GET3D and DiffGS, which rely on simple vector-shape latents, fail to capture the spatial complexity of scenes and produce unrealistic results. Although GaussianCube and AtlasGaussians show moderate results, they still struggle with fine object details and secondary elements such as wardrobes and lamps. In contrast, `Trip2GS` accurately reconstructs complex scene geometry, preserves spatial consistency, and generates realistic textures for both major and minor objects.

These results confirm that our coarse-to-fine design effectively models the complex 3D structure, providing a unified and versatile framework for unconditional 3D generation from single object to multi-object scenes.

### 4.3 Ablation Study

**Intermediate Geometry Representation.** The core of our coarse-to-fine generation scheme lies in extracting a reliable geometric prior through GIM. In this process, SOF plays a critical role in indicating the spatial regions where 3D Gaussians should exist, serving as an intermediate geometry representation between the structured triplane and unstructured 3DGS. To assess the effectiveness of our SOF, we compare with existing geometry representations commonly used in prior works: signed distance fields (SDF) and deep marching tetrahedra (DMTet) (Shen et al., 2021). For fair comparison, we evaluate a variant without PUM, where anchor points from GIM are directly used as query points for the 3DGS decoder. Since SDF requires

computationally expensive marching cubes algorithm (Lorensen & Cline, 1987) to obtain the object surface, we train SDF and 3DGS decoder separately.

Unlike SDF, DMTet enables fast mesh extraction via the differentiable marching tetrahedra algorithm, allowing the predicted surface points to be used as query points for the 3DGS decoder during training. This approach is similar to DirectTriGS (Ju & Li, 2025), which employs FlexiCubes (Shen et al., 2023) to generate anchor points. However, extracting mesh solely for point sampling introduces unnecessary computational overhead, reducing the overall efficiency of the process. To address these limitations, we propose SOF, a compact yet reliable intermediate geometry representation for 3DGS generation.

As shown in Tab. 4 and Fig. 7, our SOF achieves the highest reconstruction quality. Since the primary goal of GIM is to provide reliable surface regions where 3D Gaussians should exist, the key factor is the *fidelity* of the extracted geometry prior. Continuous representations such as SDF and DMTet are effective for smooth surface reconstruction, but they tend to oversmooth thin or high-frequency structures, leading to missing or disconnected regions (Fig. 7-(a), (b)). In contrast, the proposed SOF representation, trained with a classification objective, produces more stable surface region predictions even for thin geometries (Fig. 7-(c)). Although SOF generates fewer and discrete anchor points, their reliability enables superior 3D Gaussian generation, validating SOF as an effective intermediate representation for bridging triplane and 3DGS.

Table 5: **Ablation study on PUM for the ShapeNet airplane category after 500k iterations.** $k$ denotes upsampling ratio.

| Point mapping | $k$ | FID | KID | CLIP-FID |
|:---:|:---:|:---:|:---:|:---:|
| X | 1 | 35.42 | 27.86 | 2.53 |
| O | 1 | 23.11 | 13.65 | 2.02 |
| O | 4 | 15.86 | 8.74 | 1.69 |
| O | 16 | **12.24** | **5.76** | **1.28** |

Table 6: **Ablation study on TV regularization for the ShapeNet chair category after 1M training iterations.**

| $\mathcal{L}_{\text{TV-image}}$ | $\mathcal{L}_{\text{TV-latent}}$ | FID | KID | CLIP-FID |
|:---:|:---:|:---:|:---:|:---:|
| | | 17.01 | 4.44 | 2.15 |
| ✓ | | 15.63 | 4.28 | 2.06 |
| | ✓ | 14.78 | 4.15 | 2.02 |
| ✓ | ✓ | **14.13** | **3.97** | **1.96** |

**Point Mapping and Upsampling.** The placement and number of 3D Gaussians are crucial for capturing geometric and textural details. Since the anchor points extracted from GIM are discontinuous and sparse, we introduce PUM, which maps them into a continuous space and upsamples them into a denser point set aligned with the object surface. Tab. 5 shows that PUM consistently improves generation quality. Due to its lightweight MLP-based design, PUM can efficiently densify Gaussians with minimal computational overhead (refer to the appendix).

**Total Variation Regularization.** Tab. 6 presents an ablation study analyzing the effect of TV regularization loss when applied to both image and latent spaces. Applying it to both domains consistently yields the best performance. We attribute this improvement to the sparsity of 3D space, where reducing noisy features in empty regions enhances stability and overall generation quality.

## 5 Conclusion

We have proposed `Trip2GS`, a versatile 3D generative framework that bridges structured triplane and unstructured 3D Gaussian Splatting (3DGS), enabling high-quality generation from individual objects to complex scenes. The core of our approach lies in a coarse-to-fine generation scheme that first extracts reliable geometric priors from triplane and progressively transforms them into 3DGS for detailed geometry and textures. To realize this, we proposed Gaussian indicator module (GIM) and surface occupancy fields (SOF), which produce faithful surface regions outperforming conventional SDF-based methods. We further presented point upsampling module (PUM), which refines coarse anchor points to capture the details, consistently enhancing generation quality. Extensive experiments validated that `Trip2GS` scales across diverse 3D domains, establishing a strong foundation for future advancements in this area.

## Acknowledgments

This work was supported by the Institute of Information & Communications Technology Planning & Evaluation (IITP) grant funded by the Korea government (MSIT) (No.RS-2020-II201336, Artificial Intelligence Graduate School Program (UNIST); No.RS-2025-25442824, AI Star Fellowship Program (UNIST); No.RS-2026-25507551, Development of Egocentric Data Sensing and Spatial Immersive Experience Technology), the National Research Foundation of Korea (NRF) grant funded by the Korea government (MSIT) (No.RS-2025-02216916), and the AI Computing Infrastructure Enhancement (GPU Rental Support) User Support Program funded by the Ministry of Science and ICT (MSIT), Republic of Korea.

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

# Appendix

## Overview

In appendix, we provide further details on the following topics:

- Section A offers an in-depth discussion of the experimental setting, dataset preparation process, model architecture, and training strategies.

- Section B provides an extended comparison with TRELLIS, highlighting architectural differences, reconstruction results, and generation challenges under the unconditional setting.

- Section C presents additional details and results for the various conditional generation tasks, including implementation details for conditional generation cases.

- Section D reports a comprehensive efficiency analysis covering training time, memory usage, and inference latency across all baselines.

- Section E discusses and analyzes the versatility of the proposed approach, including dataset complexity and the pipeline designs of both the baselines and our method.

- Section F compares our method with baselines in terms of geometric quality, considering both the intermediate geometry representation and output 3D Gaussians.

- Section G includes further ablation studies focusing on the scene completion application and upsampling effects of the proposed PUM.

- Section H provides additional quantitative and qualitative results on ShapeNet, 3D-FRONT, G-Objaverse, and the real-world MVImgNet dataset.

- Section I discusses additional issues, including memorization and the key components of our approach, and outlines potential future research directions.

- In addition, we provide a supplementary video that visualizes and compares the generated 3D results.

## A  Implementation Details

We trained all networks on 8 NVIDIA A100 GPUs using the PyTorch framework. Experiments were conducted on a system equipped with an AMD EPYC 7742 64-Core Processor, 1.0 TiB of RAM, and running Ubuntu 20.04 LTS. To ensure reproducibility and fairness, we fixed the random seeds to 0 for both NumPy and PyTorch to eliminate performance variations caused by stochastic factors.

### A.1  Dataset Preparation

We utilize colored point cloud as an input for our Trip2GS VAE. To construct an input colored point cloud, we follow the data construction step of Shap-E (Jun & Nichol, 2023). To construct the input colored point cloud, we first render 20 RGB and depth images of a given mesh from randomly sampled viewpoints on a sphere. Using these rendered images, we unproject each valid pixel into 3D space based on its depth value. Specifically, for each pixel $(u, v)$ in the image, we compute its corresponding 3D ray direction and scale the depth value accordingly. The 3D coordinate $\mathbf{p} \in \mathbb{R}^3$ is obtained as $\mathbf{p} = \mathbf{o} + s \cdot \mathbf{d}$, where $\mathbf{o} \in \mathbb{R}^3$ denotes the camera origin, $\mathbf{d} \in \mathbb{R}^3$ is the unit ray direction, and $s \in \mathbb{R}$ represents the depth scale factor. To ensure that only meaningful surface points are retained, we filter out pixels with infinite depth (*e.g.*, background pixels) and those with low transparency ($\alpha < 1$). Finally, after unprojecting all valid pixels into 3D space, we randomly sample 100k points from the extracted colored point cloud to construct the set of input points.

### A.2 Details of Trip-to-Gaussian VAE

**Hyperparameters.** We train the Trip2GS decoder for 300k iterations with batch size of 8 using the AdamW optimizer with a cosine annealing learning rate scheduler, setting the initial learning rate to $1e-4$. Training with full iterations takes approximately 2.5 days on 8 A100 GPUs. The resolution of GIM is fixed at 64, ensuring computational efficiency and reliability of geometric prior for 3DGS generation. Additionally, to effectively generate a sufficient number of 3D Gaussians for detailed representation, we apply different point upsampling ratios based on the complexity of the geometry and texture of an object/scene. The upsampling ratio is set differently according to the scale and texture complexity of the object: 4 for chairs, 8 for bedrooms, and 16 for airplane and car categories. These configurations allow our model to adaptively increase the density of 3D Gaussians, ensuring high-quality reconstruction across various object and scene categories.

**Gaussian Indicator Module (GIM).** After the triplane decoder, the triplane channel is increased from 4 to 256 using a U-Net style network with triplane convolutions, making the shape to (32, 32, 256). GIM first upsamples triplane features to (64, 64, 128) to match the resolution of $R_{voxel}$. Uniform voxel points query triplane features, which are then passed to a 5-layer MLP with sigmoid activation to predict SOF values; values above 0.5 yield $M_{coarse}$ coarse anchor points.

**Point Upsampling Module (PUM).** PUM is a 3-layer MLP that maps $(M_{coarse}, 3)$ to $(M_{coarse}, 3k)$ and reshape to $(M, 3)$ refined dense anchor points, where $M = M_{coarse} \times k$.

**3DGS Decoder.** Our 3DGS decoder consists of four sub-decoders, each responsible for predicting different attributes of 3D Gaussians: offset, color, covariance, and opacity (see Fig. 8).

- **Offset o**. We apply a sigmoid function to predict the offset, scale it to the range $[-1/32, 1/32]$, and add it to the anchor point position to determine the final 3D Gaussian center: $\mathbf{p}_{\text{final}} = \mathbf{p} + (\sigma(\mathbf{o}) - 0.5)/64$, where $\mathbf{p} \in \mathbf{P}$ is the anchor point position, $\mathbf{o}$ is the predicted offset, and $\sigma(\cdot)$ denotes sigmoid function.

- **Color (RGB) c**. The feature values are directly used as RGB values without additional transformation: $\mathbf{c} = \mathbf{f}_{\text{RGB}}$, where $\mathbf{f}_{\text{RGB}}$ is the extracted triplane feature.

- **Opacity $\alpha$**. A sigmoid function is applied to ensure the opacity values are within the range $[0, 1]$: $\alpha = \sigma(o_\alpha)$, where $o_\alpha$ is the predicted opacity.

- **Scale $s$**. A sigmoid function is applied to predict the scale, followed by a multiplication by 0.1 to stabilize 3D Gaussian optimization: $s = 0.1 \cdot \sigma(o_s)$, where $o_s$ is the predicted scale.

- **Rotation q**. The predicted rotation values are normalized to form a quaternion representation, ensuring a valid rotation: $\mathbf{q} = \frac{\mathbf{o}_q}{\|\mathbf{o}_q\|}$, where $o_q$ is the predicted rotation vector, and $\|\cdot\|$ represents the Euclidean norm.

### A.3 Details of Triplane Diffusion

For the denoising network, we employ a U-Net (Ronneberger et al., 2015) based architecture following the design of ADM (Nichol & Dhariwal, 2021; Dhariwal & Nichol, 2021). It is composed of four hierarchical levels, with each level containing two residual blocks, each followed by a channel-wise self-attention block. For self-attention and cross-attention in conditional generation, the number of attention heads per channel is set to 64. At each level, the spatial resolution is halved while the number of channels is doubled. We use the batch size of 64 and AdamW optimizer with a cosine annealing learning rate scheduler, setting the learning rate to $2e-5$. The training iterations for triplane diffusion are set as follows: 500k iterations for the airplane category, 600k for car, and 1M for both chair and bedroom. The training takes about 1.5 days for the airplane and car categories and 3 days for the chair and bedroom categories.

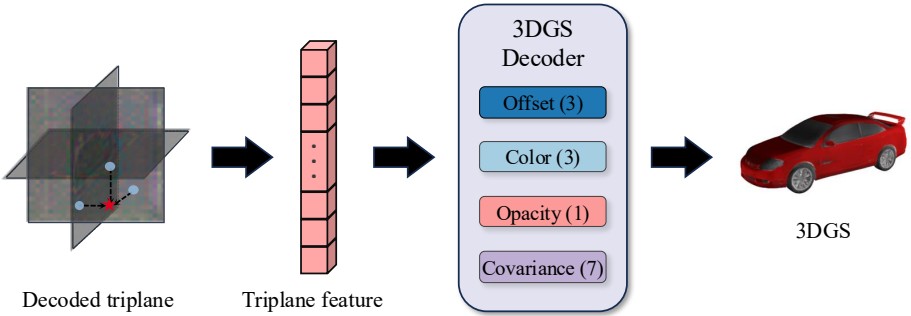

Figure 8: **Network architecture of 3DGS decoder.** The 3DGS decoder consists of four linear layers, each responsible for decoding a specific Gaussian attribute.

## B  Comparison to TRELLIS

In the main paper, we show that `Trip2GS` outperforms prior methods in unconditional 3D generation, achieving superior quality in both geometry and texture. A key contributor to this improvement is our coarse-to-fine generation strategy built on the Gaussian indicator module (GIM) and point upsampling module (PUM), which effectively bridge structured triplane and unstructured 3DGS representations.

One of the most relevant approaches to ours is TRELLIS (Xiang et al., 2025b), a large-scale conditional generation framework. TRELLIS introduces the structured latent (SLat) representation, which encodes sparse voxel grids with DINOv2 (Oquab et al., 2024) image features of a 3D asset. Conceptually, TRELLIS can be regarded as a heavyweight multi-stage 3D counterpart to ours – it explicitly constructs a voxel grid representing surface occupied regions, whereas `Trip2GS` implicitly learns such regions within an unified generative pipeline. To generate 3D Gaussians from its sparse structure, TRELLIS extracts DINOv2 features from input images and embeds them into the voxel space, followed by training an additional VAE model. In this section, we focus on comparing `Trip2GS` and TRELLIS with respect to the effect of unified training of Trip2GS against multi-stage training of TRELLIS.

Since TRELLIS is built for large-scale conditional generation, we adapt it for unconditional generation to enable fair comparison. In addition, we found that the TRELLIS preprocessing pipeline implicitly assumes a fixed normalization range for input geometry (e.g., within [-0.5, 0.5]). Because the datasets used in our experiments have different intrinsic scales and camera parameters, we adjusted the preprocessing step of TRELLIS to match the scale of each dataset, so that the comparison is not affected by scale mismatches in data construction. All models are trained under comparable computational budgets and training schedules to ensure a fair efficiency comparison (see Tab. 11).

Table 7: **PSNR comparison of TRELLIS variants and our model across datasets.** TRELLIS* indicates our reproduced version of original TRELLIS under comparable training resources.

| Method | ShapeNet | | | Objaverse | | | 3D-FRONT |
|---|---|---|---|---|---|---|---|
| | Airplane | Car | Chair | Animals | Food | Plants | Bedroom |
| TRELLIS* | **34.56** | **31.03** | 29.15 | **30.08** | **28.75** | 27.93 | 25.49 |
| Trip2GS (ours) | 32.84 | 30.13 | **30.24** | 29.88 | 27.80 | **29.20** | **26.00** |

### B.1  Reconstruction

We first compare `Trip2GS` with TRELLIS on the task of VAE reconstruction, where the goal is to generate 3D Gaussians from a constructed input representation of a 3D asset. For a fair comparison, we align the TRELLIS setting to our `Trip2GS` VAE in terms of latent size (TRELLIS: $16 \times 16 \times 16 \times 4$ vs. ours: $32 \times 32 \times 4 \times 3$), model size (TRELLIS: 229.31M vs. ours: 227.47M), and upsampling ratio (4, 8, and 16

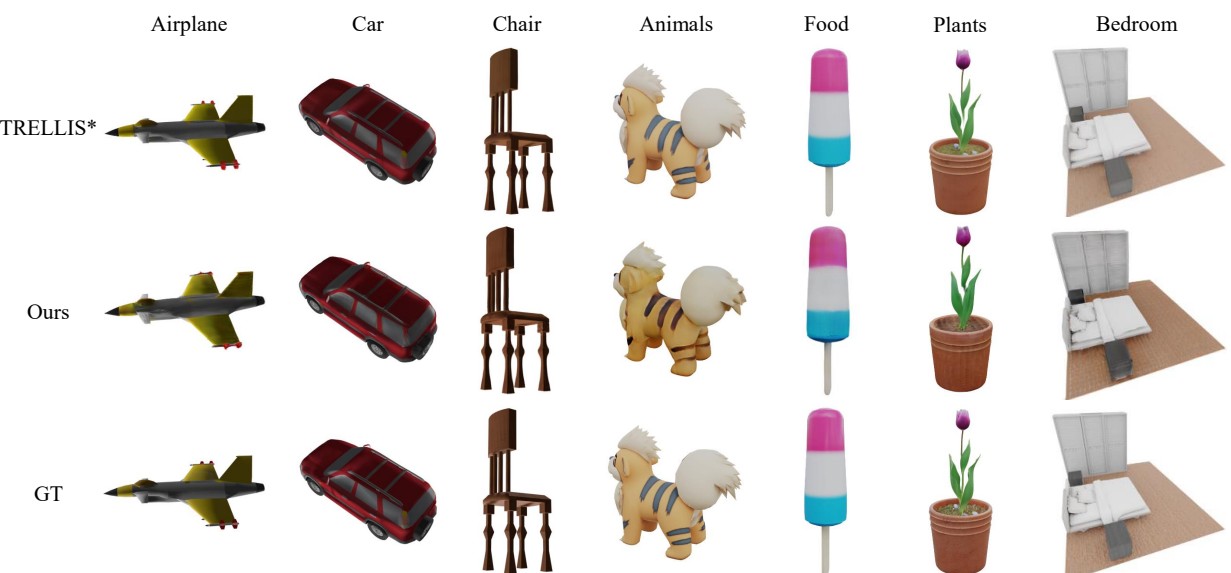

Figure 9: **Qualitative comparison of `Trip2GS` with TRELLIS on VAE reconstruction.** TRELLIS* denotes our reproduced version of the original TRELLIS (Xiang et al., 2025b), used to ensure a fair comparison in the unconditional generation setting (see Sec. D).

depending on the category). The results are shown in Fig. 9 and Tab. 7. Without relying on additional foundation models (e.g., DINOv2) or extra preprocessing, `Trip2GS` achieves reconstruction performance comparable to that of TRELLIS. Moreover, although we apply total variation (TV) regularization – which, slightly reduces reconstruction fidelity while improving generation quality – our model remains competitive overall. These results highlight the efficiency and representational strength of our unified coarse-to-fine design.

Table 8: **Generation comparison under different TRELLIS settings and ours on bedroom category.**

| Method | Batch size | # Params. (M) | FID | KID | C-FID | Train time (d) |
|---|---|---|---|---|---|---|
| TRELLIS*-B | 16 | 285.36 | 38.56 | 34.12 | 8.47 | 4.09 |
| TRELLIS*-L | 64 | 549.98 | 37.77 | 33.51 | 7.52 | 20.08 |
| Trip2GS (Ours) | 64 | 508.50 | **18.43** | **9.10** | **3.21** | 4.31 |

## B.2 Generation

For generation, we conduct a fair comparison from two complementary perspectives. The first model, `TRELLIS*-B`, is designed for a comparison under a similar training-resource budget, since training time and memory are especially important in the limited-resource generation setting considered in this work. For this setting, we use the base flow-matching model of TRELLIS (285.36M parameters, compared with 508.50M for ours), with batch size 16 (64 for ours) and 500K training iterations for both models. The second model, `TRELLIS*-L`, is designed for a comparison from the perspective of model scale and batch size. Starting from `TRELLIS*-B`, we increase the number of blocks from 12 to 20, the number of heads from 12 to 16, and the

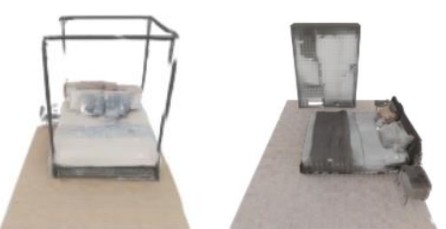

Figure 10: **Generation results of TRELLIS on bedroom category.** TRELLIS generates better results than other baselines, but is inferior to Trip2GS.

Table 9: **Quantitative comparison of `Trip2GS` and TRELLIS for unconditional generation.** TREL-LIS* denotes TRELLIS*-B, trained under comparable training resources.

| Method | Metric | Airplane | Car | Chair | Animals | Food | Plants | Bedroom | Mean |
|---|---|---|---|---|---|---|---|---|---|
| TRELLIS* | FID | 21.03 | 27.96 | 16.24 | 91.16 | 88.20 | 109.11 | 37.03 | 55.82 |
| | KID | 14.50 | 21.84 | 13.12 | 67.63 | 54.70 | 65.69 | 33.61 | 38.73 |
| | C-FID | 4.84 | 5.99 | 4.35 | 14.31 | 15.98 | 13.15 | 8.30 | 9.56 |
| Trip2GS (ours) | FID | **12.24** | **18.63** | **14.13** | **67.25** | **55.21** | **67.09** | **16.30** | **35.84** |
| | KID | **5.76** | **9.56** | **3.97** | **49.74** | **31.91** | **46.56** | **9.10** | **22.37** |
| | C-FID | **1.28** | **2.54** | **1.96** | **8.31** | **10.45** | **7.75** | **2.87** | **5.02** |

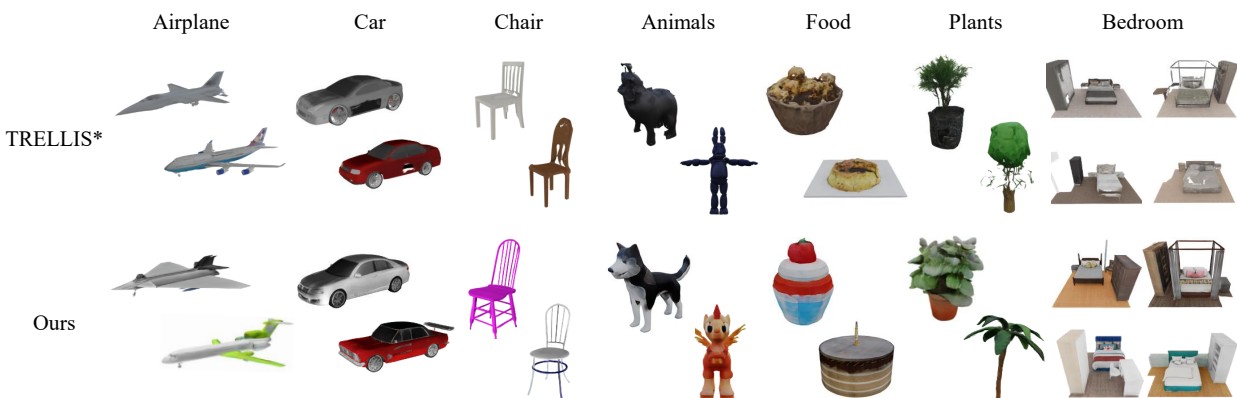

Figure 11: **Qualitative comparison of `Trip2GS` with TRELLIS on generation results.** Despite exhibiting VAE reconstruction quality comparable to our method, TRELLIS relies on a multi-stage generation pipeline that introduces suboptimalities, ultimately degrading generation quality – particularly in unconditional settings under constrained data and computational resources.

batch size from 16 to 64. The results are shown in the Fig. 10 and Tab. 8 below. As can be seen, `Trip2GS` consistently outperforms both TRELLIS variants. Moreover, even after increasing the model scale and batch size of TRELLIS, the generation quality improves only marginally. This suggests that, in the limited-resource regime, efficient model and pipeline design is more critical than simply scaling the model or batch size.

Based on the observations above, we further conduct a comprehensive comparison across all categories using `TRELLIS*-B` (hereafter `TRELLIS*`).

Although TRELLIS achieves VAE reconstruction performance comparable to our `Trip2GS`, its generation quality is substantially inferior, both quantitatively (Tab. 9) and qualitatively (Fig. 11). We attribute this gap to the difficulty of modeling the complex latent space induced by TRELLIS using a flow-matching objective, which becomes more pronounced under limited data and computational resources. In contrast, `Trip2GS` learns a more compact and structured latent space via surface occupancy fields and total variation regularization, while leveraging strong inductive biases through 3D-aware triplane convolutions. Furthermore, TRELLIS adopts a multi-stage generation pipeline, where errors introduced in early stages accumulate and hinder global optimization. In contrast, our unified VAE framework enables progressive refinement: even if coarse geometry is imperfect, subsequent modules (PUM and 3DGS decoder) can correct both geometry and appearance. This limitation is also reflected in the geometric quality. As shown in Tab. 18, TREL-LIS underperforms not only our method but also prior unconditional models such as `AtlasGaussians` and `GaussianCube`, indicating weaker geometric modeling. Such geometric errors propagate to the appearance stage and further degrade generation quality. These results highlight the importance of unified, structure-aware pipelines, particularly in low-data regimes. We provide further analysis on the source of this versatility in Appendix Sec. E.

## C    Conditional Generation

Building on Trip2GS, we showcase a diverse set of conditional generation scenarios. We consider four types of conditions: geometry (point cloud and voxel grid), layout, text, and image. Importantly, the goal of this section is to present conditional generation as an extension that demonstrates the flexibility and compatibility of the same `Trip2GS` framework, rather than to establish a new state of the art on every conditional task. In all conditional settings, we keep the Trip2GS VAE and structured-to-unstructured coarse-to-fine decoder unchanged, and modify only the triplane diffusion stage with condition-specific injection. This design is enabled by the regular, image-like structured latent space of triplanes, which naturally supports heterogeneous conditions within a unified pipeline. Our overall architecture for conditional generation is shown in Fig. 12, where a conditional encoder and condition injection layers are integrated into the triplane diffusion. For conditional generation, we train the diffusion model using the following formulation:

$$\mathcal{L}_{\text{diff}} = \left\| \mu_\theta(\hat{\mathbf{Z}}_t, t, \mathcal{C}) - \hat{\mathbf{Z}}_0 \right\|_2, \tag{7}$$

where $\mathcal{C}$ is an input condition.

### C.1    Condition Preparation

For the point cloud condition, we sample the XYZ coordinates from the input colored point cloud without the RGB values. We sample 10k points for objects and 20k points for scenes, considering each scale.

For the voxel condition, we convert a given point cloud into a binary voxel grid. First, we initialize an empty voxel grid with all values set to zero. Next, each point is mapped to a voxel index within the grid. This is achieved by normalizing the point coordinates to the voxel grid resolution. The points are scaled into the range of the voxel grid and then rounded down to the nearest integer indices. To ensure that only valid points are considered, we filter out indices that fall outside the boundary of the voxel grid. Specifically, we retain only those indices that are within the valid range $[0, R-1]$ along all three dimensions, which we use $R = 64$ for all experiments. After filtering, the valid 3D voxel indices are converted into a single-dimensional index representation. Finally, the corresponding locations in the voxel grid are marked as occupied by setting their values to 1. This step finalizes the discretization process, ensuring that all input points are represented within the voxel grid structure.

The layout condition is obtained by rendering the scene from a top-down orthographic view, where each object category is mapped to a unique label. The resulting layout maps are stored as $32 \times 32 \times 1$ tensors, where the channel dimension encodes the category ID of each pixel.

Text conditions are obtained from the Text2Shape dataset (Chen et al., 2019) for the chair category. For feature embedding, we use a pretrained CLIP text encoder from `openai/clip-vit-large-patch14-336`, which outputs (77, 768) sized feature vectors.

For image conditions, we use images of each object captured from the same viewpoint. Likewise, in the text condition, we use a pretrained CLIP image encoder from `openai/clip-vit-large-patch14-336`, which outputs (577, 1024) sized feature vectors.

### C.2    Network Architecture

For geometry-conditioned generation, which aims to generate textured objects or scenes that adhere to a given coarse geometric structure, such as point clouds or voxel grids, we jointly train the condition encoder with triplane diffusion. We first embed point cloud or voxel grid condition into a triplane representation using condition-specific encoder so that the features from the conditions have the same shape as the latent triplane. The point cloud condition encoder adopts the PointNet (Qi et al., 2017)-based encoder like ConvONet (Peng et al., 2020) similar to the point encoder used in Trip2GS VAE, consisting of three 3D-aware ResNet blocks. The voxel condition encoder processes an occupancy voxel grid of shape $V_{\text{cond}} \in \mathbb{R}^{32 \times 32 \times 32}$ using two 3D convolution blocks, followed by axis-wise mean pooling along each of the three axes to construct the triplane representation. Once the condition is embedded in the triplane space, we inject it into the diffusion model

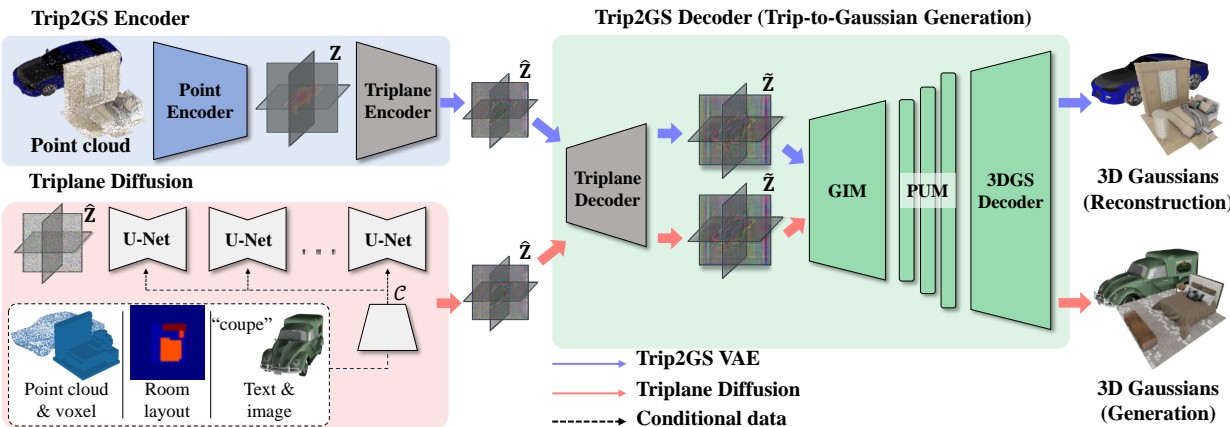

Figure 12: **The overall architecture of Trip2GS for conditional generation.**

using adaptive group normalization (AdaGN) (Dhariwal & Nichol, 2021; Zhang et al., 2024). For layout conditioning, we directly inject the layout map into the $xy$-plane-aligned triplane features. For text and image conditioning, we use pretrained CLIP encoders and integrate their features through cross-attention blocks in the denoiser U-Net. The condition encoder is jointly trained with the triplane diffusion, allowing for end-to-end learning of geometry-conditional 3D generation.

For layout-conditioned generations, we use a semantic segmentation map of top-down orthogonal view. Unlike geometry-conditioned generation, the layout condition is directly used without an additional condition encoder. Since the layout information is aligned with the xy-plane of the triplane latent, we apply AdaGN to inject the condition into the xy-plane of the triplane feature and vanilla group normalization to other planes.

For text and image-conditioned generations, we leverage a pretrained CLIP (Radford et al., 2021) encoder to extract semantic features from textual and visual inputs. During diffusion training, we integrate these features into the triplanes using cross-attention blocks, allowing the model to learn high-level semantic concepts to spatially structured triplane features. To ensure effective conditioning, all conditioning mechanisms are applied at every block of the encoder of the denoiser U-Net, allowing the model to fully capture and propagate conditional information throughout the diffusion process.

### C.3    Generation Results.

To demonstrate the applicability of Trip2GS, we present example of generated samples with diverse conditions. Further conditional generation results can be found in Sec. H

**Geometry Conditioning.** We consider two types of conditional inputs: point cloud and voxel grid. As shown in Fig. 13-(a) and (d), our method generates plausible 3D objects aligned well with given geometry conditions.

**Layout Conditioning.** For layout condition, we generate scenes aligned to a given scene layout. As shown in Fig. 13-(c), Trip2GS effectively generates a scene that closely follows the given layout.

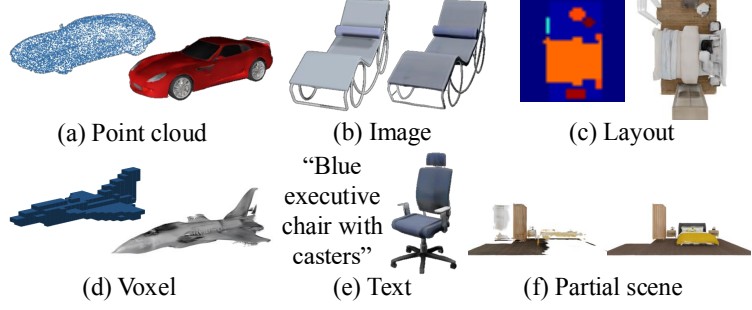

Figure 13: **Examples of various conditional generations.**

**Text/Image Conditioning.** For text/image-to-3D generation, we show that `Trip2GS` generates high-fidelity objects that follow the given text or image (see Fig. 13-(b) and (e)). This demonstrates that our method can learn semantics as well as spatial cues.

**Scene Completion.** We present scene completion as an additional application to validate the versatility of `Trip2GS`. For scene completion, we adopt RePaint (Lugmayr et al., 2022), a diffusion-based inpainting method that leverages a stochastic denoising process to refine missing regions iteratively. Given a partial scene, we utilize our triplane diffusion with RePaint to generate the missing half, ensuring structural and contextual coherence. As shown in Fig. 13-(f), `Trip2GS` effectively completes the missing scene while maintaining consistency with the given partial input. Further details on hyperparameters and additional results for scene completion are provided in Sec. G.3.

## C.4  Comparison on Text-to-3D

In this section, we compare our approach with recent text-to-3D generation methods (Peng et al., 2021; Lan et al., 2024a; Xiang et al., 2025b). We present the quantitative comparison results in Tab. 10, where our method outperforms existing approaches, even in the text-conditioned 3D generation task. As shown in Fig. 22 and Fig. 23, `Trip2GS` captures more detailed geometry while maintaining better alignment with the given text prompts. These results demonstrate the effectiveness of our conditional mechanism in following given semantic cues, in addition to the advantage of our framework in capturing fine-grained details.

Table 10: **Quantitative comparison of text-to-3D generation on ShapeNet chair and car categories using CLIP-Score.**

| Category | Shap-E | LN3Diff | TRELLIS* | Trip2GS (Ours) |
|----------|--------|---------|----------|----------------|
| Chair | 30.31 | 28.87 | 30.42 | **30.84** |
| Car | 26.18 | 26.70 | 26.22 | **27.05** |

## C.5  Comparison on Image-to-3D

We also show qualitative results on image-to-3D generation compared with existing works (Peng et al., 2021; Szymanowicz et al., 2024) in Fig. 24. The results confirm that our method generates 3D outputs that are well-aligned with the given images. It should be noted that compared to recent works curated for image-to-3D generation, `Trip2GS` shows comparable or even better results.

Through further comparisons on conditional generation tasks, we validate that our method is a scalable approach for both text and image conditioned 3D generation as well as unconditional 3D generation. Overall, these results support our claim that the same framework extends beyond unconditional generation while preserving a unified architecture.

Table 11: **Efficiency comparison between our model and baselines.** All experiments are conducted using 8 A6000 48G GPUs. TRELLIS* denotes our reproduced version of the original TRELLIS (Xiang et al., 2025b), used to ensure a fair comparison in the unconditional generation setting (see Sec. B).

| Method | Train (d) | Train Mem (G) | Inference (s) |
|--------|-----------|---------------|---------------|
| GET3D (Gao et al., 2022) | 4.3 | 5.91 | 0.35 |
| DiffGS (Zhou et al., 2025) | 6.7 | 22.34 | 33.24 |
| GaussianCube (Zhang et al., 2024) | 10.3 | 29.15 | 12.13 |
| AtlasGaussians (Yang et al., 2025) | 1.6 | 23.38 | 9.22 |
| TRELLIS* (Xiang et al., 2025b) | 7.9 | 18.19 | 13.72 |
| Ours | 7.4 | 17.98 | 9.48 |

## D  Efficiency Comparison

In this section, we provide a comprehensive comparison of our method against baseline approaches in terms of training time, training memory, and inference time. The quantitative results are summarized in Tab. 11. Regarding training time, AtlasGaussians converges the fastest with 1.6 days of training, while GaussianCube requires the longest time (10.3 days) due to its heavy preprocessing and voxel-based 3D architecture. Our model takes 7.4 days, placing it in the mid-range among baselines while achieving superior generative quality.

For training memory, GET3D shows the lowest memory usage because it employs GAN-based generative framework, which requires less memory than diffusion-based methods. Among the models that represent objects using 3D Gaussian primitives, our model demonstrates the most efficient memory consumption. This indicates that our design, which leverages a triplane latent representation together with the GIM and PUM modules, enables efficient generation of 3D Gaussians without incurring substantial memory overhead.

In terms of inference time, GAN-based GET3D achieves the fastest generation latency. However, they tend to generate less realistic samples compared to diffusion-based methods. DiffGS exhibits the longest generation time, due to their time-consuming octree-based optimization for Gaussian extraction. Among the diffusion-based generative models, our model is one of the fastest, highlighting the benefit of our compact latent representation and efficient Gaussian generation pipeline.

## E  Discussion on Versatility

We have shown that `Trip2GS` can model datasets of substantially different complexity, ranging from simple object-level benchmarks to multi-object indoor scenes. In this section, we further analyze whether the experimental setup used to support our claim of versatility is well justified, and we provide additional evidence for why `Trip2GS` remains effective across such diverse data regimes.

Table 12: **3DGS fitting results across datasets of increasing complexity.** Reconstruction quality consistently degrades from ShapeNet to G-Objaverse and further to 3D-FRONT under the same fixed-capacity fitting setup, supporting that these benchmarks form a natural complexity spectrum.

| Metric | ShapeNet | | | G-Objaverse | | | 3D-FRONT |
|---|---|---|---|---|---|---|---|
| | Airplane | Car | Chair | Animals | Food | Plants | Bedroom |
| PSNR | 38.8435 | 42.1627 | 32.3953 | 31.9954 | 29.3804 | 30.5568 | 25.7111 |
| SSIM | 0.9972 | 0.9970 | 0.9690 | 0.9829 | 0.9629 | 0.9753 | 0.9441 |
| LPIPS | 0.0085 | 0.0067 | 0.0751 | 0.0299 | 0.0590 | 0.0392 | 0.0809 |

### E.1  Benchmarking Datasets

To evaluate the versatility of our method, we use three types of datasets with progressively increasing complexity: ShapeNet, G-Objaverse, and 3D-FRONT. From a dataset-definition perspective, ShapeNet mainly consists of canonical single CAD objects, while Objaverse contains more than 800K internet-scale 3D assets with much greater diversity in geometry and appearance. 3D-FRONT further increases the difficulty by introducing professionally designed indoor scenes composed of multiple high-quality textured furniture objects arranged in non-trivial room layouts. In other words, starting from ShapeNet as the most widely used object-level benchmark, G-Objaverse represents a more diverse and higher-quality object dataset, and 3D-FRONT requires the model to handle not only individual object geometry and texture, but also multi-object composition and spatial arrangement at the scene level.

To make this complexity gap more explicit, we additionally perform a controlled 3DGS fitting experiment. Following the fitting procedure of DiffGS, we uniformly sample points over each object or scene surface (100K points in our experiments), fix the Gaussian positions and opacities, and optimize only the color, scale, and rotation parameters. The fitting results (see Tab. 12) show that reconstruction quality consistently

degrades from ShapeNet to G-Objaverse to 3D-FRONT, indicating that increasing data complexity makes fixed-capacity 3DGS modeling substantially more challenging.

Overall, these results support that ShapeNet, G-Objaverse, and 3D-FRONT form a natural complexity spectrum, where the modeling challenge increases from single-object geometry to diverse object appearance and finally to full-scene spatial composition.

Table 13: **Comparison between two-step neural Gaussian decoding and coarse-to-fine SOF-based decoding.**

| Method | PSNR ↑ | SSIM ↑ | LPIPS ↓ | CD ↓ | F-score ↑ |
|---|---|---|---|---|---|
| Two-step (Neural GS) | 25.45 | 0.8436 | 0.0737 | 16.2233 | 8.6364 |
| Coarse-to-fine (SOF) | **26.00** | **0.8553** | **0.0689** | **14.8772** | **12.4914** |

### E.2 Analysis on Baselines

To model complex scene data effectively, a method must satisfy two requirements: the latent representation should retain sufficient scene information, and the decoder should faithfully reconstruct the geometry and appearance encoded in that latent. From this perspective, `Trip2GS` combines a structured triplane latent, which is well suited for capturing spatial information, with a coarse-to-fine generation scheme that decodes this latent into high-quality 3D Gaussians. In contrast, methods such as GET3D and DiffGS rely on vector-based latents, whose structural inductive bias is relatively weak when the target data contain multiple objects and complex spatial arrangements. A triplane representation, by contrast, benefits from a regular grid structure and CNN priors, making it more stable as the spatial complexity increases from isolated objects to full scenes.

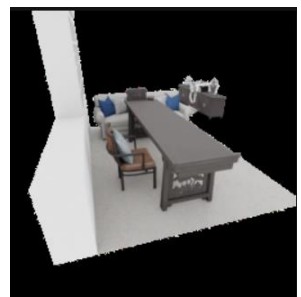 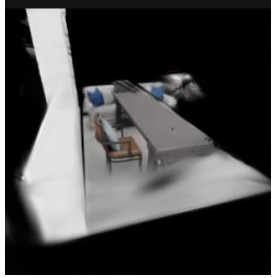

Ground truth              Fitted 3DGS

Figure 14: **Example of densification-constrained 3DGS fitting on a complex 3D-FRONT scene.** Compared with the ground-truth scene, the fitted representation becomes noticeably blurrier and loses local structure, illustrating how fixed-budget Gaussian fitting can degrade the input quality for downstream generative modeling on complex scenes.

However, a strong latent representation alone is not sufficient; the way Gaussian parameters are decoded is equally important. GaussianCube, for example, first fits each training sample with a limited set of Gaussians under a densification-constrained setting and then learns the distribution of these fitted Gaussian parameters. While this strategy is effective for reducing the representation into a fixed-size generative target, it can substantially degrade the input quality when the data are complex, especially for scene-level samples. As illustrated in Fig. 14, such constrained fitting often produces blurrier results and weaker local structure on 3D-FRONT scenes. Moreover, directly modeling Gaussian parameters increases the complexity of the target space, making the overall optimization problem harder.

This observation is also consistent with our own ablations in Sec. 4.3 and Sec. G.1. Viewed from the perspective of Gaussian decoding, our pipeline can be decomposed into three stages: (1) GIM for coarse geometry localization, (2) PUM for fine geometry densification, and (3) the 3DGS decoder for appearance and final Gaussian attributes. When multiple stages are merged into a single module, the decoding problem becomes more entangled and the performance degrades. This effect is particularly pronounced for complex data such as indoor scenes. As shown in Tab. 13, on the 3D-FRONT bedroom category, our coarse-to-fine SOF-based decoding outperforms a two-step neural Gaussian decoder, indicating that explicitly separating coarse support estimation, densification, and appearance decoding becomes increasingly important as the data complexity grows.

AtlasGaussians also attempts to reduce the complexity of Gaussian modeling through a VAE–LDM framework with two-step decoding (*point latent → patch centers → Gaussians*). However, this formulation relies on patch centers and Chamfer loss to capture geometric structure. On complex scene data, this can overemphasize global geometric similarity while underconstraining local structure, which may lead to point clustering or other structural artifacts (Härenstam-Nielsen et al., 2024; Li et al., 2026; Jayasinghe & Brilakis, 2024; Lin et al., 2023). This limitation is well aligned with recent observations that point-set losses based on nearest-neighbor matching can suffer from many-to-one matching and structural failure on complex geometry. To address this issue, we instead separate coarse and fine geometry using the GIM–PUM design and introduce the SOF representation in GIM to better capture local surface support. As shown in Fig. 15, adding SOF supervision to geometry-aware optimization improves the resulting geometry, especially in local regions.

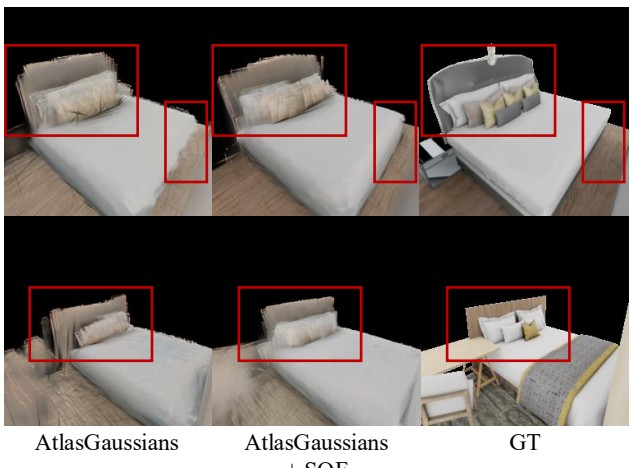

AtlasGaussians     AtlasGaussians     GT
                  + SOF

Figure 15: **Effect of SOF loss on AtlasGaussians.** SOF loss produces sharper edges and a more coherent overall structure.

---

**Algorithm 1** Soft voxelization with trilinear splatting for SOF supervision

---

**Require:** Point set $P = \{\mathbf{p}_n\}_{n=1}^N$, target occupancy $O^*$, resolution $R$
**Ensure:** Predicted occupancy $\hat{O}$, loss $\mathcal{L}$
1: Normalize each point $\mathbf{p}_n$ into voxel coordinates $\tilde{\mathbf{p}}_n$
2: Initialize voxel mass grid $M \leftarrow \mathbf{0}$
3: **for** each $\tilde{\mathbf{p}}_n$ **do**
4:      Find its 8 neighboring voxel corners
5:      Compute trilinear interpolation weights for the 8 corners
6:      Accumulate the weights into $M$ via scatter-add
7: **end for**
8: Convert voxel mass into occupancy: $\hat{O} \leftarrow 1 - \exp(-M)$
9: Compute BCE loss: $\mathcal{L} \leftarrow \mathrm{BCE}(\hat{O}, O^*)$
10: Optionally apply positive-voxel and voxel-wise weighting
11: **return** $\hat{O}, \mathcal{L}$

---

### E.3 What makes Trip2GS versatile?

We have shown that `Trip2GS` performs consistently well across datasets of substantially different complexity, including ShapeNet, G-Objaverse, and 3D-FRONT. Here, *versatility* does not mean that our method is tailored to any specific scene type, but rather that it remains effective as the target data distribution becomes progressively more complex, from canonical single objects to diverse high-quality objects and further to multi-object indoor scenes. In this subsection, we further analyze what contributes to this versatility.

One important factor is the strong **VAE reconstruction quality** of `Trip2GS`. As shown in Tab. 14, our method achieves consistently strong reconstruction performance across datasets. While reconstruction quality alone does not fully determine generation quality, it is still important because it reduces a major decoding bottleneck for downstream latent diffusion. A VAE that more faithfully compresses and reconstructs the input allows the diffusion model to learn on a less lossy latent distribution, which in turn supports higher-quality generation.

Table 14: **VAE reconstruction metric (PSNR) across datasets of different complexity.** Strong reconstruction quality across all datasets supports that `Trip2GS` provides a expressive latent representation for downstream diffusion.

| Method | Airplane | Car | Chair | Animals | Food | Plants | Bedroom | Mean |
|---|---|---|---|---|---|---|---|---|
| DiffGS | 27.1860 | 27.7216 | 23.2123 | 25.1281 | 20.5757 | 22.8552 | 10.4895 | 22.4526 |
| AtlasGaussians | 27.1993 | 28.4700 | 25.1907 | 24.5234 | 24.0561 | 23.1364 | 16.8256 | 24.2002 |
| Trip2GS (ours) | **32.8411** | **30.1260** | **30.2430** | **29.8776** | **27.7997** | **29.1976** | **26.0000** | **29.4407** |

Table 15: **Latent mean and standard deviation comparison across methods and ablations.** A mean closer to 0 and a standard deviation closer to 1 indicate better alignment with the normalized prior typically assumed in latent diffusion. Ours-tet uses DMTet as the intermediate geometry representation, while Ours-ng replaces PUM with neural Gaussians.

| | DiffGS | AG | TRELLIS* | Ours-tet | Ours-ng | Ours $(-TV_{img}, -TV_{lat})$ | Ours $(+TV_{img}, -TV_{lat})$ | Ours $(-TV_{img}, +TV_{lat})$ | Ours $(+TV_{img}, +TV_{lat})$ |
|---|---|---|---|---|---|---|---|---|---|
| Mean | 0.0081 | -0.3338 | 0.7098 | 0.1459 | 0.0101 | **0.0024** | -0.0043 | -0.0032 | 0.0041 |
| Std | 0.5956 | 2.5214 | 0.3305 | 4.1806 | 0.8298 | 0.8951 | 0.9836 | 0.9838 | **0.9911** |

A second factor is that `Trip2GS` learns a more **diffusion-friendly latent space**. In latent diffusion, latents are typically rescaled to match a normalized Gaussian prior, and the latent scale directly affects the signal-to-noise ratio and training conditioning. Therefore, it can serve as a useful indicator that the latent distribution is better aligned with the normalized prior assumed in diffusion training, in the sense that its mean is closer to 0 and its standard deviation is closer to 1. As shown in Tab. 15, the latent space of `Trip2GS` is more favorably scaled than those of prior methods and several alternative variants.

This analysis also reveals two interesting observations. First, replacing SOF with DMTet leads to a noticeably less favorable latent scale. This is consistent with the interpretation that SOF simplifies geometry learning by turning it into a more direct coarse surface support prediction problem, instead of forcing the latent to simultaneously model both coarse and fine geometry in a single stage. Second, TV regularization makes the latent standard deviation to 1 while slightly reducing reconstruction fidelity, as shown in Tab. 16. Notably, this same regularization improves the final generation quality, as discussed in Tab. 6. This suggests that good generation cannot be explained by reconstruction quality alone, and that latent regularity is also important.

Table 16: **Ablation on TV regularization.** TV regularization slightly reduces reconstruction quality, but improves latent regularity and downstream generation quality.

| TV-image | TV-latent | PSNR ↑ | SSIM ↑ | LPIPS ↓ |
|---|---|---|---|---|
| | | **31.0932** | **0.9587** | **0.0332** |
| ✓ | | 30.4395 | 0.9558 | 0.0359 |
| | ✓ | 30.7881 | 0.9561 | 0.0346 |
| ✓ | ✓ | 30.2430 | 0.9536 | 0.0376 |

Overall, these results suggest that the versatility of `Trip2GS` comes not from scene-specific tricks, but from the combination of two complementary properties: (1) strong reconstruction fidelity, which alleviates decoding bottlenecks, and (2) a coarse-to-fine design with SOF, which helps organize the latent space into a form that is easier for latent diffusion to model. In this sense, our coarse-to-fine scheme improves not only the final 3DGS decoding quality, but also the structure of the latent distribution itself, which helps `Trip2GS` remain robust across datasets of different complexity.

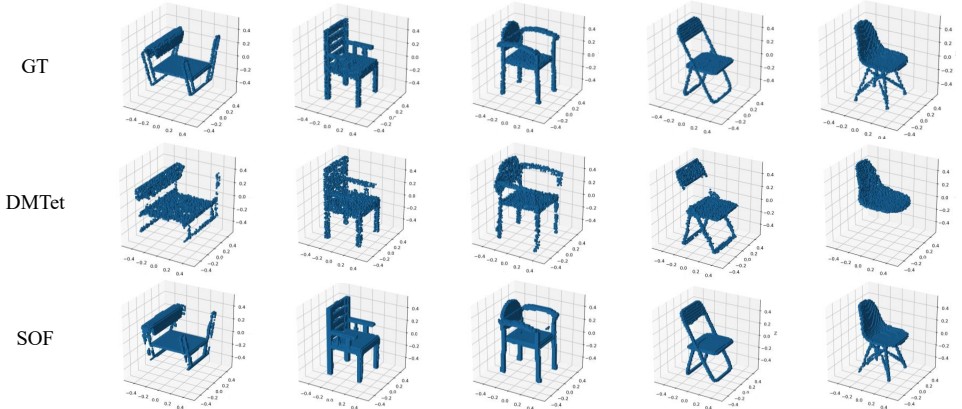

Figure 16: **Intermediate geometry comparison between DMTet and SOF on the Chair category.** We measure mIoU at resolution 64. For fair comparison, 4,000 points are sampled from the DMTet mesh to match the average number of occupied voxels predicted by SOF. SOF better preserves thin and disconnected chair structures, resulting in higher intermediate geometric fidelity.

## F Geometric Quality Comparison

We have shown that the quality of the intermediate geometry representation plays an important role in generating high-quality unstructured 3D Gaussians. In this section, we provide additional experiments to further support this claim from a geometric perspective.

### F.1 mIoU

What matters for the intermediate geometry representation is not necessarily exact mesh-level reconstruction, but rather the *geometric fidelity of the intermediate support* used to generate 3D Gaussians. As discussed, this intermediate fidelity is particularly important for preserving thin and detailed structures in the final output.

To further validate this claim, we additionally measure the intermediate-level mIoU on the Chair category when using DMTet and SOF. We use resolution 64 in both cases. For a fair comparison, we sample 4,000 points from the DMTet mesh, matching the average number of occupied voxels predicted by SOF, and use these points for evaluation. As shown in Tab. 17 and Fig. 16, SOF captures thin chair structures more faithfully than DMTet at the intermediate geometry level.

Table 17: **Intermediate geometry comparison on the Chair category.** SOF achieves substantially higher mIoU than DMTet at resolution 64.

|  | DMTet | SOF |
| --- | --- | --- |
| mIoU | 0.1719 | **0.7323** |

### F.2 MMD/COV

We further evaluate the geometric similarity between the final generated 3D Gaussians and ground-truth surface point clouds. Following the reviewer's suggestion, we adopt geometry-aware metrics based on point-set similarity. Since GET3D and DiffGS show substantially weaker generation quality overall, we compare against the strongest Gaussian-based baselines, GaussianCube, AtlasGaussians, and TRELLIS. As shown in Tab. 18, `Trip2GS` achieves the best overall performance, with the lowest mean MMD and the highest mean COV. These results further support that our method produces geometrically more faithful and more diverse generations.

Table 18: **MMD/COV (%) comparison between Gaussian-based generation methods.** Lower MMD and higher COV indicate better geometric quality.

| Model | Metric | Airplane | Car | Chair | Animals | Food | Plants | Bedroom | Mean |
|---|---|---|---|---|---|---|---|---|---|
| GaussianCube | MMD | 0.0358 | 0.0428 | 0.0705 | 0.0821 | 0.0688 | 0.0743 | 0.2903 | 0.0949 |
| | COV (%) | 54.88 | 27.02 | 37.05 | 33.42 | 52.69 | **51.61** | 57.57 | 44.89 |
| AtlasGaussians | MMD | 0.0288 | 0.0347 | 0.0606 | **0.0732** | 0.0653 | **0.0712** | 0.2671 | 0.0858 |
| | COV (%) | **70.09** | **45.30** | 59.34 | **40.26** | 56.57 | 51.36 | 54.19 | 53.87 |
| TRELLIS* | MMD | 0.0421 | 0.0682 | 0.0850 | 0.0873 | 0.0910 | 0.1053 | 0.2968 | 0.1108 |
| | COV (%) | 43.09 | 23.74 | 33.21 | 28.95 | 41.49 | 31.74 | 50.00 | 36.03 |
| Trip2GS (ours) | MMD | **0.0245** | **0.0329** | **0.0573** | 0.0746 | **0.0602** | 0.0727 | **0.1864** | **0.0727** |
| | COV (%) | 63.63 | 45.16 | **65.61** | 34.10 | **63.73** | 47.27 | **64.24** | **54.82** |

# G Additional Ablation Studies

Table 19: **Reconstruction quality on the test set of ShapeNet airplane category with different upsampling ratios using neural Gaussians.** Here, $k$ means upsampling ratio.

| $k$ | PSNR | SSIM | LPIPS |
|---|---|---|---|
| 1 | 27.13 | **0.9586** | 0.0471 |
| 4 | **27.66** | **0.9586** | **0.0451** |
| 16 | 27.24 | 0.9573 | 0.0471 |

## G.1 Comparison with Neural Gaussians

One possible approach to increase the number of Gaussians from a limited set of anchor points is generating neural Gaussians (Lu et al., 2024; Shen et al., 2024). These methods decode point features $\mathbf{f} \in \mathbb{R}^{N \times C}$ into 3D Gaussians $\mathbf{g} \in \mathbb{R}^{kN \times 14}$, where each point generates $k$ Gaussians, efficiently augmenting the Gaussians with minimal computational overhead. However, we observed that it introduces undesired grid pattern in the output rendering (refer to Fig. 17-(b)). The issue arises because the anchor points extracted from GIM are inherently discrete and discontinuous. During querying the triplane feature, the anchor points are fixed at predetermined locations, thus causing discrepancies between adjacent features. Furthermore, generating multiple Gaussians from a limited feature set imposes constraints to decode high-quality Gaussians, limiting the effectiveness of upsampling. As shown in Tab. 19, increasing the upsampling ratio $k$ of neural Gaussians results in either similar or worse reconstruction quality.

In contrast, our PUM consistently improves all metrics, validating the effectiveness of our approach for upsampling 3D Gaussians (see Tab. 20). With our lightweight MLP-based PUM network, we can efficiently generate sufficient number of Gaussians without excessive computational burden.

## G.2 Effect of Point Mapping

In this work, we proposed PUM, which upsamples the coarse anchor point from GIM while mapping discrete points into continuous space. Here, we investigate the effect of the mapping function of PUM. Specifically, we compare the visual quality of our `Trip2GS` with and without PUM, while setting the upsampling ratio 1, which means PUM does not perform upsampling. By using coarse anchor points from GIM directly as query points for 3DGS decoder, we observed unusual grid pattern or texture abruption in the rendered results, as shown in Fig. 17-(b). In contrast, by leveraging the mapping function of PUM, grid pattern is mitigated, producing visually smooth and coherent outputs (refer to Fig. 17-(c)). These findings indicate that the

Table 20: **Reconstruction quantitative results on test set of ShapeNet airplane category with different upsampling ratios.** Here, *k* denotes the upsampling ratio, **M** the training memory, and **T** the inference speed.

| $k$ | PSNR | SSIM | LPIPS | M (GB) | T (s) |
|---|---|---|---|---|---|
| 1 | 26.72 | 0.9558 | 0.0513 | 15.35 | 0.099 |
| 4 | 27.56 | 0.9641 | 0.0435 | 18.17 | 0.101 |
| 16 | **27.94** | **0.9702** | **0.0396** | 24.01 | 0.107 |

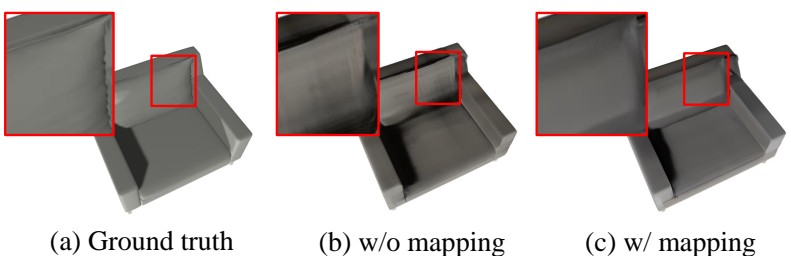

(a) Ground truth      (b) w/o mapping      (c) w/ mapping

Figure 17: **Visualization of the generated results with respect to the mapping function of our PUM.** Mapping function alleviates tile artifacts caused by the discrete nature of coarse anchor points from GIM.

mapping function enhances the visual quality of the generated 3D Gaussians. With the mapping function of our PUM, reconstruction quality is consistently improved as the upsampling ratio increases (see Tab. 20).

### G.3 Hyperparameters for Scene Completion

Table 21: **Comparison of FID, KID, and C-FID scores for different hyperparameters for repaint of scene completion.**

| | $j = 10$ | | | $j = 20$ | | | $j = 30$ | | | $j = 40$ | | | $j = 50$ | | |
|---|---|---|---|---|---|---|---|---|---|---|---|---|---|---|---|
| | FID | KID | C-FID | FID | KID | C-FID | FID | KID | C-FID | FID | KID | C-FID | FID | KID | C-FID |
| $r = 5$ | 17.79 | 5.31 | 1.78 | 17.24 | 5.04 | 1.73 | 17.13 | 5.02 | 1.73 | 17.08 | 4.96 | 1.73 | 17.18 | 5.12 | 1.77 |
| $r = 10$ | 17.63 | 5.13 | 1.78 | 17.14 | 4.92 | 1.72 | 17.16 | 4.99 | **1.71** | **16.98** | **4.91** | 1.73 | 17.01 | 4.98 | 1.75 |
| $r = 15$ | 17.59 | 5.07 | 1.74 | 17.21 | 4.98 | **1.71** | 17.05 | 5.00 | 1.72 | 17.05 | 5.01 | 1.73 | 16.99 | 4.97 | 1.72 |

Scene completion application is achieved by using our trained triplane diffusion model, where half-masked triplane is provided as input and used as a condition. Our completion is based on RePaint (Lugmayr et al., 2022). RePaint proposes an iterative resampling strategy that repeats forward and reverse processes and addresses the disharmony problem that arises due to a mismatch between sampled masked region and denoised result. Selecting an appropriate *jump* ($j$), which denotes the length of diffusion step for back-and-forth iterations, and *resampling* ($r$), which indicates the number of iterative resample steps, is crucial for scene completion quality. Experiment results are shown in Tab. 21.

# H    Additional Results

## H.1    Results on ShapeNet/3D-FRONT Datasets

In this section, we present additional generation results, including unconditional generation in Fig. 26, as well as conditional generation with different inputs: geometry (point cloud and voxel) in Fig. 27, room-scale layout in Fig. 28, text in Fig. 29, and image in Fig. 30 on the ShapeNet and 3D-FRONT datasets. In Fig. 31, we present scene completion results using partial scenes.

Additionally, to further examine the versatility of `Trip2GS` on scene-level data, we conduct extra experiments on the *library* category of 3D-FRONT. Compared with the *bedroom* category, the library split contains substantially fewer samples (435 vs. 4,573), making distribution learning much more challenging for all methods. As shown in Tab. 22 and Fig. 18, all models struggle more in this small-data regime. Nevertheless, `Trip2GS` still produces more detailed scene structures, which further supports the robustness of our framework beyond a single scene category.

Table 22: **Quantitative comparison on the 3D-FRONT library category.**

| Method | FID ↓ | KID ↓ | C-FID ↓ |
|---|---|---|---|
| GaussianCube | 177.12 | 158.86 | 31.74 |
| AtlasGaussians | 124.20 | 106.23 | 28.10 |
| Trip2GS (ours) | **31.72** | **25.46** | **8.49** |

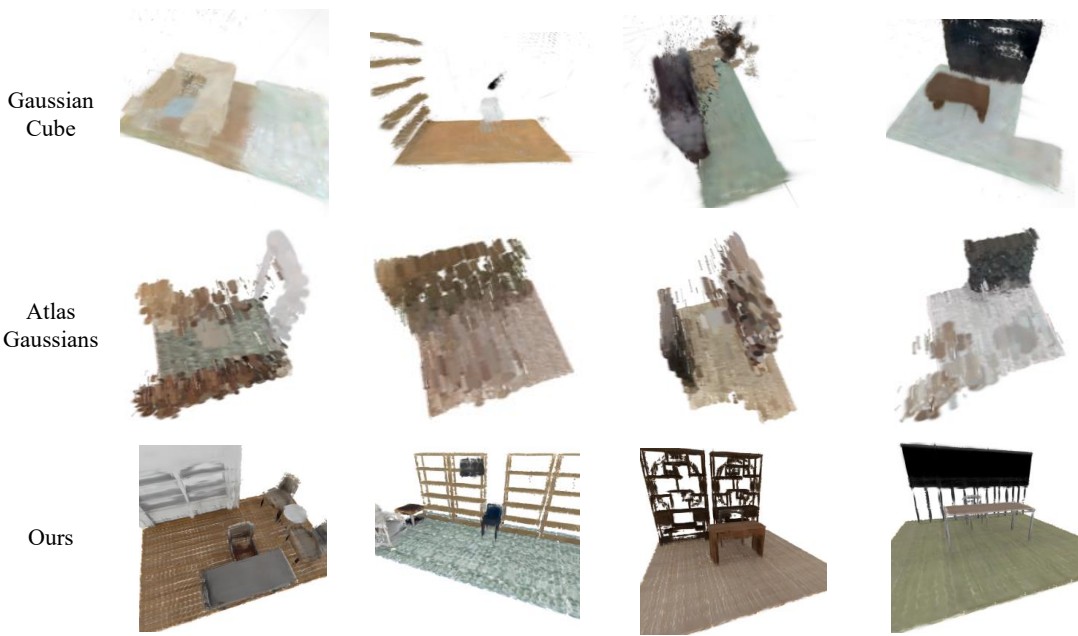

Figure 18:    **Qualitative comparison on the 3D-FRONT library category.** Compared with prior methods, our approach better preserves thin structures and generates more detailed scene geometry.

## H.2    Results on Objaverse Dataset

To further validate the generalizability of our method to the high-quality 3D assets, we showcase additional results on the recent G-buffer Objaverse (Zuo et al., 2024; Qiu et al., 2024; Deitke et al., 2022) dataset.

G-buffer Objaverse dataset contains a subset of Objaverse (Deitke et al., 2022) dataset, including 10 general categories (*e.g.*, Animals, Furnitures, Food, etc.). Among them, we show unconditional generation results on three categories: Plants, Food and Animals, which contain samples with diverse geometries and textures.

Fig. 25 presents a qualitative comparison of unconditional generation results on the G-buffer Objaverse dataset. Overall, our method produces the most realistic and coherent samples across both geometry and texture compared to baseline models. From a geometry perspective, GET3D generates incomplete structures, and DiffGS sometimes produces empty holes or places Gaussians outside the object boundaries. Gaussian-Cube produces torn surfaces which fails to reconstruct complete surfaces, while AtlasGaussians captures geometry relatively well but shows minor misalignment at fine details. In contrast, our method explicitly captures object surfaces and predicts Gaussian positions accordingly, resulting in accurate, well-aligned geometry. Regarding texture and appearance, baseline models often generate oversmoothed textures that lack high-frequency details. However, our approach synthesizes realistic, diverse, and detailed textures, capturing both global structure and fine details. Further qualitative results are presented in Fig. 32, which demonstrate strong performance on a dataset with diverse geometries and textures.

### H.3    Application: Real-World Scenario

In this work, we primarily evaluate on synthetic 3D datasets in order to measure the representational power and learning capability of the generative model as directly as possible. This evaluation protocol is also consistent with many prior unconditional 3D generation works. In real-world multi-view datasets, factors such as capture noise, pose errors, background clutter, partial occlusion, segmentation errors, and scene-dependent reconstruction quality are all entangled with the model's own generative ability. By contrast, synthetic benchmarks allow these external factors to be more tightly controlled, making them more suitable for analyzing representation design and generative pipeline capacity in a cleaner setting.

Nevertheless, extending unconditional 3D generation to real-world data is an important and highly interesting direction. In particular, reconstructing and generating 3D assets from casual real-world multi-view captures, rather than from clean scanned 3D assets, is a much more challenging yet practically meaningful problem. In this section, we explore the potential of `Trip2GS` in such a real-world scenario.

To this end, we conduct an additional qualitative experiment on the MVImgNet dataset (Yu et al., 2023). MVImgNet is a large-scale real-world multi-view dataset built from videos of everyday objects captured in the wild. Unlike synthetic benchmarks, it does not provide clean watertight 3D geometry or dense depth supervision, but instead offers posed multi-view RGB images together with object masks and other auxiliary annotations. For our experiment, since dense depth is not directly available, we reconstruct the input colored point clouds required by `Trip2GS` using structure-from-motion (COLMAP) (Schonberger & Frahm, 2016) from the input images. Following the protocol of DD-IBR, we train the VAE and class-conditioned diffusion model on three categories: chair, table, and sofa. For object reconstruction, we use the provided foreground masks to filter out background pixels from the RGB images. This setup allows us to examine whether `Trip2GS` can still learn a meaningful 3D prior from real-world multi-view supervision alone.

However, as illustrated in Fig. 19, supervision from MVImgNet is substantially noisier than that from synthetic datasets. Real-world captures often contain inconsistent lighting, reflective or non-Lambertian materials, mask inaccuracies, imperfect camera poses, sparse or noisy reconstructed geometry, and residual background regions or truncation in some views. As highlighted by the red boxes in the figure, these issues can produce inconsistent guidance across views, making both reconstruction and distribution learning significantly more difficult.

Despite this difficulty, the design of `Trip2GS` remains beneficial in this setting. In particular, the SOF-based geometric supervision provides an additional 3D constraint through $L_{\text{geo}}$, which helps compensate for inconsistencies in the multi-view image supervision. As highlighted by the blue boxes, even when the input views provide imperfect or partially inconsistent evidence, `Trip2GS` is still able to recover a plausible and geometrically complete 3D structure. This suggests that the coarse-to-fine design of `Trip2GS` can be advantageous even in noisy real-world capture scenarios.

The generation results in Fig. 19 further show that `Trip2GS` can produce plausible 3D assets from such noisy real-world inputs. Although this experiment is qualitative and should be viewed as an exploratory result rather than a definitive benchmark, it indicates that the coarse-to-fine 3D prior learned by `Trip2GS` on synthetic data can remain useful in real-world settings. We believe this direction is a promising avenue for future work toward more practical and realistic 3D generation.

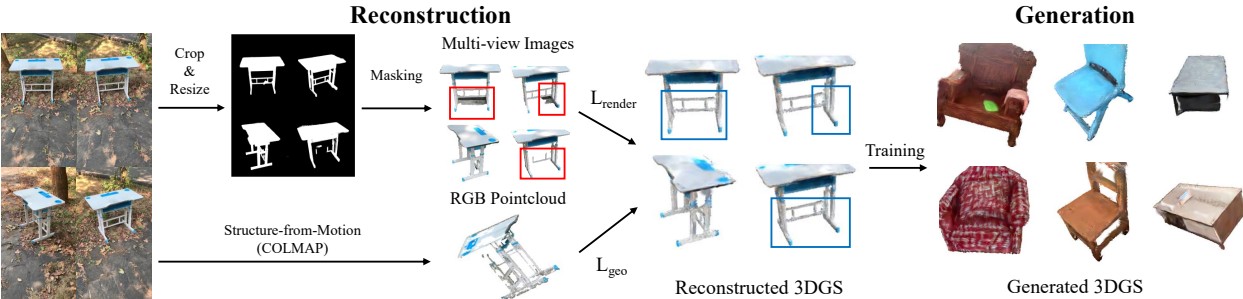

Figure 19: **Real-world application on MVImgNet from multi-view captures.** We apply `Trip2GS` to real-world object videos by using foreground masks and sparse RGB point clouds constructed by COLMAP. The red boxes highlight inconsistent image evidence across views, while the blue boxes show that `Trip2GS` can still recover plausible and geometrically complete 3D structures under such noise. The generated samples on the right further suggest the potential of `Trip2GS` in real-world capture scenarios.

# I  Further Discussions, Analysis and Future Work

## I.1  Discussion on Unconditional Generation.

**Theoretical Importance.** Unconditional generation is important for two main reasons. First, it isolates the core ability of a generative model to learn the underlying data distribution from examples, which is one of the most fundamental goals of generative modeling. In contrast to recent large-scale approaches that rely heavily on scaling model parameters, training data, and compute, our work focuses on a complementary regime: designing efficient architectures and pipelines that remain effective under limited data and computational budgets. We believe this setting is particularly important because it better reflects many realistic scenarios, including academic research environments, individual users, and product deployment settings, where efficiency, accessibility, and practical value are critical. Second, unconditional generation also serves as a foundation for conditional generation. Effective modeling of the data distribution in the unconditional setting provides a strong basis for subsequently incorporating text, image, or other conditions, and for scaling models and datasets more effectively in future work.

**Practical Importance.** Beyond its theoretical role, unconditional generation is also useful in several practical applications. One example is content creation and asset bootstrapping, where the goal is often not to produce a single final asset from the start, but to rapidly generate many plausible 3D drafts or variations for artists and designers to select from; in this setting, unconditional generation is useful for populating an initial asset library across categories such as chairs, airplanes, indoor objects, or custom classes. A second example is synthetic data generation for simulation, digital twins, and robotics, where the need is often for diverse environments, object arrangements, and scene variations rather than one specific output; unconditional generation can therefore act as a scalable source of diverse 3D objects and scenes sampled directly from the learned distribution. A third example is early-stage design exploration, where users may not yet have precise conditions in mind and instead want to explore a broad space of possible structures, layouts, or assemblies. In such cases, unconditional generation can be especially valuable because it naturally proposes diverse candidates from the data distribution before more specific constraints are introduced.

### I.2 Generalization and Memorization

One of the common issues in 3D generation is memorization issue and generalization to unseen data. In this section, we report the generation metrics on test set and additional analysis on learned latent feature space.

**Evaluations on Test Set.** First, we describe the data splits used in our experiments. For all three datasets used in the paper, we divide each category into train/validation/test splits with a ratio of 7:1:2 and train all models accordingly. In the main paper, we report quantitative results against the training split, whereas here we additionally provide the corresponding results on the test split. For evaluating generalization, it is desirable that the gap between train-set and test-set metrics remain small. However, a small train–test gap alone is not sufficient, since a model with uniformly poor quality may also exhibit only a small difference. Therefore, both the absolute fidelity on the training split and the train–test gap are important for a meaningful assessment of generalization. The dataset statistics and quantitative results are summarized below and in Tab. 27.

**ShapeNet** uses the official 7:1:2 split: Airplane (2832 / 404 / 809), Car (5248 / 749 / 1499), and Chair (4742 / 677 / 1354).

**3D-FRONT** uses a random 7:1:2 split: Bedroom (4573 / 653 / 1308), Livingroom (2048 / 292 / 586), and Library (435 / 62 / 125).

**G-Objaverse** also uses a random 7:1:2 split: Animals (5748 / 821 / 1642), Food (2345 / 335 / 670), and Plants (2821 / 403 / 806).

**Feature Space Interpolation.** As an additional result demonstrating the generalization capability of our model, we present feature-space interpolation results between objects from the ShapeNet and G-Objaverse categories in Fig. 33. As shown in the figure, the transition between two different objects is smooth and semantically natural. This suggests that our model learns a smooth latent space and captures the underlying data distribution, rather than merely memorizing individual training instances.

### I.3 Discussion on Components of Trip2GS

**Discussion on Triplane.** A potential concern of triplane representations is that, because 3D geometry is encoded on three 2D planes, non-convex objects or cluttered scenes may induce ambiguity due to occlusion across different depths. We mitigate this issue from three perspectives. First, the input point cloud is projected onto three orthogonal planes ($xy/xz/yz$), and the point features retain positional information, so even when multiple structures overlap on a plane, depth-related cues are not entirely discarded. Second, simply concatenating the three planes and applying standard 2D convolutions may blur or lose 3D spatial information. To preserve spatial structure, we adopt 3D-aware convolutions following prior triplane-based works such as Rodin and BlockFusion. Third, GIM introduces explicit 3D supervision: instead of relying only on rendering loss, it learns whether each queried 3D location should contain geometry, which helps recover spatial information that may otherwise be weakened during projection and planar processing. Together, these design choices allow the triplane latent to remain effective even for non-convex objects and complex indoor scenes.

**Discussion on GIM.** In our framework, GIM is a key component for directly generating 3D Gaussians from triplane features. Its primary role is to extract surface information and produce coarse anchor points, which serve as a reliable geometric prior for downstream Gaussian decoding. To study the effect of GIM resolution, we evaluate three resolutions (48, 64, and 96) on two categories with relatively thin or complex structures: Chair and Bedroom. As shown in Tab. 23 and Tab. 24, increasing the resolution from 48 to 64 yields a clear improvement in reconstruction quality, while increasing it further to 96 gives only marginal gains or even slight degradation despite a large increase in the number of Gaussians. We interpret this as a trade-off between anchor precision and downstream decoding burden. A lower resolution tends to produce fewer anchors and fewer Gaussians, which may limit the ability to represent detailed textures and structures. A higher resolution produces many more anchors and Gaussians, but this additional density does not necessarily improve the quality of the geometric prior, and instead increases redundancy and decoding

complexity. In both categories, resolution 64 provides the best balance between geometric support and final rendering quality.

This interpretation is consistent with recent 3DGS literature. Structured or anchor-based methods such as Scaffold-GS (Lu et al., 2024) improve robustness by organizing Gaussians around sparse anchors, while compactness-oriented methods such as MesonGS (Xie et al., 2024) show that many learned Gaussians are redundant and can be removed with limited quality loss. These observations support our view that simply increasing the number of anchors or Gaussians is not sufficient; what matters more is the quality of the geometric prior and how effectively it can be refined downstream. This is also why GIM is paired with PUM in our design: once a reasonable coarse support is obtained, PUM provides a more parameter-efficient way to recover fine detail than further increasing the GIM resolution alone.

Table 23: **Effect of GIM resolution on the chair category.**

| Resolution | PSNR | SSIM | LPIPS | # Gaussians | mIoU |
|---|---|---|---|---|---|
| 48 | 29.85 | 0.9492 | 0.0419 | 5,156 | **0.8100** |
| 64 | **30.24** | **0.9536** | 0.0376 | 10,814 | 0.7209 |
| 96 | 30.22 | 0.9527 | **0.0367** | **23,197** | 0.5948 |

Table 24: **Effect of GIM resolution on the bedroom category.**

| Resolution | PSNR | SSIM | LPIPS | # of Gaussians | mIoU |
|---|---|---|---|---|---|
| 48 | 25.18 | 0.8475 | 0.1012 | 22608 | **0.7324** |
| 64 | **26.00** | **0.8553** | **0.0689** | 33056 | 0.6519 |
| 96 | 25.47 | 0.8485 | 0.0755 | **82216** | 0.5020 |

## I.4 Limitations and Future Works

**Failure Cases.** Since `Trip2GS` generates the entire scene in a feed-forward manner, rather than relying on scene-specific design choices or explicit layout-based generation, its performance can degrade when the scene becomes substantially larger or contains many small objects. In such cases, the model must allocate its representational capacity across a much broader spatial extent, which can reduce the fidelity of individual object details.

This issue is particularly visible for large-scale 3D-FRONT living-room scenes, whose spatial scale is considerably larger than that of bedrooms (approximately 6.0 vs. 3.0 in our normalized setting). Because objects become relatively smaller with respect to the whole scene, it becomes more difficult to preserve the geometry and texture of small furniture items when modeling the full scene at once. As shown in Fig. 20, this can lead to oversmoothed or incomplete local structures in complex large-scale scenes. We believe that handling such large scenes more faithfully may require substantially more training data, scene-specific architectural design, or alternative strategies such as optimization-based refinement or layout-guided generation.

**Model and Data Scalability.** The main focus of this work is improving generation quality under limited data and computational budgets, which we believe is an important setting for research laboratories, individual users, and practical deployment. Nevertheless, model and data scalability are also important issues, and here we provide several additional observations in that direction.

Figure 20: **Failure case on large-scale living-room scenes.** When the scene scale becomes much larger and contains many relatively small objects, `Trip2GS` may fail to preserve fine local geometry and texture for individual items.

First, from the perspective of **model scalability**, we evaluate performance on the 3D-FRONT living-room category, which we identified earlier as a challenging large-scale scene setting. Our original VAE uses `model_channels=256` (227.47M parameters), and we compare it with a larger variant using `model_channels=384` (495.60M parameters). As shown in Tab. 25 and Fig. 21, increasing the model scale improves both quantitative and qualitative results on the living-room category. These results suggest that model capacity is indeed one important factor for handling larger and more complex scene distributions.

Table 25: **Quantitative comparison on the 3D-FRONT living-room category with different model scales.** Increasing the model size improves both reconstruction quality and geometric fidelity on this more challenging large-scale scene setting.

| Model | PSNR ↑ | SSIM ↑ | LPIPS ↓ | CD ($\times 10^3$) ↓ | F-Score ($\times 10^3$) ↑ |
|---|---|---|---|---|---|
| Original | 22.49 | 0.7854 | 0.1535 | 34.10 | 2.99 |
| Larger | **22.68** | **0.7884** | **0.1412** | **27.94** | **3.11** |

Table 26: **Effect of training batch size on reconstruction quality.** Larger batch sizes consistently improve the quality metrics, suggesting that additional computational resources can further benefit training.

| Metric | Batch 4 | Batch 8 | Batch 16 |
|---|---|---|---|
| PSNR ↑ | 28.91 | 30.24 | **31.06** |
| SSIM ↑ | 0.9443 | 0.9536 | **0.9598** |
| LPIPS ↓ | 0.0513 | 0.0376 | **0.0314** |

Second, from the perspective of **data scalability**, although we do not conduct experiments on very large-scale datasets in this paper, recent works such as Direct3D (Wu et al., 2024b) and SAR3D (Chen et al., 2025a) suggest that triplane-based representations can scale to larger datasets while maintaining high-quality 3D generation. More importantly, the main contribution of our work lies in the coarse-to-fine decoding process from a *structured latent* to an *unstructured output*. This idea is not restricted to triplanes: it can also be applied to other structured representations such as voxel-based features (e.g., TRELLIS-style latents), and more broadly to different output formats including 3D Gaussians, point clouds, or meshes. In this sense, we view our contribution as a generally applicable generation scheme rather than one tied to a single representation.

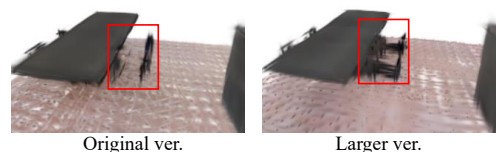

Original ver.          Larger ver.

Figure 21: **Effect of model scaling on large-scale living-room scenes.** Increasing the model capacity improves both geometric fidelity and visual quality on complex large-scale scenes, suggesting that model scale is one important factor for handling more challenging scene distributions.

Finally, scaling to larger datasets in practice also requires greater computational resources, including larger GPU memory, faster accelerators, and longer training schedules. With more computation, one can further increase the batch size, and our additional experiment in Tab. 26 suggests that this can indeed improve performance.

**Large-scale Scene Generation.** We have demonstrated that our method successfully scales from object-level to scene-level generation. However, generating scenes containing numerous objects or producing large-scale environments remains a challenging problem, and addressing such scenarios typically requires compositional generation strategies or autoregressive generation. Prior works (Wu et al., 2024c; Meng et al., 2025; Lee et al., 2024) have highlighted the potential of image inpainting techniques as a promising solution for scalable scene construction. Given that our model exhibits strong modeling capacity across both object and scene-level generation, and achieves high-quality scene completion (see Fig. 31) using the RePaint (Lugmayr et al., 2022) algorithm, we believe it provides a solid foundation for tackling these challenges. These capabilities indicate that our framework can be naturally extended to large-scale environments. Advancing in this direction would enable more flexible and scalable scene synthesis pipelines, broaden potential real-world applications, and further advance research in 3D scene generation.

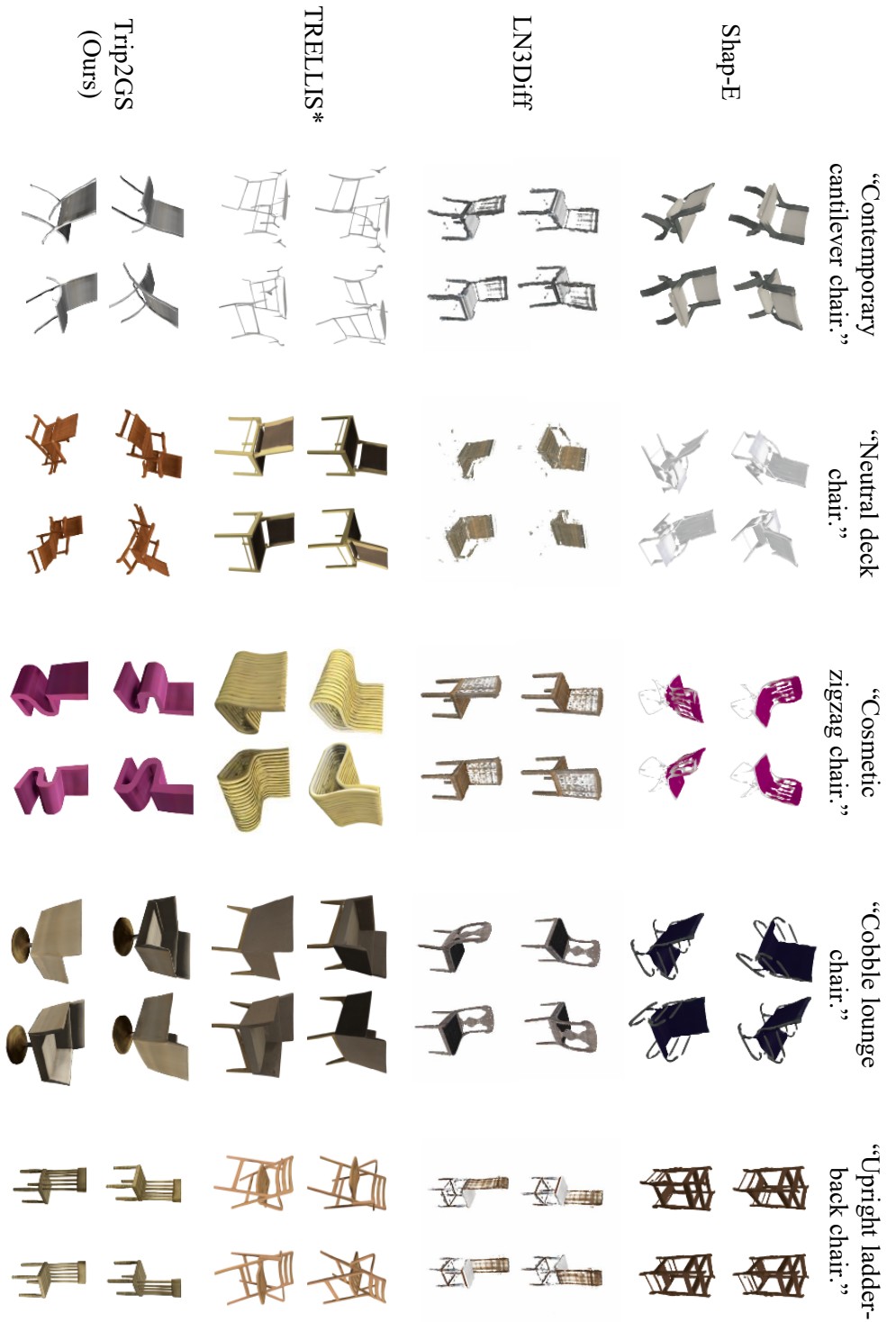

Figure 22: **Qualitative comparisons on text-to-3D generation on the ShapeNet chair category.**

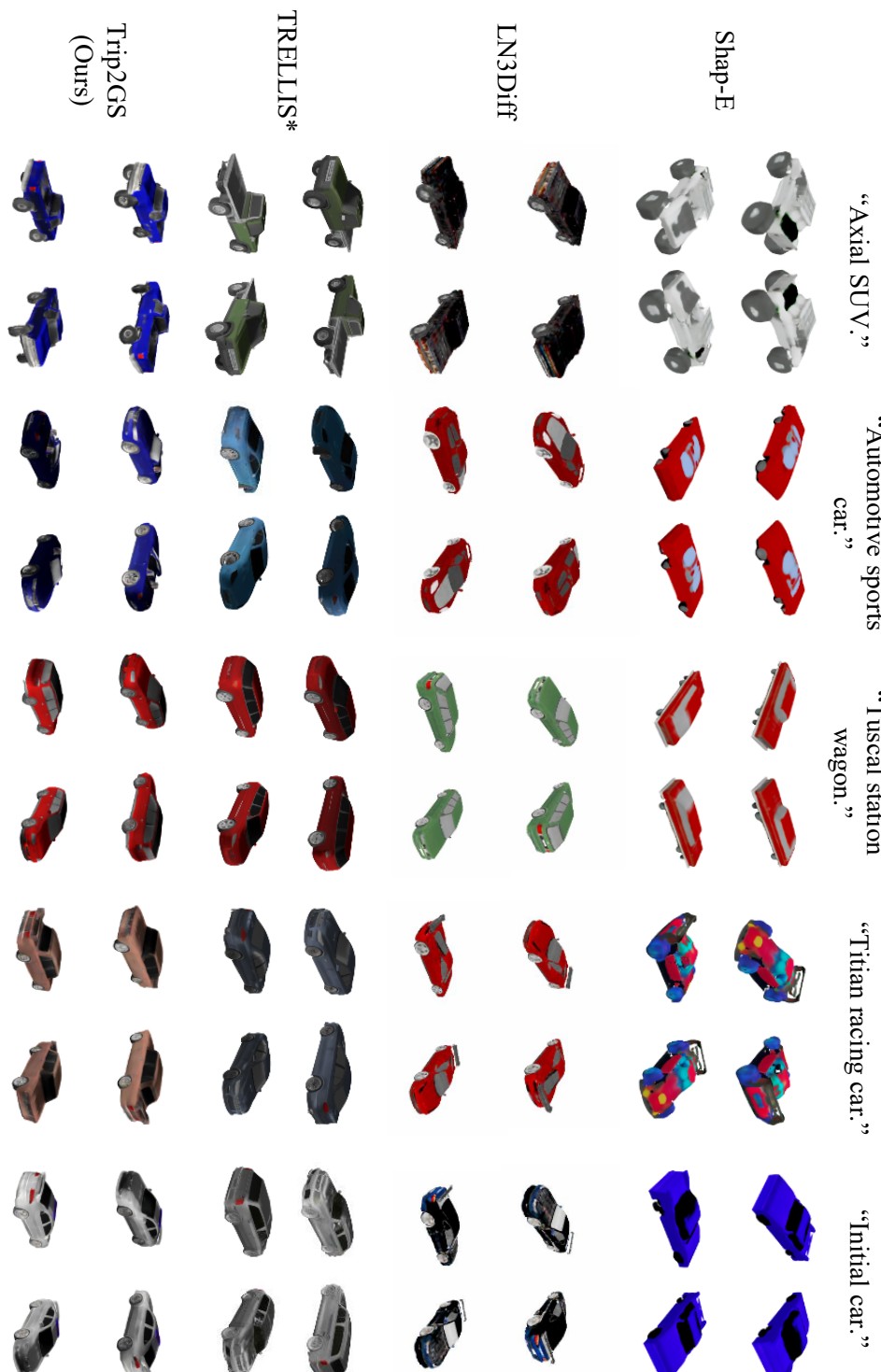

Figure 23: **Qualitative comparisons on text-to-3D generation on the ShapeNet car category.**

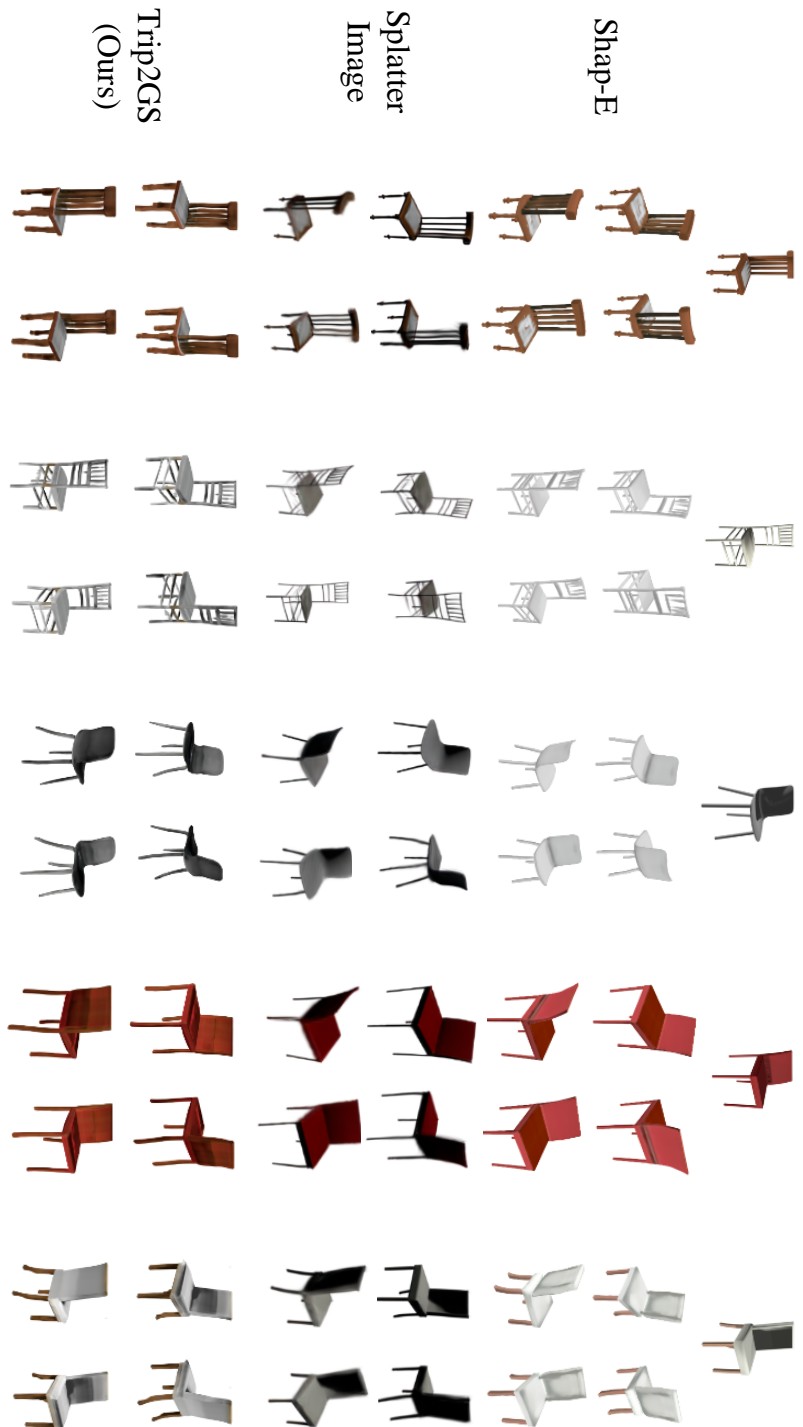

Figure 24: **Qualitative comparisons on image-to-3D generation on the ShapeNet chair category.**

Table 27: **Train/test quantitative comparison across categories.** We report FID, KID, and C-FID separately on the train and test splits. Lower is better for all metrics. The *Gap* column denotes the difference between the test-set mean and the train-set mean for each metric.

| Method | Split | Metric | Airplane | Car | Chair | Animals | Food | Plants | Bedroom | Mean | Gap |
|---|---|---|---|---|---|---|---|---|---|---|---|
| GET3D | Train | FID | 72.61 | 109.35 | 75.89 | 104.69 | 108.82 | 122.69 | 248.77 | 120.42 | – |
| | | KID | 57.66 | 83.83 | 48.71 | 83.22 | 79.54 | 110.03 | 285.11 | 106.87 | – |
| | | C-FID | 7.18 | 14.83 | 15.23 | 14.59 | 18.79 | 15.97 | 60.54 | 21.02 | – |
| | Test | FID | 72.06 | 109.00 | 77.45 | 108.56 | 113.51 | 130.94 | 247.56 | 122.73 | **+2.31** |
| | | KID | 55.24 | 82.20 | 48.37 | 85.71 | 79.29 | 113.34 | 284.97 | 107.02 | **+0.15** |
| | | C-FID | 7.03 | 14.87 | 15.33 | 14.97 | 19.27 | 16.41 | 60.42 | 21.19 | **+0.17** |
| DiffGS | Train | FID | 65.18 | 128.01 | 75.21 | 199.53 | 171.39 | 160.78 | 376.51 | 168.09 | – |
| | | KID | 52.76 | 114.64 | 55.45 | 100.35 | 73.48 | 111.00 | 346.70 | 122.05 | – |
| | | C-FID | 10.57 | 14.94 | 17.42 | 33.98 | 32.70 | 22.18 | 58.87 | 27.24 | – |
| | Test | FID | 64.48 | 126.36 | 76.66 | 208.37 | 174.57 | 166.46 | 383.60 | 171.50 | +3.41 |
| | | KID | 50.19 | 111.33 | 55.00 | 113.04 | 74.09 | 113.27 | 358.64 | 125.08 | +3.03 |
| | | C-FID | 10.69 | 15.19 | 17.38 | 36.99 | 33.29 | 22.47 | 57.96 | 27.71 | +0.47 |
| GaussianCube | Train | FID | 17.84 | 20.92 | 23.45 | 131.05 | 96.03 | 127.39 | 96.78 | 73.35 | – |
| | | KID | 11.92 | 13.26 | 14.40 | 99.38 | 67.98 | 98.15 | 97.33 | 57.49 | – |
| | | C-FID | 3.28 | 4.43 | 6.24 | 16.32 | 15.46 | 17.07 | 22.19 | 12.14 | – |
| | Test | FID | 21.67 | 23.60 | 24.77 | 139.69 | 97.48 | 132.71 | 98.91 | 76.98 | +3.63 |
| | | KID | 14.53 | 14.98 | 14.50 | 108.09 | 69.65 | 93.75 | 98.46 | 59.14 | +1.65 |
| | | C-FID | 3.83 | 4.81 | 6.11 | 22.90 | 17.12 | 21.39 | 22.50 | 14.09 | +1.95 |
| AtlasGaussians | Train | FID | 12.87 | 19.79 | **13.46** | 87.98 | 73.85 | 89.70 | 74.86 | 53.22 | – |
| | | KID | 6.34 | 12.23 | 4.61 | 65.22 | 43.41 | 56.80 | 62.69 | 35.90 | – |
| | | C-FID | 2.76 | 3.36 | 2.72 | 12.83 | 13.96 | 13.20 | 15.18 | 9.14 | – |
| | Test | FID | 14.26 | 23.05 | **13.46** | 91.79 | 105.46 | 94.53 | 74.56 | 59.59 | +6.37 |
| | | KID | 7.18 | 14.24 | 4.80 | 67.49 | 64.97 | 57.00 | 62.95 | 39.80 | +3.90 |
| | | C-FID | 3.00 | 3.77 | 2.93 | 13.33 | 17.24 | 13.93 | 15.17 | 9.91 | +0.77 |
| TRELLIS* | Train | FID | 21.03 | 27.96 | 16.24 | 91.16 | 88.20 | 109.11 | 37.03 | 55.82 | – |
| | | KID | 14.50 | 21.84 | 13.12 | 67.63 | 54.70 | 65.69 | 33.61 | 38.73 | – |
| | | C-FID | 4.84 | 5.99 | 4.35 | 14.31 | 15.98 | 13.15 | 8.30 | 9.56 | – |
| | Test | FID | 25.75 | 32.95 | 18.05 | 93.63 | 98.71 | 124.68 | 38.56 | 61.76 | +5.94 |
| | | KID | 17.96 | 26.66 | 14.01 | 68.19 | 64.70 | 85.03 | 34.12 | 44.38 | +5.65 |
| | | C-FID | 6.28 | 7.31 | 4.58 | 15.84 | 17.94 | 17.23 | 8.47 | 11.09 | +1.53 |
| Trip2GS (ours) | Train | FID | **12.24** | **18.63** | 14.13 | **67.25** | **55.21** | **67.09** | **16.30** | **35.84** | – |
| | | KID | **5.76** | **9.56** | **3.97** | **49.74** | **31.91** | **46.56** | **9.10** | **22.37** | – |
| | | C-FID | **1.28** | **2.54** | **1.96** | **8.31** | **10.45** | **7.75** | **2.87** | **5.02** | – |
| | Test | FID | **12.71** | **19.60** | 17.14 | **75.07** | **66.92** | **69.41** | **18.43** | **39.90** | +4.06 |
| | | KID | **4.47** | **10.01** | **3.92** | **56.43** | **40.66** | **47.97** | **9.10** | **24.65** | +2.28 |
| | | C-FID | **1.39** | **2.77** | **2.18** | **8.80** | **11.89** | **8.17** | **3.21** | **5.49** | +0.47 |

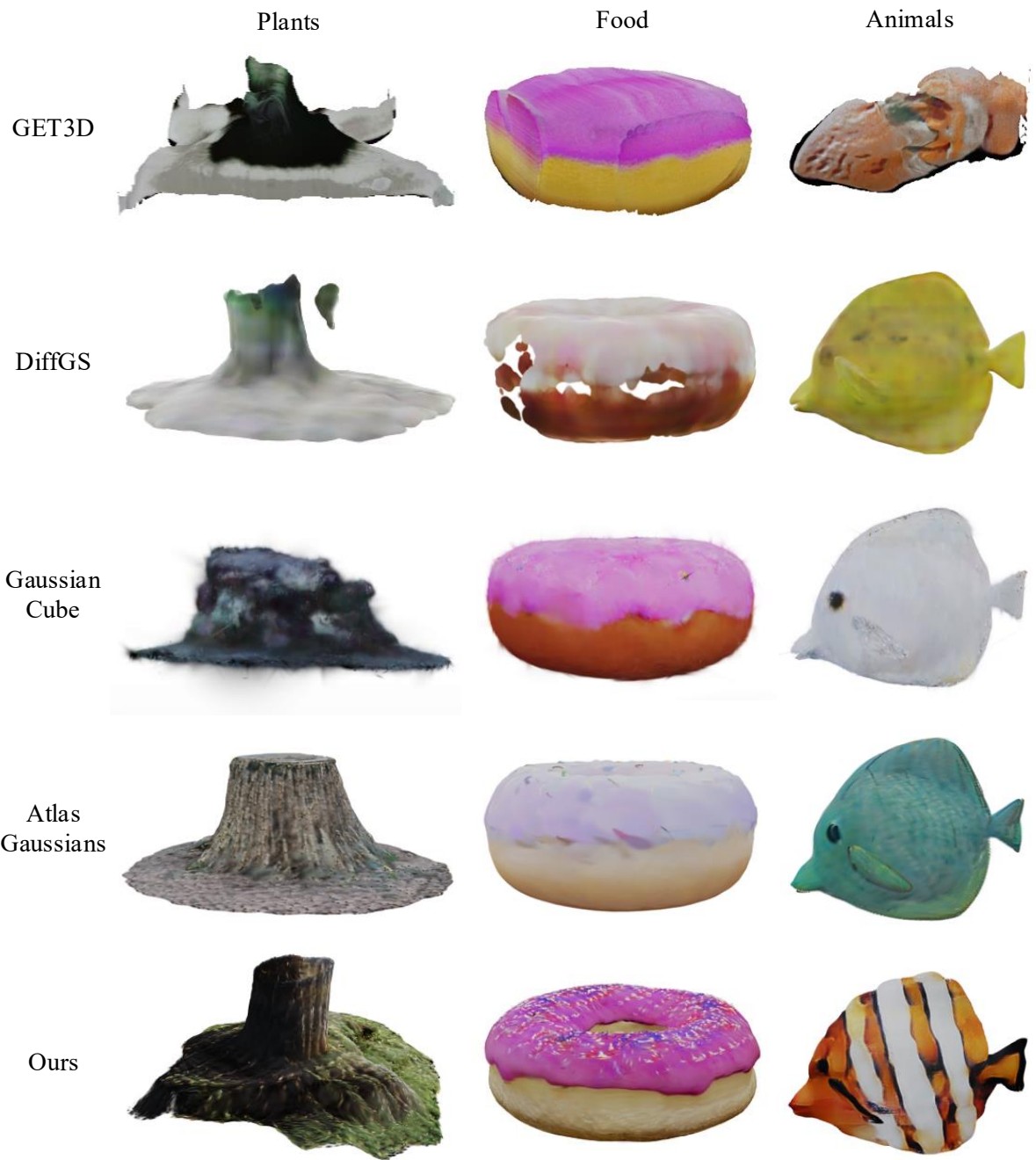

Figure 25: **Detailed comparison of unconditional generation on G-buffer Objaverse plants, food, and animals categories.**

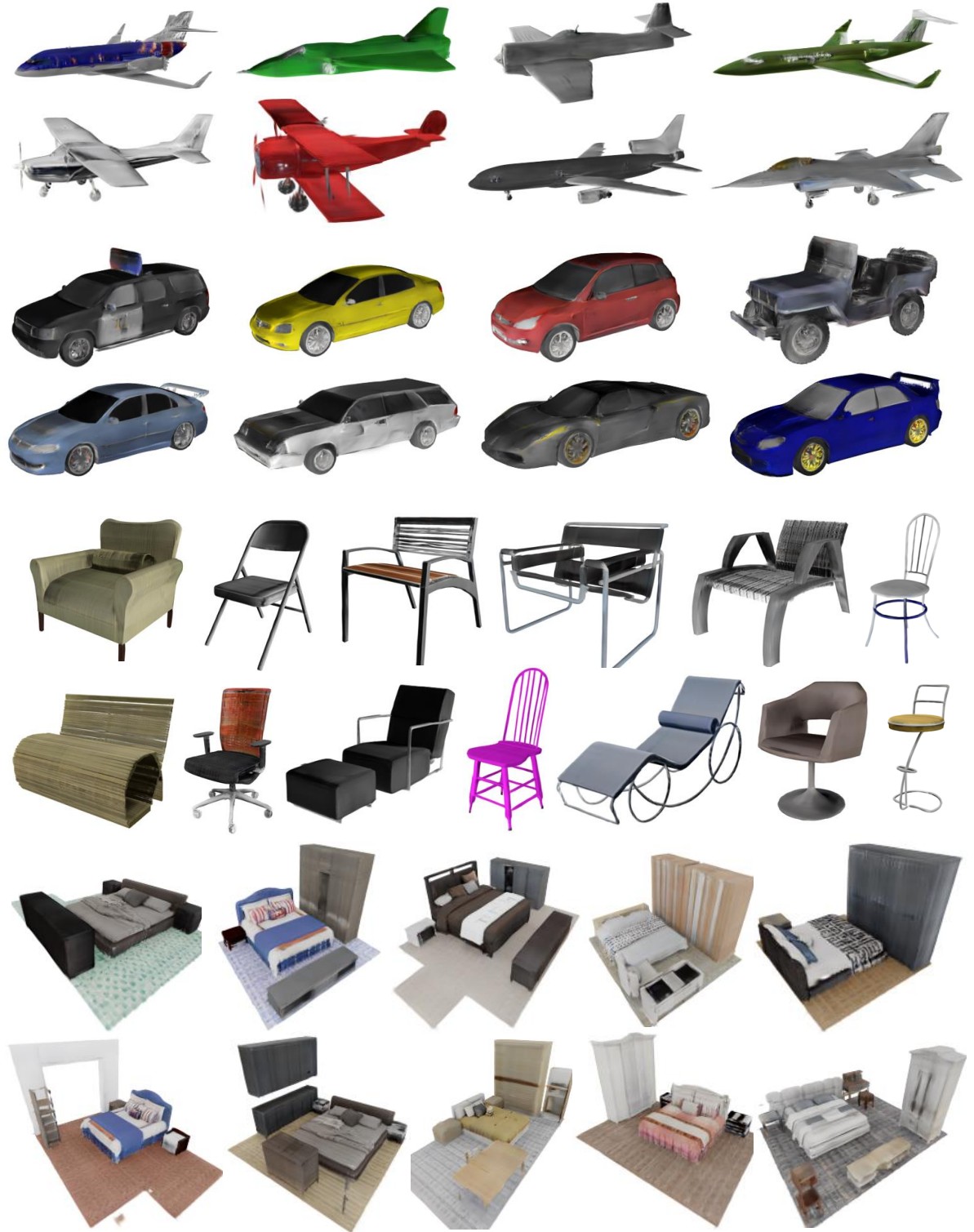

Figure 26: **Various unconditional generation results.**

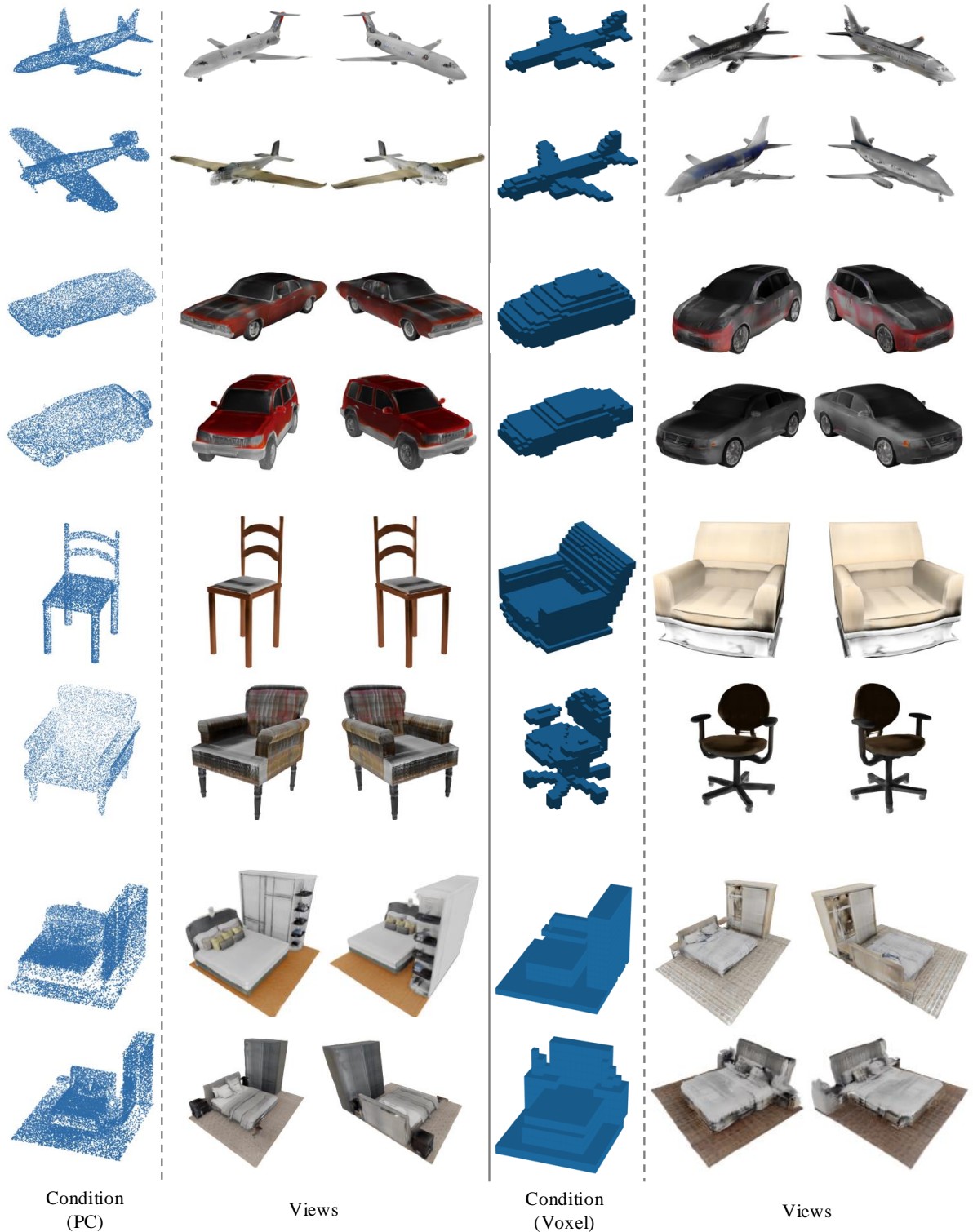

Figure 27: **Various point cloud-conditioned or voxel-conditioned generation results.**

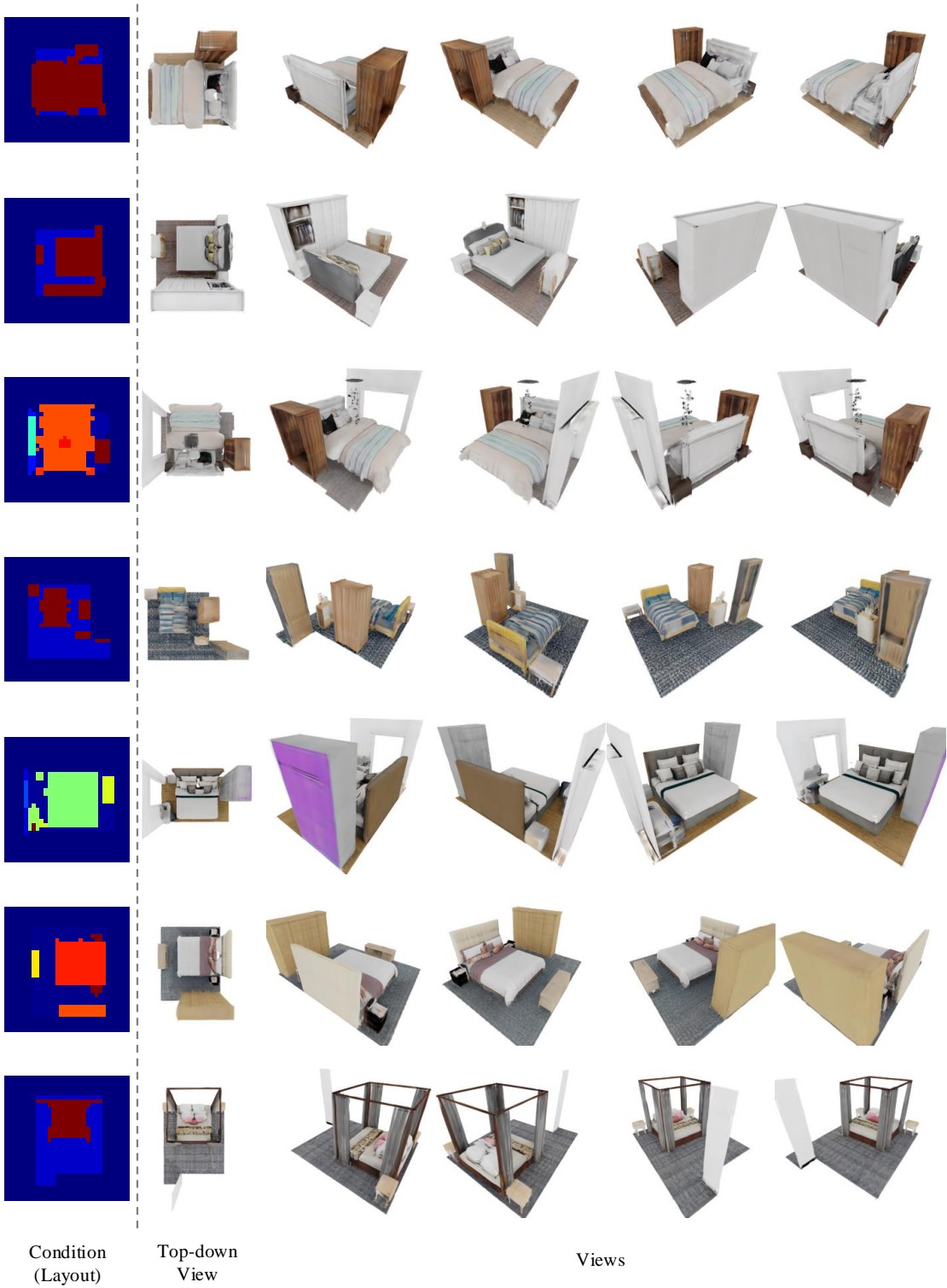

Condition
(Layout)

Top-down
View

Views

Figure 28: **Various layout-conditioned generation results on the 3D-FRONT bedroom category.**

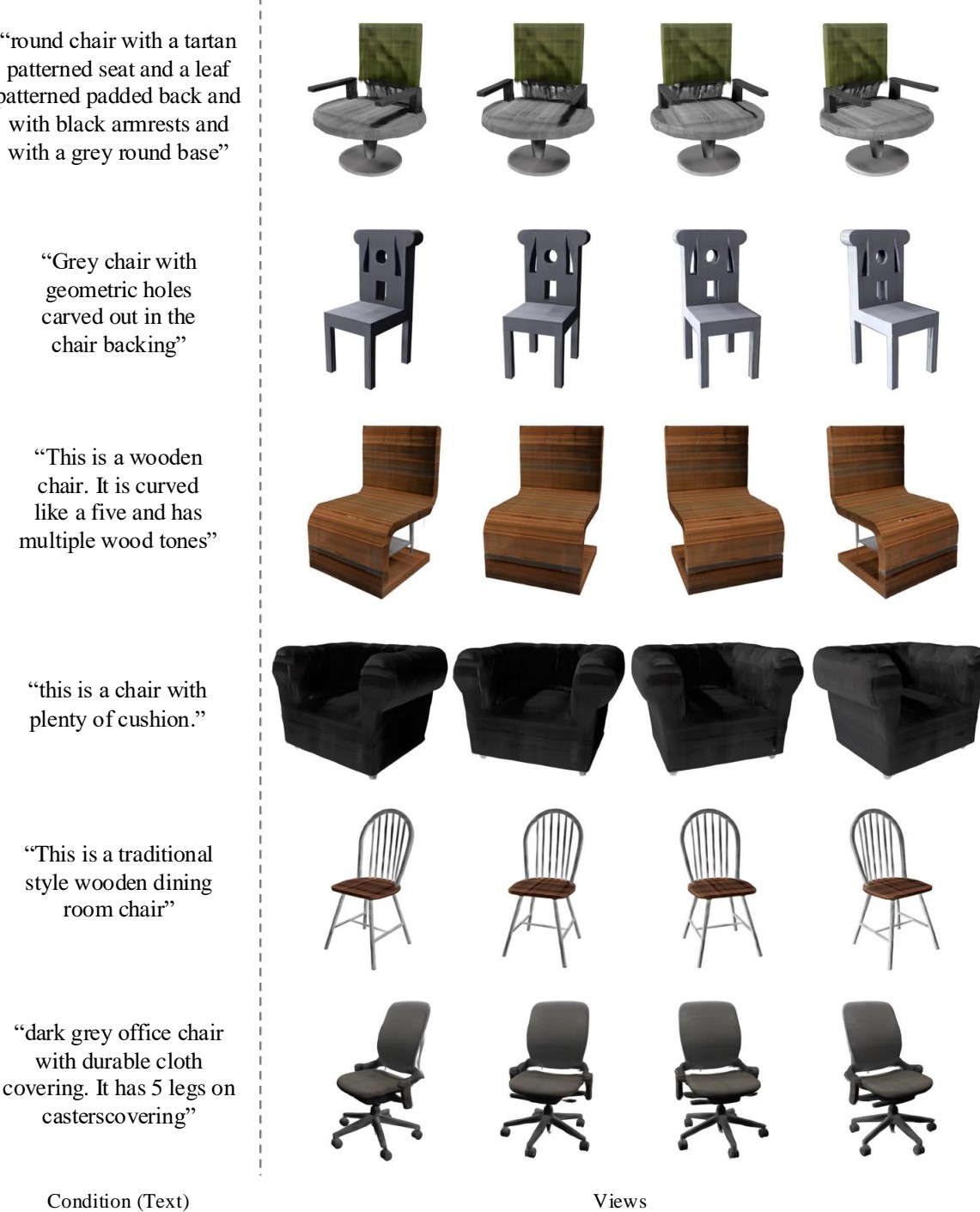

"round chair with a tartan patterned seat and a leaf patterned padded back and with black armrests and with a grey round base"

"Grey chair with geometric holes carved out in the chair backing"

"This is a wooden chair. It is curved like a five and has multiple wood tones"

"this is a chair with plenty of cushion."

"This is a traditional style wooden dining room chair"

"dark grey office chair with durable cloth covering. It has 5 legs on casterscovering"

Condition (Text)                    Views

Figure 29: **Various text-conditioned generation results on the ShapeNet chair category.**

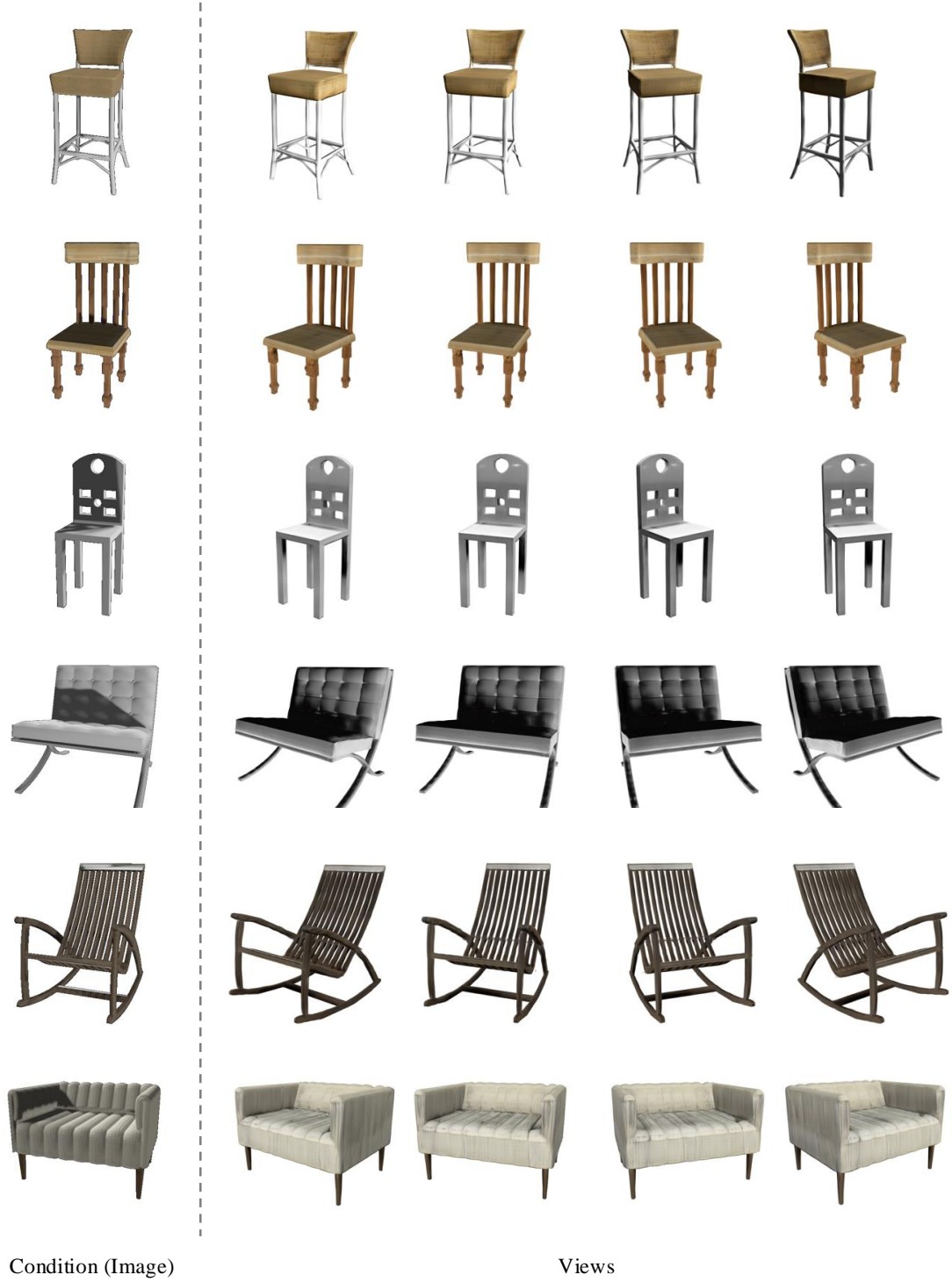

Condition (Image)                    Views

Figure 30: **Various image-conditioned generation results on the ShapeNet chair category.**

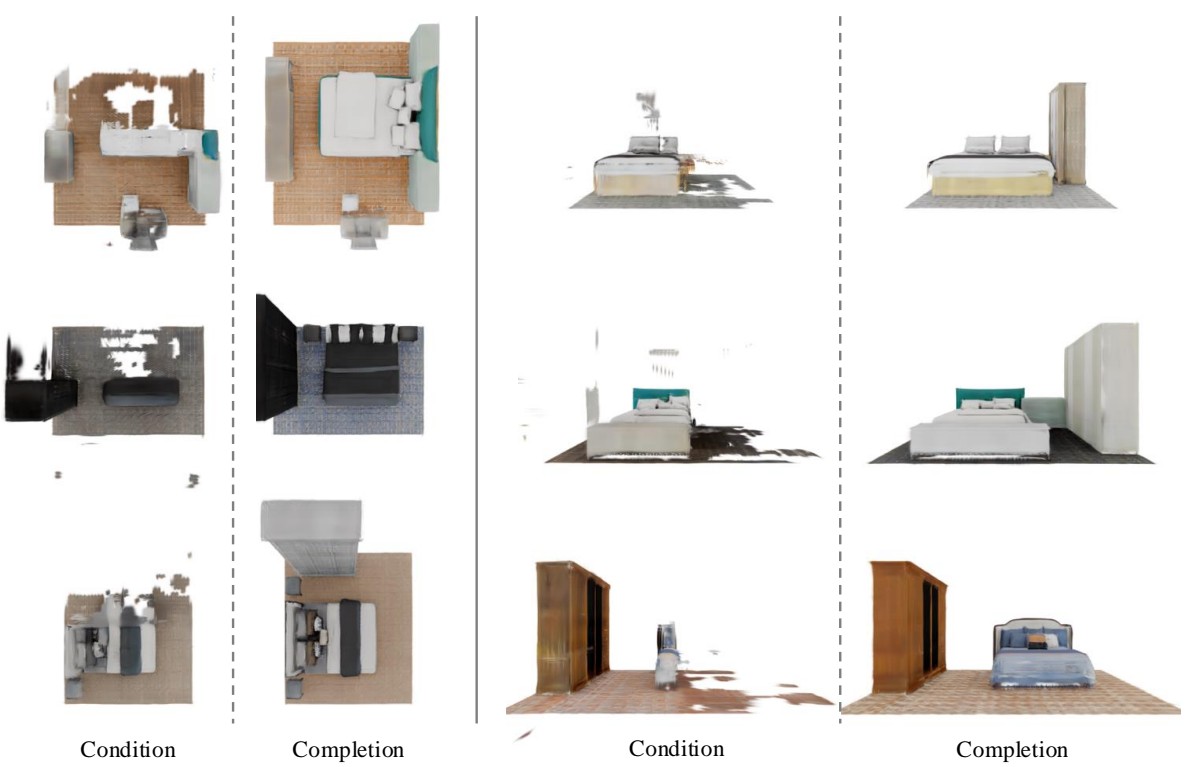

Condition          Completion                    Condition          Completion

Figure 31: **Various scene completion results on the 3D-FRONT bedroom category.**

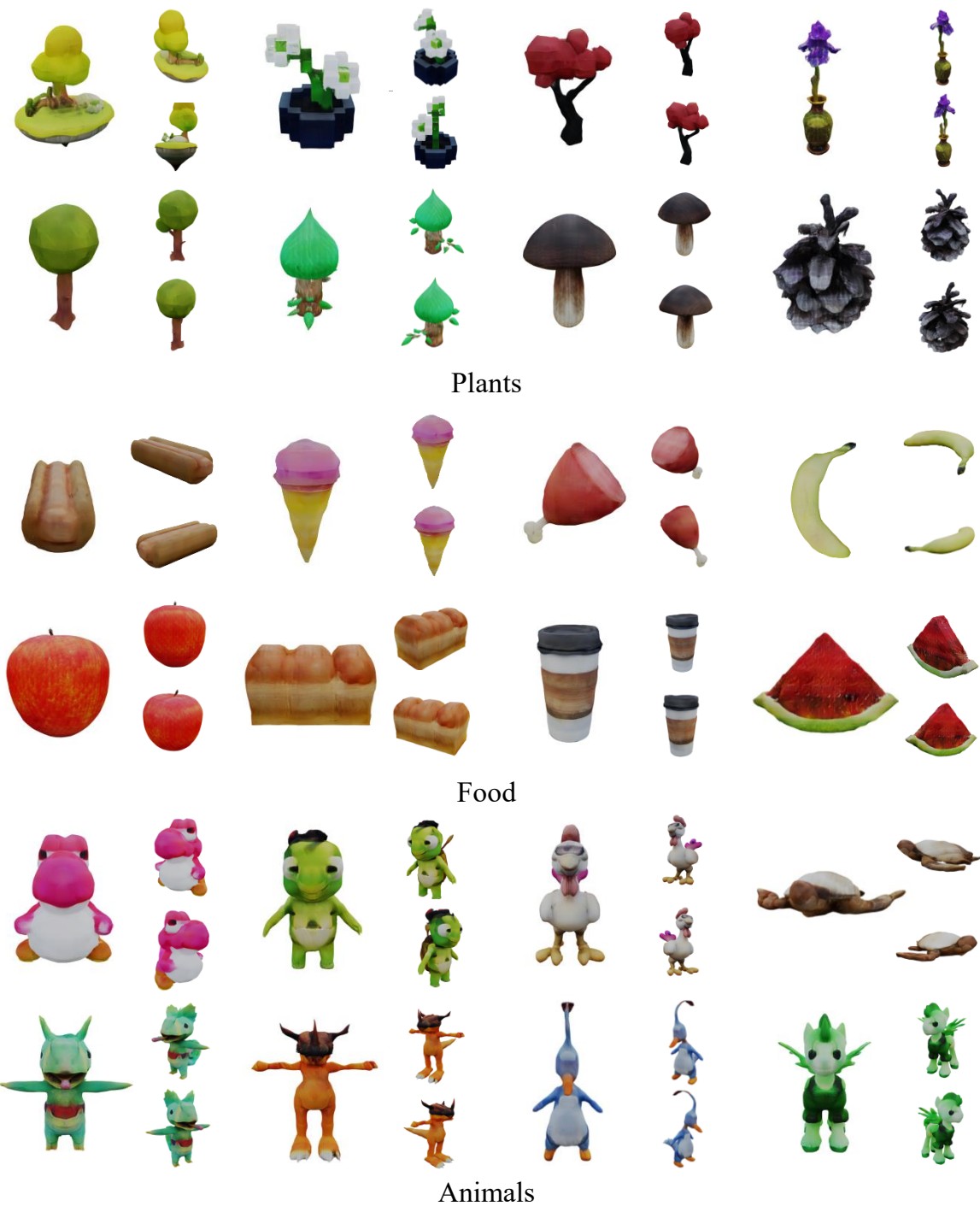

Figure 32: **Various unconditional generation results on the G-buffer Objaverse plants, food, and animals categories.**

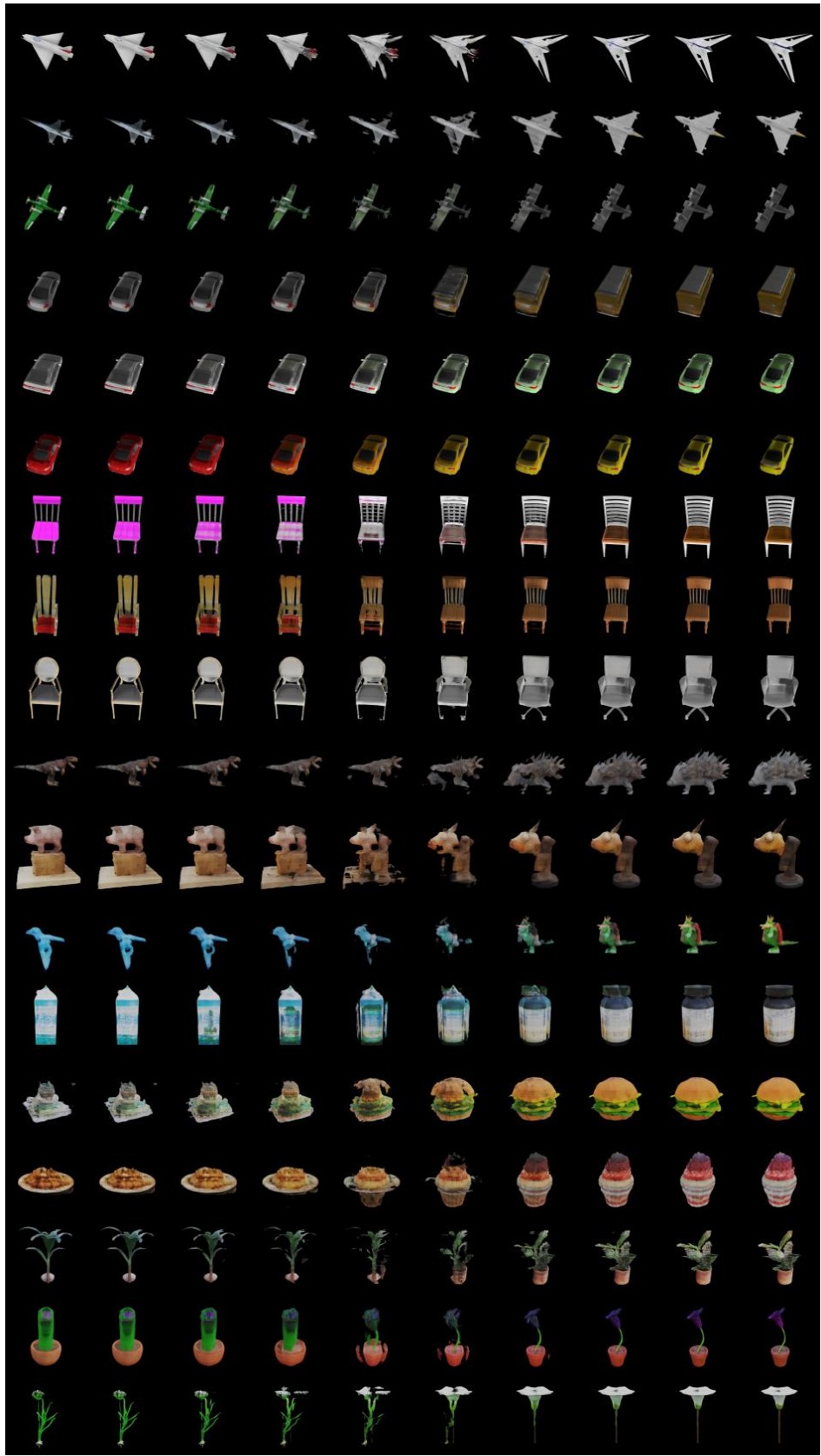

Figure 33: **Feature interpolation results on ShapeNet and G-Objaverse datasets.** The results show smooth transition between two different objects, which means Trip2GS learned smooth underlying distribution of 3D assets.

