# OpenReview forum: "Trip-to-Gaussian: A Versatile Framework for Unconditional 3D Generation"
_TMLR — Accepted by TMLR_

### Review · Reviewer_CUbf · 2026-02-17

**Summary Of Contributions:**

This paper presents a method called Trip2GS that generates 3D Gaussians using triplane as an intermediate representation. Specifically, the authors design a VAE that first transforms point clouds into triplane features, which are then projected into surface occupancy fields to extract coarse anchor point at surface and further upsampled to dense points for GS extraction. The module is trained with rendering loss, geometry loss and regularization loss. The authors validated their method on both conditional and unconditional 3D generation and demonstrated good results.

**Audience:**

Yes

**Audience Explanation:**

This paper uses maching learning for 3D generation, which is an active research area in ML.

**Claims And Evidence:**

No

**Claims Explanation:**

This paper lacks results of videos or visualizations in 3D form.

**Requested Changes:**

This paper overall looks okay to me, but I have following questions:
- The first question: in what situation or applications that make unconditional generation important, it is strongly related to the motivation of this paper but are less discussed.
- Although this paper claims to do scene-level generation, it is still limited to simple datasets like 3D-front, which seems to me not much more complex that other object-level datasets. The authors should clarify is there any component that makes the model works on it and other models cannot or adjust the tone of scene-level part.
- How the proposed method deals with non-convex objects since it will be occluded in the triplane features.
- The authors are encouraged to incoporate some of the method details and equations (A.2) in the appendix to the main paper to increase the clarity of the paper.
- The authors are strongly suggested to include video and 3D visualization results and comparisons to demonstrate the effectiveness of the proposed method.
- I don't quite understand why TRELLIS fail on unconditional generation. Its VAE also seems to produce better results than Trip2GS.
- Lack of comparison of TRELLIS on the conditional generation setting.

---

> ### Author Response · Authors · 2026-03-23
> **Brief version of response to Reviewer CUbf**
>
> For the reviewer’s convenience, all revised or newly added contents are marked in blue in both the main paper and the supplementary material.
>
> **Q1.  In what situation or applications that make unconditional generation important?**
>
> We thank the reviewer for this important question. To better clarify our motivation for studying unconditional generation, especially under resource-constrained settings, we revised the first paragraph of the Introduction and added **Sec. I.1 in appendix**.
>
> From a research perspective, unconditional generation is a fundamental problem that emphasizes efficient and effective network architecture and pipeline design for generative modeling, rather than relying primarily on large-scale model or data scaling. In this sense, it serves as a foundational study for a wide range of downstream applications.
>
> Unconditional generation is also practically valuable. When only a limited number of 3D assets are available, it can be used to generate diverse but similar assets to expand an asset library. It can also serve as a form of data augmentation in low-data regimes, which is useful for simulation and related applications.
>
> We now discuss these motivations and use cases more explicitly in **Sec. I.1 in appendix**.
>
> **Q2. The scenes in 3D-FRONT seems not much more complex than other object-level datasets. What component makes the model works on the scene-level dataset and the others cannot?**
>
> We appreciate this comment and have added additional analysis to clarify this point.
>
> First, to quantitatively assess dataset complexity, in **Sec. E.1** we perform controlled 3DGS fitting experiments across multiple datasets using a fixed number of Gaussians and fixed Gaussian positions. The reconstruction quality consistently degrades from **ShapeNet → G-Objaverse → 3D-FRONT**, which supports our claim that 3D-FRONT is indeed more challenging in terms of both geometric and texture complexity.
>
> Second, to understand why our method works better than prior methods on 3D-FRONT, we provide additional baseline analysis and experiments in **Sec. E.2**, and further ablations on our own design in **Sec. E.3**. These results show that our **SOF-based coarse-to-fine generation scheme** is the key component: it enables both high-quality VAE reconstruction and a compact latent space, which together are crucial for scene-level generation
>
> **Q3. How can our triplane-based method deal with occluded parts of non-convex objects?**
>
> We now discuss this explicitly in **Section I.3 in appendix (Discussion on Triplane)**.
>
> **Q4. Please incorporate some method details and equations (A.2) in the appendix to the main paper.**
>
> We thank the reviewer for the suggestion and have revised the paper accordingly.
>
> Specifically, the **loss computation** and **training strategy** have now been moved to the main paper as **Section 3.3 (Training Objective)** and **Section 3.4 (Training Strategy)**, respectively. We did not move the detailed network architectures of **GIM, PUM, and the 3DGS decoder** into the main paper, since we believe it is important to distinguish the high-level methodological workflow from lower-level implementation details.
>
> **Q5. Please include video and 3D visualization results and comparisons.**
>
> We have added qualitative visualizations of unconditional generation results for both our method and the compared methods in the supplementary material. In addition, we also include visualizations for conditional generation results. These supplementary results provide clearer 3D and video-based comparisons of the generated outputs.
>
> **Q6. Why TRELLIS fails on unconditional generation, even if produces better results for VAE compared to Trip2GS?**
>
> We thank the reviewer for pointing out this important issue. During reproduction, we found that the TRELLIS pipeline implicitly assumes that the data are normalized to a fixed range (specifically within **[-0.5, 0.5]**), and this assumption is effectively hard-coded in the generation pipeline. However, the datasets used in our experiments (**ShapeNet, G-Objaverse, and 3D-FRONT**) have different intrinsic scales and camera parameters. As a result, directly applying the TRELLIS generation pipeline to these datasets led to degraded generation quality.
>
> For a fair comparison, we aligned the TRELLIS preprocessing with the intrinsic scale of each dataset, and we clarified these preprocessing details in **Sec. B.1**. Please refer to **Sec. B** for further results.
>
> **Q7. Comparison to TRELLIS on the conditional generation setting.**
>
> Following the correction described in **Q6**, we first reproduced TRELLIS generation results more properly and then extended the comparison to the conditional generation setting.
>
> In particular, we compare our method with TRELLIS on **text-to-3D generation**, and the results are presented in **Sec. C.4**. As shown there, our method achieves both visually and quantitatively better performance than TRELLIS on **text-to-3D chair and car generation**.

---

> ### Author Response · Authors · 2026-04-03
> **Detailed version of response to Reviewer CUbf (Part 1/2)**
>
> For the reviewer’s convenience, all revised or newly added contents are marked in blue in both the main paper and the supplementary material.
>
> **Q1.  In what situation or applications that make unconditional generation important?**
>
> We thank the reviewer for this important question. To better clarify our motivation for studying unconditional generation, especially under resource-constrained settings, we revised the first paragraph of the Introduction and added **Section I.1 in appendix**.
>
> From a research perspective, unconditional generation is a fundamental problem that emphasizes efficient and effective network architecture and pipeline design for generative modeling, rather than relying primarily on large-scale model or data scaling. In this sense, it serves as a foundational study for a wide range of downstream applications.
>
> Unconditional generation is also practically valuable. When only a limited number of 3D assets are available, it can be used to generate diverse but similar assets to expand an asset library. It can also serve as a form of data augmentation in low-data regimes, which is useful for simulation and related applications. We now discuss these motivations and use cases more explicitly in **Section I.1 in appendix**.
>
> **Q2. The scenes in 3D-FRONT seems not much more complex than other object-level datasets. What component makes the model works on the scene-level dataset and the others cannot? OR Adjust the tone of scene-level part**
>
> We appreciate this comment and have added additional analysis to clarify this point.
>
> First, to quantitatively assess dataset complexity, in **Sec. E.1** we perform controlled 3DGS fitting experiments across multiple datasets using a fixed number of Gaussians and fixed Gaussian positions. The reconstruction quality consistently degrades from **ShapeNet → G-Objaverse → 3D-FRONT**, which supports our claim that 3D-FRONT is indeed more challenging in terms of both geometric and texture complexity.
>
> Second, to better understand why our method performs more robustly than prior approaches on 3D-FRONT, we analyze both the baseline pipelines (**Sec. E.2**) and our own design (**Sec. E.3**). We argue that modeling complex scene-level data requires two properties: **(1)** the latent representation must encode scene information sufficiently well, and **(2)** the decoder must faithfully reconstruct and decode that information. From this perspective, GET3D and DiffGS rely on vector latent representations, which are less well suited for encoding complex spatial structure at the scene level.
>
> GaussianCube applies diffusion directly to fitted 3DGS data, which enlarges the search space of the generative model and makes the modeling problem more difficult. In contrast, our method decomposes decoding into three stages: **(1) GIM for coarse geometry, (2) PUM for fine geometry, and (3) a 3DGS decoder for appearance**. The importance of this decomposition is supported by our ablations: **Fig. 7 and Tab. 4** (SOF vs. DMTet, corresponding to stages 1+2) and **Tab. 18 and Tab. 19** (PUM vs. Neural Gaussians, corresponding to stages 2+3) show that collapsing these stages into a more direct decoding process consistently degrades performance.
>
> AtlasGaussians adopts a two-stage decoding scheme in which geometry is represented by patches and appearance is decoded into 3DGS. In that framework, geometric structure is supervised through a Chamfer loss on patch centers. However, Chamfer loss tends to emphasize global geometric similarity and permits many-to-one matching, which makes it less effective for preserving fine details in complex 3D structures ([1, 2, 3, 4]). In our supplementary analysis, we improve this component by applying soft voxelization to the patch centers and supervising them with the proposed SOF loss, which leads to visible improvements (see **Fig. 14**).
>
> In **Sec. E.3**, we further analyze the versatility of our method from two perspectives. The first is **VAE reconstruction quality**: compared with other VAE-LDM style baselines, our method achieves consistently stronger reconstruction metrics. The second is **latent compactness and diffusability**: we find that SOF and TV regularization make the latent distribution easier for the diffusion model to learn (**Tab. 14**). Taken together, these results support our claim that **SOF and the coarse-to-fine decoding scheme are the key factors behind the versatility of Trip2GS**.
>
> We refer the reader to **Sec. E** for more detailed analysis.

---

> ### Author Response · Authors · 2026-04-03
> **Detailed version of response to Reviewer CUbf (Part 2/2)**
>
> **Q3. How can our triplane-based method deal with occluded parts of non-convex objects?**
>
> We now discuss this explicitly in **Section I.3 in appendix (Discussion on Triplane)**.
>
> While triplane representation projects 3D structure onto three orthogonal planes and may therefore suffer from occlusion on an individual plane, our method mitigates this issue in three ways. First, our **point cloud projection** encodes features together with depth information during projection onto the triplanes, which helps preserve spatial cues. Second, our **3D-aware convolution** compensates for 3D positional information that could otherwise be lost by vanilla 2D convolution. Third, our **3D supervision via the SOF loss** explicitly encourages the triplane features to learn coarse 3D anchor structure, thereby reinforcing the learning of 3D geometry.
>
> These points are now clarified in **Section I.3 in appendix**.
>
> **Q4. Please incorporate some method details and equations (A.2) in the appendix to the main paper.**
>
> We thank the reviewer for the suggestion and have revised the paper accordingly.
>
> Specifically, the **loss computation** and **training strategy** that were previously described in **Section A.2** of the appendix have now been moved to the main paper as **Section 3.3 (Training Objective)** and **Section 3.4 (Training Strategy)**, respectively. We did not move the detailed network architectures of **GIM, PUM, and the 3DGS decoder** into the main paper, since we believe it is important to distinguish the high-level methodological workflow from lower-level implementation details.
>
> Q5. **Please include video and 3D visualization results and comparisons.**
>
> We have added qualitative visualizations of unconditional generation results for both our method and the compared methods in the supplementary material. In addition, we also include visualizations for conditional generation results. These supplementary results provide clearer 3D and video-based comparisons of the generated outputs.
>
> **Q6. Why TRELLIS fails on unconditional generation, even if produces better results for VAE compared to Trip2GS?**
>
> We thank the reviewer for pointing out this important issue. During reproduction, we found that the TRELLIS pipeline implicitly assumes that the data are normalized to a fixed range (specifically within **[-0.5, 0.5]**), and this assumption is effectively hard-coded in the generation pipeline. However, the datasets used in our experiments (**ShapeNet, G-Objaverse, and 3D-FRONT**) have different intrinsic scales and camera parameters. As a result, directly applying the TRELLIS generation pipeline to these datasets led to degraded generation quality.
>
> For a fair comparison, we aligned the TRELLIS preprocessing with the intrinsic scale of each dataset, and we clarified these preprocessing details in **Sec. B.1**.
>
> In addition, reviewer **GZrc** requested a stricter comparison with better-matched model scale and hyperparameters for TRELLIS. In response, we conducted a more carefully controlled comparison between our model and TRELLIS, and we report the updated results in **Sec. B**. As shown in **Tab. 7**, the \texttt{Trip2GS} VAE achieves reconstruction performance comparable to that of the TRELLIS VAE. This result is particularly meaningful because our model attains it **without** relying on a foundation model (e.g., DINOv2), extra preprocessing, or multi-stage training, which further highlights the effectiveness of our unified training pipeline.
>
> Furthermore, in **Tab. 8**, we compare the TRELLIS flow model against our method under two matched settings: **(1)** comparable computational resources, and **(2)** comparable model scale and batch size. Under both settings, TRELLIS still shows a clear gap in generation quality, while our method consistently produces substantially better results. We believe this suggests that, in the limited-resource regime, generation quality depends more critically on effective pipeline design than on simply increasing model size or batch size. In particular, these results further support the effectiveness of our coarse-to-fine generation scheme.
>
> Due to the revision timeline, experiments on categories beyond **bedroom** are still in progress.
>
> **Q7. Comparison to TRELLIS on the conditional generation setting.**
>
> Following the correction described in **Q6**, we first reproduced TRELLIS generation results more properly and then extended the comparison to the conditional generation setting.
>
> In particular, we compare our method with TRELLIS on **text-to-3D generation**, and the results are presented in **Section C.4**. As shown there, our method achieves both visually and quantitatively better performance than TRELLIS on **text-to-3D chair and car generation**.

---

### Review · Reviewer_GZrc · 2026-03-02

**Summary Of Contributions:**

This paper proposes Trip2GS, a framework for unconditional 3D generation that can be extended to conditional generation, aiming to bridge structured triplane representations with unstructured 3D Gaussian Splatting (3DGS). The key idea is a coarse-to-fine generation scheme: a Gaussian Indicator Module (GIM) extracts coarse anchor points from triplane features via Surface Occupancy Fields (SOF), a Point Upsampling Module (PUM) densifies and smooths these points, and a 3DGS decoder predicts final Gaussian parameters, with all features derived from triplane queries. The framework consists of a Trip2GS VAE and a triplane diffusion model, and is evaluated on ShapeNet, G-Objaverse, and 3D-FRONT datasets for both object- and scene-level unconditional generation.

S:
* The coarse-to-fine design bridging triplane and 3DGS is well-motivated and effectively reduces the difficulty of data processing.
* The framework scales from object-level to scene-level within a unified architecture, achieving favorable scene-level generation results on 3D-FRONT.
* No external 3DGS fitting stage is required, simplifying the overall pipeline.
* Comparative and ablation experiments are comprehensive, with rich supporting material.
* The writing is clear and easy to follow.

W:
* The individual modules (GIM, PUM, SOF) are technically straightforward, and the overall novelty is somewhat limited.
* While SOF avoids the non-differentiable Marching Cubes operation, it is essentially an occupancy field learning approach and does not yield an explicit geometric structure.
* Scene-level evaluation is limited to a single room type with a small-scale dataset.
* Only 2D image-based metrics (FID/KID) are employed, with no direct evaluation of 3D geometric quality.
* Lack of training on large-scale datasets to verify the scalability of architecture. Whether triplane representations can be effectively applied to large-scale 3D datasets such as Trellis500K remains to be discussed.

**Audience:**

Yes

**Audience Explanation:**

Given the rapid adoption of 3DGS in various applications, the triplane-to-3DGS bridging approach provides valuable insights for the community. However, the generalization capability of triplane representations still requires further evaluation, particularly on large-scale, category-agnostic datasets.

**Claims And Evidence:**

Yes

**Claims Explanation:**

1. The quantitative results (Tables 2-3) and qualitative comparisons (Figures 5-6) convincingly demonstrate that Trip2GS outperforms baselines on ShapeNet and G-Objaverse across FID, KID, and C-FID metrics. The improvements are consistent across categories.
2. The 3D-FRONT bedroom results show a large margin over baselines (FID 16.30 vs. 74.86 for the next best). However, evaluation is limited to one room type, which weakens the generality of the "versatility" claim. More scene categories and scene-specific metrics would strengthen this.
3. The ablations on SOF vs. SDF/DMTet (Table 4), PUM upsampling (Table 5), and TV regularization (Table 6) are well-designed and clearly demonstrate the contribution of each component. However, the lack of explicit mesh 3D comparisons weakens the support for claims regarding fine geometric details.
4. The authors retrained TRELLIS for comparison on small-scale datasets, but it would be preferable to adjust the network architecture to the same level to further verify the superiority of the architecture. Under the current qualitative and quantitative comparison results (Secion B), there is no clear advantage.
5. The paper lacks 3D geometric evaluation metrics (e.g., Chamfer Distance on generated samples), which would more directly validate the claimed geometric fidelity improvements. The reliance on 2D rendering-based metrics alone leaves the 3D quality claims partially unsubstantiated.

**Requested Changes:**

1. The current evaluation relies entirely on 2D image metrics (FID, KID, C-FID). Please report Chamfer distance, F-score, or similar 3D metrics. Given that geometric quality is the core claim of the paper, this is essential. Optionally, meshes can be generated directly from higher-resolution occupancy fields using Marching Cubes (MC), or an SDF field can be estimated. Otherwise, it is recommended to clarify the paper's contributions.
2. Consider evaluating additional room types from 3D-FRONT or other scene datasets. Also consider scene-specific metrics such as layout accuracy or per-object quality. Consider autonomous driving scene generation like [A] Uniscene: Unified occupancy-centric driving scene generation.
3. No failure cases are shown. Understanding when and why the method fails would be informative for the community.
4. Revise the TRELLIS comparison. The current comparison (Section B) has improper controls; it is recommended to use fair comparisons with the same model scale parameters and K ratios.
5. The gradient control strategy and staged training (Appendix A.2) suggest that VAE is not trained end-to-end; it is recommended to clarify this description in the main text.
6. The fixed resolution of 64 is used throughout. How does this affect fine-grained structures at scene level? An ablation varying GIM resolution would be helpful.
7. Add discussion on model and data scalability.
8. Recommending some references: [B] Hi3dgen: High-fidelity 3d geometry generation from images via normal bridging, ICCV 2025; [C] Native and compact structured latents for 3d generation, CVPR 2026

---

> ### Author Response · Authors · 2026-03-23
> **Brief version of response to Reviewer GZrc**
>
> For the reviewer’s convenience, all revised or newly added contents are marked in blue in both the main paper and the supplementary material.
>
> **Q1. 3D metrics.**
>
> We added new geometric evaluations in Sec. F. For the intermediate geometry, the key property is not exact mesh accuracy but reliable surface-region prediction; thus we report mIoU against voxelized ground truth (Fig. 15, Tab. 16), showing that SOF preserves thin structures better than DMTet. For the final 3DGS output, we treat Gaussian centers as a point cloud and report MMD/COV against ground-truth test point clouds (Sec. F.2). Our method improves not only texture quality but also geometry-related metrics.
>
> **Q2. Additional room types / scene metrics.**
>
> We added experiments on the 3D-FRONT library category in addition to bedroom (Sec. H.1). Although library is much smaller and therefore harder for distribution learning, our method still shows stronger qualitative and quantitative results, further supporting the importance of our coarse-to-fine generation scheme. Layout-based metrics are not directly applicable because, unlike layout-based methods, our approach performs full-scene feed-forward generation of the entire scene geometry. Instead, we report MMD/COV for scene-level evaluation (Sec. F.2).
>
> **Q3. Failure cases.**
>
> We now discuss failure cases in Sec. I.4. As scene scale increases, a limited latent must represent much more geometry and appearance, so fine details can be lost. This is especially evident in 3D-FRONT living-room scenes, whose normalized scale is about 6.0 versus 3.0 for bedrooms, and which also contain more diverse and smaller objects. We discuss possible extensions such as autoregressive scene expansion or inpainting-based generation.
>
> **Q4. TRELLIS comparison.**
>
> During reproduction, we found that TRELLIS implicitly assumes data normalized to [-0.5, 0.5], whereas our datasets have different intrinsic scales and camera parameters. Directly applying TRELLIS without adjustment therefore degraded generation quality. We aligned TRELLIS preprocessing to each dataset scale and clarified this in Sec. B.1. Following the reviewer’s suggestion, we also performed stricter comparisons with better-matched TRELLIS model scale and hyperparameters (Secs. B.1–B.2). Under this setup, our Trip2GS VAE remains close to the TRELLIS VAE (Tab. 7) despite not using DINOv2, extra preprocessing, or multi-stage training, and our method still achieves clearly better generation quality (Tab. 8). For bedroom, scaling up TRELLIS yields only limited gains despite much higher cost, suggesting that under limited resources, effective pipeline design matters more than simply increasing model size or batch size.
>
> **Q5. “End-to-end” wording.**
>
> We agree that this should be clarified. Although SOF is differentiable, we intentionally stop gradients from the final 3DGS rendering loss through our gradient-control strategy; otherwise SOF quality becomes unstable. In this sense, our training differs from the usual meaning of end-to-end, where output gradients flow through the entire model. We therefore removed the “end-to-end” wording and replaced it with “unified pipeline/training”, which more accurately reflects that our framework maps input mesh to output 3DGS within one VAE framework without separate preprocessing, post-processing, or independently trained geometry/texture modules.
>
> **Q6. GIM resolution.**
>
> We added experiments in Sec. I.3 on chair and bedroom with GIM resolutions 48/64/96. Increasing resolution does not always improve fine details; instead, we observe a trade-off between the number of Gaussians and geometric fidelity. Very low resolution limits detail, while very high resolution increases Gaussians but degrades geometric fidelity. Resolution 64 gives the best balance. We also show in Sec. G.2 that the key module for recovering fine details is PUM, and increasing its upsampling ratio further improves detail reconstruction.
>
> **Q7. Model/data scalability.**
>
> Our main focus is generation quality under limited data and compute, which we believe is practically important. We now discuss scalability in Sec. I.4. For model scalability, a larger variant improves reconstruction on the challenging living-room category, suggesting that more capacity helps with large and complex scenes. For data scalability, while we do not directly train on very large-scale datasets here, recent works show that triplane features can scale well. More importantly, our core contribution is the coarse-to-fine decoding from structured latent features to unstructured outputs, which is not limited to triplanes or 3DGS and can extend to other structured latents and output formats such as point clouds or meshes.
>
> **Q8. Suggested references.**
>
> We thank the reviewer for these helpful suggestions and have added them to the Introduction as representative recent state-of-the-art conditional 3D generation methods.

---

> ### Author Response · Authors · 2026-04-03
> **Detailed version of response to Reviewer GZrc (Part 1/3)**
>
> For the reviewer’s convenience, all revised or newly added contents are marked in blue in both the main paper and the supplementary material.
>
> **Q1. As geometric quality is the core claim of the paper, please report 3D metrics (e.g., Chamfer distance, F-score). Otherwise, it is recommended to clarify the paper’s contributions.**
>
> We thank the reviewer for this important comment. To better evaluate geometric quality, we added two sets of experiments in **Section F**.
>
> Importantly, we would like to emphasize that the key property of the **intermediate geometry representation** is different from that of the **final 3DGS output**. For the intermediate representation, as discussed in **Section 4.3** of the main paper, what matters is the **fidelity and reliability of surface-region prediction**, rather than mesh-level accuracy. To evaluate this, we discretize the 3D space into voxels and measure **mIoU** against the ground-truth voxelized geometry (see **Fig. 15** and **Tab. 16**). The results show that, unlike DMTet, which tends to lose thin structures and details, our **SOF** provides more reliable surface reconstruction.
>
> For the final **3DGS output**, while 3DGS is not itself an explicit surface representation, it is still desirable for the Gaussians to adhere closely to object surfaces. To evaluate this property, we treat the **3DGS centers as a point cloud** and report **MMD/COV** against the ground-truth test point clouds. As shown in **Section F.2**, our method outperforms prior approaches not only in texture quality but also in geometry-related metrics.
>
> **Q2. Evaluation on additional room types from 3D-FRONT and scene-specific metrics (e.g., layout accuracy, per-object quality).**
>
> To further strengthen the versatility of our method at the scene scale, we added experiments on the **library** category in addition to **bedroom**, as reported in **Section H.1**.
>
> The library category is particularly challenging because it contains substantially fewer training samples than bedroom, making it harder for a generative model to capture the underlying data distribution and, in turn, the geometric structure. Nevertheless, our method still shows clearly superior qualitative and quantitative performance compared with the baselines, which further supports the importance of our **coarse-to-fine generation scheme**.
>
> Regarding scene-specific metrics such as layout accuracy or per-object quality, our method is fundamentally different from **layout-based scene generation** methods such as ATISS[5], DiffuScene[6], InstructScene[7], LayoutGPT[8], and DirectLayout[9], which generate scenes as a set of object bounding boxes or object-wise layouts. In contrast, our method performs **full-scene feed-forward generation** of the entire scene geometry as a single structure. Therefore, layout-based or object-instance-based metrics are not directly applicable. Instead, we report **MMD/COV** for scene-level generation on the bedroom category in **Section F.2**, where our method shows substantially better results.
>
> - [5] “ATISS: Autoregressive Transformers for Indoor Scene Synthesis”, NeurIPS 2021
> - [6] “DiffuScene: Denoising Diffusion Models for Generative Indoor Scene Synthesis”, CVPR 2024
> - [7] “InstructScene: Instruction-Driven 3D Indoor Scene Synthesis with Semantic Graph Prior”, ICLR 2024
> - [8] “LayoutGPT: Compositional Visual Planning and Generation with Large Language Models”, NeurIPS 2023
> - [9] “Direct Numerical Layout Generation for 3D Indoor Scene Synthesis via Spatial Reasoning”, NeurIPS 2025
>
> **Q3. Please showcase some failure cases and describe when and why it appears.**
>
> We agree that analyzing failure cases is very important, and we now discuss this in **Section I.4 in appendix**.
>
> Like other 3D unconditional generation methods, our approach becomes more challenged as the scene scale increases, because a limited latent representation must model a much larger amount of geometric and appearance information. As a result, fine details can be lost. This issue becomes more severe when a large scene contains many small objects, since each object occupies only a small fraction of the overall representational budget.
>
> A representative example is the **3D-FRONT living room** category. After preprocessing, its normalized scale is approximately **6.0**, whereas the bedroom category is around **3.0**. In addition, living-room scenes typically contain more diverse, numerous, and smaller objects. This makes it harder for the model to capture the details of each object. We therefore discuss in **Section I.4 in appendix** that large-scale scene generation may require extensions such as **autoregressive scene expansion or inpainting-based generation**.

---

> ### Author Response · Authors · 2026-04-03
> **Detailed version of response to Reviewer GZrc (Part 2/3)**
>
> **Q4. TRELLIS comparison: Use the same model scale parameters and K ratios for the fair comparison.**
>
> We thank the reviewer for raising this important point. During reproduction, we found that the TRELLIS pipeline implicitly assumes that the data lie within a fixed normalization range (specifically **[-0.5, 0.5]**), and this assumption is effectively hard-coded in the generation pipeline. However, the datasets used in our experiments (**ShapeNet, G-Objaverse, and 3D-FRONT**) have different intrinsic scales and camera parameters. Directly applying the TRELLIS generation pipeline without adjustment therefore led to degraded generation quality.
>
> To ensure a fair comparison, we aligned the TRELLIS preprocessing stage to the intrinsic scale of each dataset, and we clarified these preprocessing details in **Section B.1**.
>
> In addition, following the reviewer’s suggestion, we conducted stricter experiments in which the **model scale and hyperparameters of TRELLIS** were matched more closely to those of our method. These updated results are reported in **Sections B.1 and B.2**.
>
> Under this stricter setup, our **Trip2GS VAE** achieves performance close to the TRELLIS VAE (see Tab. 7), despite the fact that TRELLIS uses a strong foundation model (**DINOv2**), additional data construction steps, and multi-stage training, while our method uses **colored point clouds** as input. We believe this is already a highly encouraging result. More importantly, in actual generation quality, our method still performs substantially better.
>
> For the **bedroom** category, we further report two TRELLIS settings: one using a training budget similar to ours, and another with scaled-up model parameters and batch size. The latter requires much longer training time, yet yields only limited gains. This suggests that, under resource-constrained settings, **effective architectural design** has a greater impact on generation performance than simply increasing model size or batch size. At present, we report the bedroom results, and experiments on additional categories are ongoing.
>
> **Q5. The gradient control strategy and staged training suggest that VAE is not trained end-to-end; it is recommended to clarify this description in the main text.**
>
> Thank your for pointing this out. Although **SOF** is a differentiable representation unlike mesh point sampling, we intentionally **stop the gradients** from the output 3DGS rendering loss through our gradient control strategy.
>
> Empirically, when we do not stop these gradients, the predicted SOF quality degrades and the geometric fidelity becomes less stable. We therefore block these gradients so that the **GIM** can focus on predicting the **coarse regions where 3DGS should exist**, rather than being overly influenced by the final rendering objective.
>
> In this sense, our training should be distinguished from the usual meaning of **end-to-end**, where the output gradients flow through the entire model. To make this clearer, we removed the “end-to-end” wording from the **TRELLIS discussion in Section 2 (Related Work)** and from the second paragraph of **Section 4.3 (Ablation Study: Intermediate Geometry Representation)**.
>
> More broadly, we believe the term **“unified pipeline/training”** is more appropriate than **“end-to-end pipeline/training”**, especially in **Appendix B (Comparison to TRELLIS)**. What we intended to highlight is that our framework maps **input mesh to output 3DGS** within a single VAE training framework, without requiring separate stages for 3DGS preprocessing, post-processing, or independently trained geometry/texture modules. Compared with methods that require explicit 3DGS preprocessing (**GaussianCube, L3DG, DirectTriGS**), separate post-processing (**DiffGS**), or split geometry/texture modeling (**TRELLIS**), our **Trip2GS** performs a more **unified mesh-to-3DGS generation** within a single model.

---

> ### Author Response · Authors · 2026-04-03
> **Detailed version of response to Reviewer GZrc (Part 3/3)**
>
> **Q6. How does the resolution of GIM affect fine-grained structures at scene level?**
>
> To analyze the contribution of GIM to fine-grained structural modeling, we conducted additional experiments in **Section I.3 in appendix**. Specifically, we varied the GIM resolution across **48, 64, and 96** on the **chair** and **bedroom** categories, which contain diverse structures and many thin parts.
>
> Interestingly, increasing the GIM resolution does not always improve fine details. We interpret this as a **trade-off between the number of Gaussians and geometric fidelity**. As discussed above, the core role of GIM is to predict a **coarse but reliable surface prior**. When the resolution is too low, the **geometric fidelity (mIoU)** can improve, but the number of Gaussians becomes too limited to represent sufficiently detailed textures. When the resolution is too high, the number of Gaussians increases, but the geometric fidelity degrades, meaning that more Gaussians are allocated to less meaningful regions. We find that **resolution 64** provides the best balance and serves as a practical sweet spot. This observation is also consistent with prior work on efficient and compressed 3DGS using voxel-based structures, such as Scaffold-GS[10] and MasonGS[11].
>
> - [10] “Scaffold-GS: Structured 3D Gaussians for View-Adaptive Rendering”, CVPR 2024
> - [11] “MesonGS: Post-training Compression of 3D Gaussians via Efficient Attribute Transformation”, ECCV 2024
>
> As further shown in **Section G.2**, the key module for recovering fine details is not GIM but **PUM**. By increasing the **upsampling ratio of PUM**, we can effectively improve fine-grained detail reconstruction. Overall, our proposed **GIM+PUM** design alleviates this trade-off and enables better detail modeling.
>
> **Q7. Add discussion on model and data scalability. Whether triplane representations can be effectively applied to large-scale 3D datasets remains to be discussed.**
>
> Our main focus is the learning capability and generation quality of generative models under **limited data and compute resources**, which we believe is highly important for research labs, individual users, and practical deployment. Nevertheless, we agree that model/data scalability is an important issue, and we now discuss this in **Section I.4 in appendix (Model and Data Scalability)**.
>
> First, regarding **model scalability**, we conduct experiments on the **living-room** category, which is one of the most challenging large-scene cases discussed above. We compare our original model with a **larger variant** that has increased channel width and thus more parameters. Increasing the model size leads to consistent improvements in reconstruction metrics, and qualitatively it also recovers objects that were previously not modeled well. This suggests that larger model capacity indeed improves expressiveness and helps capture more complex structures and finer details.
>
> Second, regarding **data scalability**, while our paper does not directly include experiments on very large-scale datasets, recent works have shown that **triplane features can scale to high-quality 3D shape and texture generation on large-scale data**, for example:
>
> - [12] “Direct3D: Scalable Image-to-3D Generation via 3D Latent Diffusion Transformer”, NeurIPS 2024
> - [13] “SAR3D: Autoregressive 3D Object Generation and Understanding via Multi-scale 3D VQVAE”, CVPR 2025
>
> More importantly, we would like to emphasize that the core contribution of our paper is the **coarse-to-fine decoding process from structured latent features to unstructured outputs**. This idea is not limited to triplanes: it can also be applied to other structured representations such as **voxels** (e.g., in methods like **TRELLIS/TRELLIS2**), and it is also not limited to 3DGS as the output format, but can be extended to other unstructured outputs such as **point clouds** or **meshes**. In this paper, we show that this design yields better geometric quality, including **MMD/COV**, than prior methods. We therefore view our contribution as a **representation-agnostic coarse-to-fine generation scheme** rather than a method restricted to a specific triplane setup.
>
> Finally, from a practical standpoint, training on more data typically requires substantially larger VRAM capacity, faster GPUs, and many more iterations. For example, **TRELLIS** was trained using **64 A100 GPUs**. With more compute resources, we can also increase the batch size further, and our experiments already show that **VAE reconstruction quality improves consistently with larger training budgets**.
>
> **Q8. Recommending some references: [B] Hi3dgen: High-fidelity 3d geometry generation from images via normal bridging, ICCV 2025; [C] Native and compact structured latents for 3d generation, CVPR 2026.**
>
> We thank the reviewer for these helpful suggestions. We have added these works to the **Introduction** as representative examples of recent **state-of-the-art conditional 3D generation methods**.

---

### Review · Reviewer_jFLd · 2026-03-09

**Summary Of Contributions:**

This paper presents an approach to unconditional generation of 3D scenes represented as clouds of gaussian splats. The method is a latent diffusion model, operating over triplane features. The triplane features are decoded to splats in a slightly different way than prior works – first a continuous surface occupancy field is extracted, then this used to guide placement of splats, which are subsequently refined. The diffusion part is standard. The method is compared against several baselines from the last couple of years, on unconditional generation tasks (measured through render-space FID etc) on ShapeNet, ObjVerse, and 3D-FRONT. It is shown to yield higher performance than these on certain combinations of metrics and datasets.

**Additional Comments:**

n/a

**Audience:**

No

**Audience Explanation:**

Not yet – but they might be if several of the following issues were addressed.

The proposed pipeline is novel overall, with some specific components also having innovations versus standard parts. The idea of predicting a surface probability volume as an initial proxy for the gaussians is interesting, and to my knowledge has not previously been explored in this context.

However, the technical novelty is somewhat small – while the overall structure of the pipeline is novel and the VAE decoder has some changes versus standard practice, this is essentially a chain of off-the-shelf parts without deep technical innovation. If the results were stronger, that would counterbalance this concern; however, given that they are not, the limited technical substance becomes an issue.

There are no experiments on real-world data (except for the small fraction of 3D scans included in Objaverse). In particular, it is trivial to generate real-world datasets of comparable size to e.g. ShapeNet, by performing standard gaussian splatting reconstructions of multi-view image datasets such as CO3D and MVImgNet. In my view, this exclusive focus on synthetic data greatly limits interest to the community.

The supplementary material also includes conditional generation results in certain settings, with reasonable looking results, though no particular claims are made about these.

The paper is clear, well-organised and pleasant to read throughout – no issues in this regard.

**Broader Impact Concerns:**

This work does not raise any significant concerns.

**Claims And Evidence:**

No

**Claims Explanation:**

Not in the current form, though some additional experiments and adjustments to phrasing could perhaps mitigate this.

The results on unconditional generation show somewhat higher metrics than four baselines (GET3D, DiffGS, GaussianCube, Atlas-Gaussians) across three ShapeNet classes (chair, airplane, car), and on the bedroom class from 3D-FRONT. This holds across three different generative metrics calculated on rendered images – FID, KID, and CLIP-FID. Qualitative results on the same datasets are impressive, with crisp generations exhibiting a high level of detail, including realistic textures and geometry. On the G-Objaverse dataset (which defines some high-level semantic splits on a subset of Objaverse), results again are stronger than the baselines, both quantitative and qualitatively.

However, the main experiments measure unconditional generation performance on small (individual ShapeNet classes) and relatively small (one 3D-FRONT room category) datasets. There is no experiment demonstrating that the model is not simply memorising the training data, rather than learning a general model that can synthesise unseen instances. In particular, some of the ShapeNet chairs look extremely similar to examples in the training data. Such overfitting is not revealed by FID etc if they are computed with respect to the training set (and this is implied, since it is not said they use a validation split). This could be mitigated first by measuring FID with respect to an IID validation split (which means overfitting is penalised in the metrics); and the reader could be somewhat reassured memorisation is not occurring if the authors included visualisations of feature-space nearest neighbors. In addition, visualising latent-space interpolations between samples would be informative.

There are ablation experiments on some minor design decisions, showing that these improve the performance of the method noticeably. In particular, performance with the intermediate surface-occupancy field representation is compared with a version using a standard SDF, and DMTet; also usefulness of regularization is measured.

However, there is no evaluation specifically of the AE compared with those of the baselines in the main paper – whether the AE reconstructs better than baselines, or whether there is some other reason for the good generation results. Thus it is unclear what the cause of the good results is.

A key claim in the introduction is the lack of scalability of other methods from objects to scenes. However, it is not at all clear what technical aspect of the proposed method means that it should be better in this regard. There is no component here that is obviously better suited to both object-scale and scene-scale reconstruction than the baselines. This requires further justification if the claim is to be retained. If the evidence is merely empirical, this should be made clear, and also why the baselines might fail (and how their hyperparameters were adjusted in attempt to handle different scene scales / complexities).

It is unclear if the rendering loss guides the GIM part, i.e. the first part of the decoder that predicts an occupancy field. If it does not (as appears to be the case, since from the discussion there seems to be a non-differentiable step), then the method becomes an ad-hoc chain of several models, and would mean the specific counter-claim (versus TRELLIS; "TRELLIS have shown …") near the end of the introduction seems unjustified.

The most relevant baseline results (generation comparison against the unconditional version of TRELLIS, given in the supplementary only) do not seem to have been done very thoroughly – the paper shows good reconstruction results, but then says it was not possible to train the generative part, with only a vague justification of why.

**Requested Changes:**

Beyond the issues noted above, most of which require addressing for me to support acceptance of the paper, there are a few more minor points that should be fixed.

A number of statements early in the paper are rather strange, debatable in correctness, and require revision; in particular:

- "... which explores effective network architecture to learn the underlying distribution …" – this is a surprising misrepresentation of the field, which is concerned with much more than mere 'architectures', but rather with losses, datasets, and of course much more

- "unconditional generation focuses on designing networks that maximize generation quality under constrained data and computational budgets" – there may be a small subset of work with this focus (and the present work may frame itself as such), but the field as a whole deals with both large datasets (e.g. Objaverse-XL) and high compute budgets (e.g. TRELLIS, as this work itself discusses – among many others)

One strand of work – on methods that use 2D data to train (unconditional or conditional) generative models is missing and should be cited in Section 2. In particular:

- Viewset Diffusion [Szymanowicz et al, ICCV 2023]

- Denoising Diffusion via Image-Based Rendering [Anciukevicius et al, ICLR 2024]

- Diffusion with Forward Models [Tewari, 2023]

Those three methods all work in the more challenging setting where only multi-view images are available; yet DD-IBR in particular shows unconditional generation results on CO3D and MVImgNet.

Another method that should be cited in the first paragraph of sec 2 is Sampling 3D Gaussian Scenes in Seconds with Latent Diffusion Models [Henderson et al, 2024], which presents yet another latent diffusion over splats, with a different (image-centric) latent representation; again this works with only posed images as the training data, hence a more challenging setting (and more relevant to the real world).

The discussion of conditional experiments in supplementary should make clear what the train/test splits were, and how exactly the conditioning data was prepared – the discussion is rather brief.

It is mentioned that x0-prediction is used instead of noise-prediction or v-prediction – it would be good however to justify this choice, either empirically or theoretically, since noise- and v-prediction are better on many other tasks.

The first reference in the bibliography [Barthel et al 2024] seems to be duplicated

---

> ### Author Response · Authors · 2026-03-23
> **Brief version of response to Reviewer jFLd**
>
> For the reviewer’s convenience, all revised or newly added contents are marked in blue in both the main paper and the supplementary material.
>
> **Q1. Memorization / overfitting.**
>
> We address this in Sec. I.2 using (1) train/test metrics (Tab. 26) and (2) feature-space interpolation (Fig. 32). While a small train - test gap alone is insufficient, together with smooth interpolation it suggests that the model learns a meaningful latent structure rather than memorizing training samples.
>
> **Q2. Why generation is better than baselines.**
>
> In latent generative modeling, the VAE is critical. In Sec. E.3, we compare against VAE-based baselines (DiffGS, AtlasGaussians) and show that the Trip2GS VAE achieves substantially better reconstruction.
>
> **Q3. Why the method scales from objects to scenes.**
>
> We apply Trip2GS across datasets of very different complexity (ShapeNet, G-Objaverse, 3D-FRONT) without scene-specific tricks. In Sec. E.3, we attribute this versatility to strong VAE reconstruction (Tab. 13) and more diffusion-friendly latents (Tab. 14). We do not claim that mean near 0 and std near 1 prove Gaussianity, but use them as weak indicators of better alignment with the normalized prior used in latent diffusion. Replacing DMTet with SOF and using TV regularization support this interpretation.
>
> **Q4. Differentiability of GIM / end-to-end training.**
>
> We agree this needed clarification. Although SOF is differentiable, in VAE training we intentionally use gradient control and staged training for stability: GIM is trained only with the SOF loss, and early on we use ground-truth points as anchors, so appearance losses do not backpropagate into geometry. We therefore replaced “end-to-end” with “unified pipeline/training.” Our point is that Trip2GS maps input point clouds to 3DGS within one VAE framework, unlike TRELLIS, which trains components separately.
>
> **Q5. TRELLIS comparison.**
>
> During reproduction, we found that TRELLIS implicitly assumes data normalized to [-0.5, 0.5], while our datasets have different intrinsic scales and camera parameters. We therefore aligned TRELLIS preprocessing to each dataset and clarified this in Sec. B. Even after this correction, TRELLIS still underperforms Trip2GS in unconditional generation. It supports the effectiveness of our coarse-to-fine decoding and unified pipeline.
>
> **Q6. Novelty concern.**
>
> We acknowledge that our method is not centered on a single highly innovative module. However, we believe the main contribution is the coarse-to-fine generation scheme itself: instead of directly decoding unstructured 3DGS from a structured latent, we decompose the process into (1) coarse geometry localization (GIM/SOF), (2) fine geometry densification (PUM), and (3) 3DGS decoder. We argue that this decomposition is the key reason why the method remains robust across datasets of very different complexity. In Sec. E, we show that it improves not only reconstruction, but also latent regularity and downstream geometric metrics such as MMD/COV. More broadly, this idea - decoding from structured latents to unstructured outputs through a coarse-to-fine process - is not limited to triplanes or 3DGS, and can naturally extend to other structured latents (e.g., voxels) and other outputs (e.g., point clouds or meshes).
>
> **Q7. Real-world data.**
>
> We agree this is important and added discussion in Sec. H.3. Methods such as DD-IBR and Viewset Diffusion target a different setting centered on real-world 3D novel view generation from multi-view images. Our paper studies unconditional 3D generation from structured 3D assets. In Sec. H.3, we discuss how Trip2GS could be extended to real-world data such as MVImgNet.
>
> **Q8. Conditional generation claims / setup.**
>
> Our goal in Sec. C was to present flexibility of Trip2GS, not a separate main claim. Triplanes are well suited for conditioning because they provide a regular, image-like structured latent. We support geometry, layout, text, and image conditions by modifying only the diffusion stage while keeping the same VAE/decoder. As shown in Tab. 9, Fig. 21, and Fig. 22, our method also outperforms recent text-to-3D methods, including TRELLIS.
>
> **Q9–Q10. Wording and citations.**
>
> We softened the overly strong statements in the Abstract/Introduction and added the missing citations in Related Work.
>
> **Q11. Why x0-prediction?**
>
> Prior works such as EDM and Variational Diffusion Models suggest that it can be a reasonable choice, especially for stable learning in low-data regimes. We also adopted it for fairness, since diffusion baselines in our comparison, including GaussianCube and AtlasGaussians, use clean-sample (x0) prediction as well. Because diffusion is not the main contribution of our paper, we followed the same design for a fair comparison. We nevertheless agree that v-prediction is a promising alternative and may further improve performance in future work.
>
> **Q12. Duplicate citation.**
>
> We corrected the duplicated citation.

---

> ### Author Response · Authors · 2026-04-03
> **Detailed version of response to Reviewer jFLd (Part 1/4)**
>
> For the reviewer’s convenience, all revised or newly added contents are marked in blue in both the main paper and the supplementary material.
>
> **Q1. Memorization issue (concern on single-category overfitting).**
>
> We agree that generalization and memorization are critical issues in 3D generation, and we now discuss this explicitly in **Section I.2**. In our view, memorization occurs when the generative model fails to learn the underlying data distribution and instead overfits to instance-specific features, preventing the formation of a meaningful latent space.
>
> There are two common ways to assess memorization in generative models. The first is to compare generation performance on the **training set** and the **test set**; a large gap may indicate memorization. The second is to inspect the **feature space** itself. If feature interpolations show smooth transitions, this suggests that the model has learned a meaningful latent structure rather than memorizing the training data.
>
> Accordingly, in **Section I.2** we provide the results: **train/test metrics (Tab. 26) and feature-space interpolation (Fig. 32)**. These results consistently indicate that our method does not overfit to the training data and instead learns a meaningful latent space.
>
> **Q2. Autoencoder comparison (why the generation results are better than those of baselines?).**
>
> We thank the reviewer for this question. In latent generative modeling, learning a VAE that compresses 3D assets into a meaningful latent space is critical for strong generation performance.
>
> To analyze this, in **Sec. E.3** we compare the reconstruction performance of our method against baseline methods that also rely on a VAE, specifically **DiffGS** and **AtlasGaussians**. Our results show that the **Trip2GS VAE** achieves substantially better reconstruction quality than these prior approaches. In addition, the ablations in **Section 4.3 (main paper)** and **Section G.2 (appendix)** show that the key reason is our **coarse-to-fine generation strategy** based on **SOF, GIM, and PUM**. We therefore believe that the superior generation quality mainly stems from the stronger latent representation learned by our VAE.

---

> ### Author Response · Authors · 2026-04-03
> **Detailed version of response to Reviewer jFLd (Part 2/4)**
>
> **Q3. What is the core component which makes the method scalable from objects to scenes different with other baselines?**
>
> In this work, we propose **Trip2GS** as a 3D generative pipeline that operates consistently across datasets with very different complexity and granularity, including **ShapeNet, G-Objaverse, and 3D-FRONT**. By “versatility,” we mean that the model can be trained and can generate robustly across progressively more difficult data distributions, from single objects to complex indoor scenes, **without relying on separate tricks specialized for a particular scene type**. We analyze this in **Sec. E.3**, and we interpret this versatility as arising from two main factors.
>
> The first is **strong VAE reconstruction quality (see Tab. 13)**. As shown in our experiments, Trip2GS consistently achieves better reconstruction quality than prior unconditional generation baselines. While reconstruction performance does not alone determine generation quality, it is still important because it reduces the **decoding bottleneck** that limits the final quality reachable by latent diffusion. If the VAE can compress and reconstruct the input more faithfully, then the diffusion model can learn over a latent distribution with less information loss, which in turn leads to better generation quality.
>
> The second is a more **diffusion-friendly latent space (see Tab. 14)**, in terms of both compactness and regularity. In latent diffusion [14], the latent distribution is typically handled under a normalized Gaussian prior, and the mean and standard deviation of the latent directly affect the signal-to-noise behavior during training. In this context, we do not claim that having mean closer to 0 and standard deviation closer to 1 directly proves Gaussianity, but we interpret it as a useful weak indicator that the latent distribution is better aligned with the normalized prior expected by diffusion training. From this perspective, Trip2GS produces a latent space that is easier for diffusion to model than prior methods.
>
> - [14] “High-Resolution Image Synthesis with Latent Diffusion Models”, CVPR 2022
>
> We also make two additional observations. First, replacing **DMTet** with **SOF** leads to a more stable latent scale factor. We interpret this as evidence that SOF reformulates geometry extraction into a more direct **coarse surface support prediction** problem, thereby distributing the burden across a **coarse-to-fine** process instead of forcing both coarse and fine geometry to be encoded at once. Second, **TV regularization** slightly sacrifices VAE reconstruction metrics but moves the latent scale factor closer to 1 and improves diffusion generation quality. This suggests that strong generation quality cannot be explained by reconstruction fidelity alone; **latent regularity also matter**.
>
> Overall, we believe that the versatility of Trip2GS does not come from a scene-specific module, but rather from the combination of **(1) high reconstruction fidelity, which reduces the decoding bottleneck, and (2) a coarse-to-fine latent organization, enabled by SOF, that makes the latent space more stable and diffusion-friendly**. This is, in our view, the key reason why Trip2GS remains robust across datasets of very different complexity.
>
> **Q4. Differentiability of GIM and doubt on end-to-end training (similarity to TRELLIS).**
>
> Thank you for pointing out the point. Although **SOF itself is differentiable**, in the **Trip2GS VAE training stage** we intentionally use a **gradient control strategy** and **staged training** for stability and quality. Specifically, we train **GIM only with the SOF loss**, and in the initial training stage we use **ground-truth points as anchor points**, so that appearance-level losses from the **3DGS decoder** do not flow back into the geometry-level module.
>
> As the reviewer correctly pointed out, this should be distinguished from the usual meaning of **end-to-end training**, where gradients from the final output propagate through the entire model. To avoid confusion, we therefore removed the “end-to-end” wording from the **TRELLIS discussion in Sec. 2 (Related Work)** and from the second paragraph of **Sec. 4.3 (Ablation Study: Intermediate Geometry Representation)**.
>
> More broadly, we believe the term **“unified pipeline/training”** is more accurate than **“end-to-end pipeline/training”**, especially in **Sec. B (Comparison to TRELLIS)**. What we intended to emphasize is that our framework maps **input pointcloud to output 3DGS** within a single VAE training framework, without requiring separately trained geometry and texture models or additional disconnected stages. This clearly differs from **TRELLIS**, where the modules are trained separately and the errors of earlier stages can accumulate and limit the final generation quality, especially in limited-data settings. In contrast, **Trip2GS** performs a more unified **point-to-3DGS** generation within a single model.

---

> ### Author Response · Authors · 2026-04-03
> **Detailed version of response to Reviewer jFLd (Part 3/4)**
>
> **Q5. Clarification on comparison with TRELLIS (mismatch between reconstruction and generation results).**
>
> We thank the reviewer for highlighting this important point. During reproduction, we found that the TRELLIS pipeline implicitly assumes that the data lie within a fixed normalization range (specifically **[-0.5, 0.5]**), and this assumption is effectively hard-coded in the generation pipeline. However, the datasets used in our experiments (**ShapeNet, G-Objaverse, and 3D-FRONT**) have different intrinsic scales and camera parameters. As a result, directly applying the TRELLIS generation pipeline to these datasets led to degraded generation quality.
>
> To ensure a fair comparison, we aligned the TRELLIS preprocessing stage with the intrinsic scale of each dataset, and we clarified these preprocessing details in **Sec. B**.
>
> Even after this correction, however, TRELLIS still underperforms **Trip2GS** in unconditional generation. When the **VAE and diffusion model size** and related hyperparameters (such as the **upsampling ratio**) are made more comparable to ours, TRELLIS shows inferior performance in both **reconstruction** and **generation**, especially on the **bedroom** category. We interpret this as evidence that our **three-stage coarse-to-fine decoding scheme** and **unified pipeline** can learn complex geometry, such as bedroom scenes, more efficiently and effectively. This claim is further supported by **Sec. E.3** (**Tab. 14**), which our method constructs much more compact latent space than TRELLIS.
>
> Detailed experimental results and analysis are provided in **Sec. B**. Due to the revision timeline, results for categories other than bedroom are still being trained, and we will include them within the revision period.
>
> **Q6. While the overall structure of the pipeline is novel and the VAE decoder has some changes versus standard practice, this is essentially a chain of off-the-shelf parts without deep technical innovation. If the results were stronger, that would counterbalance this concern; however, given that they are not, the limited technical substance becomes an issue.**
>
> We respectfully acknowledge that our method may not be based on a highly complicated or individually radical technical component. However, we believe the key contribution of this work is not the introduction of a single highly elaborate module, but rather the proposal of a **new coarse-to-fine generation scheme** that substantially improves **versatility** across datasets of very different scales and complexity.
>
> Experimentally, we show that our method consistently performs better than prior approaches across multiple datasets and across both **object-level** and **scene-level** generation. We further analyze this versatility in **Sec. E**. Across these experiments, the proposed **coarse-to-fine generation scheme**, enabled by **SOF** and **GIM+PUM**, improves not only reconstruction quality but also the compactness and regularity of the latent space (**Sec. E.3**), which we argue is the key reason for the robustness of the method.
>
> More importantly, the central idea of our paper is the **decoding process from structured latents to unstructured outputs**. This idea is not limited to triplanes: it can also be applied to other structured feature representations such as **voxels** (e.g., methods like **TRELLIS/TRELLIS2**), and it is not limited to **3DGS** as the output format, but can also be extended to other unstructured outputs such as **point clouds** or **meshes**. In this paper, we further show that this design leads to better **geometric quality**, including **MMD** and **COV**, than prior methods. We therefore believe that our contribution is a **representation-agnostic coarse-to-fine generation framework** with broader significance than the particular instantiation studied here.
>
> **Q7. Experiments on real-world data.**
>
> We agree that extension to real-world data is an interesting and important direction, and we have added discussion on this point in **Sec. H.3**.
>
> At the same time, we would like to clarify that cited methods such as **DD-IBR** or **Viewset Diffusion** address a somewhat different problem setting. Our paper studies **unconditional 3D generation**, where the model learns a distribution over 3D assets or scenes and directly generates **3D Gaussian representations** from structured 3D latents. In contrast, those prior works mainly focus on **real-world 3D scene inference from multi-view image observations**, often emphasizing **image-conditioned reconstruction or view-consistent scene completion**. Thus, the difference is not merely in dataset choice, but also in the underlying **task, supervision, and evaluation protocol**.
>
> In **Sec. H.3**, we discuss how Trip2GS could be trained on real-world multi-view captures such as **MVImgNet** and how it can be extended toward real-world 3D generation. We believe this supports the claim that the proposed framework is not inherently limited to synthetic data.

---

> ### Author Response · Authors · 2026-04-03
> **Detailed version of response to Reviewer jFLd (Part 4/4)**
>
> **Q8. Claims on conditional generation results and more precise experiment settings.**
>
> We thank the reviewer for pointing this out. Our intention in **Sec. C** was to present conditional generation as an **extension demonstrating the flexibility of the same Trip2GS framework**, rather than as a separate main claim. We believe the triplane representation is particularly suitable for conditional generation because it provides a regular, image-like structured latent space, which makes it easier to incorporate heterogeneous conditions.
>
> Concretely, we support four condition types - geometry (point cloud / voxel grid), layout, text, and image - by modifying only the diffusion stage with condition-specific injection, while keeping the same Trip2GS VAE/decoder pipeline unchanged. Geometry conditions are embedded with condition-specific encoders and injected via AdaGN; layout conditions are injected directly into the xy-plane-aligned triplane; and text/image conditions are incorporated using pretrained CLIP features with cross-attention. This design allows the same structured-to-unstructured coarse-to-fine decoder to be reused across diverse conditional settings. Moreover, as shown in **Tab. 9**, **Fig. 21**, and **Fig. 22**, our method also outperforms recent state-of-the-art text-to-3D generation methods, including TRELLIS, which further demonstrates both the effectiveness of Trip2GS and the strong compatibility of triplane-based latents with diverse conditional generation settings.
>
> **Q9. Some detable statements in the paper.**
>
> We thank the reviewer for pointing this out and agree that several statements in the original manuscript were overly strong.
>
> For the sentence, *“Unconditional 3D generation is a classical task, which explores effective network architecture to learn the underlying distribution of 3D assets,”* we agree that this wording is too narrow. The field is not concerned only with architectures, but more broadly with **representation design, objectives, training strategies, datasets, and computational trade-offs**. In the revision, we soften this sentence in the **first paragraph of the Abstract** to better reflect the scope of the field.
>
> Similarly, for the sentence, *“unconditional generation focuses on designing networks that maximize generation quality under constrained data and computational budgets,”* we agree that this characterization is too strong for the field as a whole. Unconditional 3D generation has been studied in both **limited-resource** and **large-scale/high-compute** settings. Our intention was not to define the entire field in that way, but rather to emphasize the specific focus of **this work**: improving generation quality under constrained data and computational budgets, which we believe remains an important and practically relevant setting. We revise this statement in the **second paragraph of the Introduction** accordingly.
>
> **Q10. Some missing citations.**
>
> We thank the reviewer for these helpful suggestions.
>
> We have added the missing discussion and citations on methods that use **2D data** to train unconditional or conditional generative models, including **Viewset Diffusion** (Szymanowicz et al., ICCV 2023), **Denoising Diffusion via Image-Based Rendering** (Anciukevicius et al., ICLR 2024), and **Diffusion with Forward Models** (Tewari et al., 2023), to the **Related Work** section.
>
> We have also added the citation to **Sampling 3D Gaussian Scenes in Seconds with Latent Diffusion Models** (Henderson et al., 2024).
>
> **Q11. Justification of x0-prediction rather than noise- and v-prediction.**
>
> We agree that one cannot universally claim that **x0-prediction** is always better than **noise-prediction**. However, especially in low-data regimes, x0-prediction can have advantages in terms of **training stability** and **sample quality**, as discussed in prior works such as **EDM** and **Variational Diffusion Models**.
>
> In addition, we followed x0-prediction because recent diffusion-based baselines in our comparison, including **GaussianCube** and **AtlasGaussians**, also use **clean-sample (x0) prediction**. Since diffusion itself is not the main contribution of our paper, we adopted the same design for a fairer comparison.
>
> That said, we agree that more recent works often adopt **v-prediction**, which can provide a better trade-off between noise and clean-sample prediction. We therefore acknowledge that adopting v-prediction may further improve generation quality and is an interesting direction for future work.
>
> **Q12. Citation duplication ([Barthel et al 2024]).**
>
> We thank the reviewer for catching this issue. We have corrected the duplicated citation in the revision.

---

### Author Response · Authors · 2026-04-03
**Additional Clarifications and Detailed Responses**

We sincerely thank the reviewers for their thoughtful feedback.

We hope our rebuttal has clarified the main points and addressed the concerns raised.

For clarity, we will provide a more detailed version of our responses in follow-up comments, expanding on several points that were concise in the initial brief version of rebuttal.

If there are any remaining questions or aspects that would benefit from further clarification, we would be more than happy to discuss them.

We appreciate your time and consideration.

---

### Decision · Action_Editor_EsRC · 2026-05-23

**Recommendation:** Accept as is

**Audience:**

Yes

**Audience Explanation:**

The paper presents a working framework that bridges triplane representations and 3D Gaussian Splatting. Given the high interest in 3D generation and the efficiency of the proposed coarse-to-fine scheme, this work is likely to be of interest to researchers working on generative models and 3D computer vision.

**Claims And Evidence:**

Yes

**Claims Explanation:**

The authors have addressed the concerns regarding overfitting by providing train/test metrics and feature interpolation results. They have also added 3D geometric metrics (MMD/COV, mIoU) to support the geometric quality claims. The revisions have clarified the "unified pipeline" aspect and provided a fairer comparison with TRELLIS by aligning preprocessing and model scales.

---

> ### Author Response · Authors · 2026-06-03
> **Camera-Ready Version Uploaded**
>
> Dear Action Editor,
>
> We sincerely thank you for the positive decision and for the constructive summary of our revisions. We also thank the reviewers for their valuable feedback, which helped us improve the clarity and rigor of the paper.
>
> In the camera-ready version, we have updated the author information and corrected minor typos. We have also reflected the TRELLIS generation results across all categories, which were discussed during the revision process.
>
> Sincerely,
> The Authors